# TIGHT TIME COMPLEXITIES IN PARALLEL STOCHASTIC OPTIMIZATION WITH ARBITRARY COMPUTATION DYNAMICS

**Alexander Tyurin**
AIRI, Moscow, Russia
Skoltech, Moscow, Russia
alexandertiurin@gmail.com

## ABSTRACT

In distributed stochastic optimization, where parallel and asynchronous methods are employed, we establish optimal time complexities under virtually any computation behavior of workers/devices/CPUs/GPUs, capturing potential disconnections due to hardware and network delays, time-varying computation powers, and any possible fluctuations and trends of computation speeds. These real-world scenarios are formalized by our new *universal computation model*. Leveraging this model and new proof techniques, we discover tight lower bounds that apply to virtually all synchronous and asynchronous methods, including Minibatch SGD, Asynchronous SGD (Recht et al., 2011), and Picky SGD (Cohen et al., 2021). We show that these lower bounds, up to constant factors, are matched by the optimal Rennala SGD and Malenia SGD methods (Tyurin & Richtárik, 2023).

## 1 INTRODUCTION

Optimization is one of the main workhorses in machine learning (ML), data science (DS), and federated learning (FL) (Bottou et al., 2018; Kairouz et al., 2021). These fields rely on stochastic optimization methods, with notable examples including the stochastic gradient descent method (SGD) (Robbins & Monro, 1951) and ADAM (Kingma & Ba, 2015), which are considered to be the de facto choices for solving large-scale optimization problems (Schmidt et al., 2021). Due to the computational demands of modern functions, the size of datasets, and the need for data privacy, parallelization and distribution are essential for building efficient systems (Kairouz et al., 2021; Mayer & Jacobsen, 2020). However, it brings many challenges, including *computation heterogeneity*: many workers/CPUs/GPUs/phones work in parallel but with varying computation speeds, fluctuating performance over time, and potential disconnections due to hardware and network delays (Li et al., 2020).

### 1.1 PROBLEM SETUP

Unconstrained smooth optimization problems that arise in ML, DS, and FL are described by

$$\min_{x \in \mathbb{R}^d} f(x), \tag{1}$$

where $f : \mathbb{R}^d \to \mathbb{R}$ with the following standard assumptions:

**Assumption 1.1.** $f$ is differentiable & $L$–smooth, i.e., $\|\nabla f(x) - \nabla f(y)\| \leq L \|x - y\|, \forall x, y \in \mathbb{R}^d$.

**Assumption 1.2.** There exist $f^* \in \mathbb{R}$ such that $f(x) \geq f^*$ for all $x \in \mathbb{R}^d$.

In the nonconvex world, we want to find an $\varepsilon$–stationary point, a (random) vector $\bar{x} \in \mathbb{R}^d$ such that $\mathbb{E}[\|\nabla f(\bar{x})\|^2] \leq \varepsilon$, since, in general, it is intractable to find a global minimum in the nonconvex setting (Nemirovskij & Yudin, 1983; Murty & Kabadi, 1985). We analyze convex functions in Section H.

We consider a problem where workers *do not* have access to the gradients of the function $f$. Instead, they can only calculate stochastic gradients. Such a problem arises when the computation cost of an

exact gradient is huge or even infeasible due to batch normalization (Ioffe & Szegedy, 2015), dropout, random data augmentation, and many other handcrafted and naturally occurring noise sources that do not allow to calculate a gradient (Goodfellow et al., 2016).

Assume that $n$ workers work asynchronously in parallel and calculate stochastic gradients. We focus on two setups:

**Homogeneous setup.** For all $i \in [n]$, worker $i$ has access to an unbiased stochastic gradients $\nabla f(x; \xi)$ with $\sigma^2$-variance-bounded variances, where $\xi$ is a random variable from some distribution $\mathcal{D}$ on $\mathbb{S}_\xi$. In ML and FL, this would mean that all workers have access to the *same data*.

**Assumption 1.3** (Homogeneous setup). For all $i \in [n]$, worker $i$ can only calculate $\nabla f(x; \xi)$ and $\mathbb{E}_\xi[\nabla f(x; \xi)] = \nabla f(x)$ and $\mathbb{E}_\xi[\|\nabla f(x; \xi) - \nabla f(x)\|^2] \le \sigma^2$ for all $x \in \mathbb{R}^d$, where $\sigma^2 \ge 0$.

**Heterogeneous setup.** Unlike the previous setup, we assume that $f(x) = \frac{1}{n} \sum_{i=1}^n f_i(x)$ in this setting, where $f_i : \mathbb{R}^d \to \mathbb{R}$ for all $i \in [n]$, and worker $i$ can only access stochastic gradients $\nabla f_i(x; \xi_i)$ of the local function $f_i$, where $\xi_i$ is a random variable from some distribution $\mathcal{D}_i$ on $\mathbb{S}_\xi$. In ML and FL, this would mean that all workers have access to *different data*.

**Assumption 1.4** (Heterogeneous setup). For all $i \in [n]$, worker $i$ can only calculate $\nabla f_i(x; \xi_i)$, and $\mathbb{E}_{\xi_i}[\nabla f_i(x; \xi_i)] = \nabla f_i(x)$ and $\mathbb{E}_{\xi_i}[\|\nabla f_i(x; \xi_i) - \nabla f_i(x)\|^2] \le \sigma^2$ for all $x \in \mathbb{R}^d$, where $\sigma^2 \ge 0$.

## 1.2 RELATED WORK

**Oracle complexity with *one worker*.** The optimal (dimension-free) *oracle complexity*, # of stochastic gradient calls, and an optimal method are well-known in the homogeneous and heterogeneous setups. In particular, Arjevani et al. (2022); Carmon et al. (2020) showed that the optimal oracle complexity is $O\left(L\Delta/\varepsilon + \sigma^2 L\Delta/\varepsilon^2\right)$ achieved by SGD, i.e., $x^{k+1} = x^k - \gamma \nabla f(x^k; \xi^k)$, where $\xi^k$ are i.i.d. random samples, $\Delta := f(x^0) - f^*$, $x^0 \in \mathbb{R}^d$ is a starting point, $\gamma = \Theta\left(\min\{1/L, \varepsilon/L\sigma^2\}\right)$ is a step size.

**Oracle complexities with $n$ *workers*.** There were several approaches, e.g., (Scaman et al., 2017; Arjevani et al., 2020; Lu & De Sa, 2021), to generalize the classical oracle complexity (Nemirovskij & Yudin, 1983; Arjevani et al., 2022) to the parallel setup with $n$ workers. In the homogeneous convex setup, the most relevant to our setup work (Woodworth et al., 2018), using the *graph-based oracle model*, obtained the tight oracle complexities for several parallel setups. The heterogeneous setup with the local smoothness assumption was addressed in (Arjevani & Shamir, 2015; Hanzely et al., 2020; Lu & De Sa, 2021).

The listed works address key questions in parallel optimization by establishing lower bounds and developing methods to achieve them. However, the assumptions regarding the computation processes are overly idealistic, as they assume stable, unchanging, and equal computation speeds for all workers. They fail to capture practical scenarios such as partial participation, random outages, computation heterogeneity (some workers being faster than others), communication heterogeneity (some workers sending vectors faster than others), and changing computation performance over time. *It is unclear if the optimal methods for their lower bounds will maintain optimality in more realistic computation scenarios.*

**Time complexities with bounded computation times.** Instead of using *oracle complexities*, another way to compare algorithms is to use *time complexities*. Using this paradigm, Mishchenko et al. (2022); Koloskova et al. (2022) showed that the celebrated Asynchronous SGD method (Recht et al., 2011; Dean et al., 2012) with the proposer step sizes can provably improve Minibatch SGD in the homogeneous setup. Assume for now that worker $i$ requires at most $\tau_i$ seconds to calculate one stochastic gradient for all $i \in [n]$. Minibatch SGD is the iterative process $x^{k+1} = x^k - \gamma/n \sum_{i=1}^n \nabla f(x^k; \xi_i^k)$, where $\gamma$ is a stepsize, $\xi_i^k$ are i.i.d. samples, and $\nabla f(x^k; \xi_i^k)$ are calculated in parallel. This method converges after $O\left(L\Delta/\varepsilon + \sigma^2 L\Delta/n\varepsilon^2\right)$ iterations (Cotter et al., 2011; Goyal et al., 2017; Gower et al., 2019) and after

$$O\left(\max_{i\in[n]} \tau_i \times \left(\frac{L\Delta}{\varepsilon} + \frac{\sigma^2 L\Delta}{n\varepsilon^2}\right)\right) \tag{2}$$

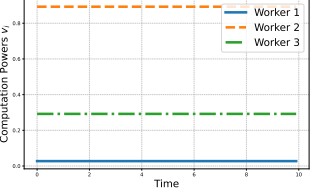

Figure 1: *Fixed Computation Model*: The previous computation paradigm (Mishchenko et al., 2022) assumes that the performances/powers of the workers remain constant over time. Tyurin & Richtárik (2023) established the optimal time complexities (13) and (20) for this paradigm.

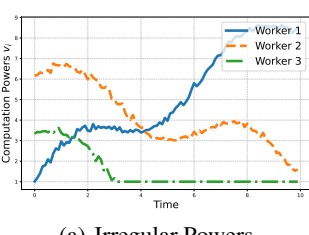 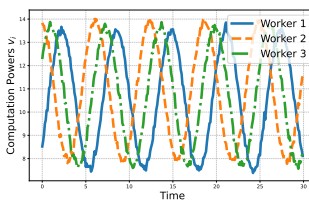 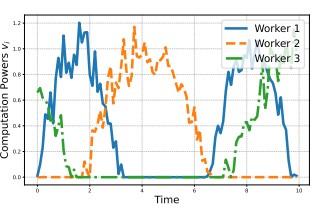

(a) Irregular Powers       (b) Periodic Powers       (c) Random Outages

Figure 2: *Universal Computation Model*: A new computation paradigm that captures virtually all possible computation scenarios. The three subplots present illustrative and non-exhaustive examples of irregular $\{v_i\}$ (Fig. 2(a)), periodic noisy powers $\{v_i\}$ (Fig. 2(b)), and random outages of the workers, where $v_i$ equals 0 periodically (Fig. 2(c)). For all possible scenarios, we establish optimal time complexities (see Theorems 5.1, 5.3, 6.2, and 6.4). It is possible to get interpretable and explicit formulas for the optimal time complexities in some scenarios (see Examples 3.2, 5.4, 5.5, 6.5, and 6.6). However, for Fig. 2(a), Fig. 2(b), and Fig. 2(c), it is arguably intractable to find $\bar{t}_{\lceil L\Delta/\varepsilon \rceil}$ analytically. Instead, we can easily do it numerically in Fig. 2(a) and get the optimal time complexities 6.57 and 13.02 sec with $L\Delta/\varepsilon = 10$ and $\sigma^2/\varepsilon = 100$ in the homogeneous and heterogeneous settings, respectively (Fig. 2(b): 2.34 and 2.53 sec; Fig. 2(c): 77.04 and 84.62 sec).

seconds because this method waits for the slowest worker with the time $\max_{i \in [n]} \tau_i$. Using Asynchronous SGD, the time complexity (2) can be improved to

$$\mathrm{O}\left( \left( \frac{1}{n} \sum_{i=1}^{n} \frac{1}{\tau_i} \right)^{-1} \left( \frac{L\Delta}{\varepsilon} + \frac{\sigma^2 L\Delta}{n\varepsilon^2} \right) \right), \tag{3}$$

where the dependence on the processing times is harmonic (Mishchenko et al., 2022). An alternative method that also achieves this time complexity is Picky SGD (Cohen et al., 2021).

Subsequently, Tyurin & Richtárik (2023) formalized the notion of time complexity using the *time oracle protocol*, and under the assumption that worker $i$ requires at most $\tau_i$ seconds to calculate one stochastic gradient for all $i \in [n]$, proved that the time complexity lower bound is

$$\Theta\left( \min_{m \in [n]} \left[ \left( \frac{1}{m} \sum_{i=1}^{m} \frac{1}{\tau_{\pi_i}} \right)^{-1} \left( \frac{L\Delta}{\varepsilon} + \frac{\sigma^2 L\Delta}{m\varepsilon^2} \right) \right] \right) \tag{4}$$

seconds in the homogeneous setup, where $\pi$ is a permutation that sorts $\tau_i : \tau_{\pi_1} \leq \cdots \leq \tau_{\pi_n}$. Moreover, they developed a new method, Rennala SGD, that achieves the lower bound in the homogeneous setup. Under the bounded computation and communication assumptions, Tyurin et al. (2024b); Tyurin & Richtárik (2024) provided optimal time complexities in a setup where the communication time between the workers cannot be ignored.

## 2 CONTRIBUTIONS

In this work, we aim to determine the optimal time complexities of distributed stochastic optimization with parallel and asynchronous methods in scenarios where the computational performance of workers can be arbitrarily heterogeneous and variable. We want to capture all possible cases, including random outages, time-changing computation performances, and slow and straggler workers.

♠ We consider a new computation paradigm, for which we coin the name *universal computation model*, that includes virtually all possible computation scenarios that can appear in practical distributed, parallel, and asynchronous optimization environments.

♣ Using the universal computation model, we prove a tight lower bound for time complexities of parallel and asynchronous optimization methods in the homogeneous setting, which is matched, up to constants, by our Theorem 5.3, saying that Rennala SGD (Tyurin & Richtárik, 2023) is optimal.

♦ We also close the problem in the heterogeneous setup. We discover the optimal time complexity and prove the optimality of Malenia SGD (Tyurin & Richtárik, 2023) (see Theorem 6.4). The proofs of the lower bounds are based on new proof techniques and constructions (see Section A for an overview).

♥ In Section H, we provide time complexities of (Accelerated) Rennala SGD and (Accelerated) Malenia SGD in the *convex setting*.

## 3  Universal Computation Model

To achieve the goal of finding the optimal time complexities for the stochastic optimization problem under any heterogeneous asynchronous computation setup, we first have to formalize the computation model of the workers. To formalize all possible cases, we propose using the following computation model, called the *universal computation model*.

For all $i \in [n]$, we consider a non-negative *continuous almost everywhere* function $v_i : \mathbb{R}_+ \to \mathbb{R}_+$[1] called a *computation power* of worker $i$.

**Assumption 3.1.** For all $i \in [n]$, $v_i$ is non-negative continuous almost everywhere.

Without loss of generality, we assume that the time starts from *zero*. Using the same reasoning as in physics, where the energy is the integral of power, in our domain, the number of stochastic gradients that worker $i$ can calculate from a time $t_0$ to a time $t_1$ is the Riemann integral[2] of the computation power $v_i$ followed by the floor operation (because we can not partially calculate a stochastic gradient):

$$\text{``\# of stoch. grad. by worker } i \text{ in } [t_0, t_1]\text{''} = \left\lfloor \int_{t_0}^{t_1} v_i(\tau)d\tau \right\rfloor = \lfloor V_i(t_1) - V_i(t_0) \rfloor, \qquad (5)$$

where we additionally define a mapping $V_i : \mathbb{R}_+^\infty \to \mathbb{R}_+^\infty$ such that

$$V_i(t) := \int_0^t v_i(\tau)d\tau. \qquad (6)$$

For $t_1 \geq t_0$, $V_i(t_1) - V_i(t_0)$ is called a *computation work* of worker $i$ from a time $t_0$ to a time $t_1$.

**Example 3.2** (Fixed Computation Model). Let us consider the simplest example and take the performances that do not change through time, i.e., $v_i(t) = v_i \in \mathbb{R}_+$ for all $t \geq 0$. If we take $t_0 = 0$, then

$$\left\lfloor \int_{t_0}^{t_1} v_i(\tau)d\tau \right\rfloor = \lfloor V_i(t_1) - V_i(t_0) \rfloor = \lfloor v_i t_1 \rfloor. \qquad (7)$$

This formula formalizes simple logic that it takes $1/v_i$ seconds to find one stochastic gradient in worker $i$ because $\lfloor v_i \times 1/v_i \rfloor = 1$, $2/v_i$ seconds to find two stochastic gradients in worker $i$ because $\lfloor v_i \times 2/v_i \rfloor = 2$, and so forth. The higher the power $v_i$, the less time it takes to find a new stochastic gradient. (7) provides the number of stochastic gradients under the fixed computation model when reparametrized with $v_i = 1/\tau_i$. However, the fixed computation model is limited; for a visual comparison, see Figures 1 and 2. Section 5.1 has more examples.

**Theorem 3.3** (e.g. (Bartle & Sherbert, 2000)). *For all $i \in [n]$, $V_i$ is continuous and non-decreasing on $\mathbb{R}_+$ if $v_i$ is non-negative continuous almost everywhere (Assumption 3.1).*

---

[1] Notations: $\mathbb{R}_+ := [0, \infty)$, $\mathbb{R}_+^\infty := [0, \infty]$, $\mathbb{N} := \{1, 2, 3, \dots\}$, $\mathbb{N}_0 := \{0, 1, 2, \dots\}$, $[n] := \{1, \dots, n\}$.

[2] It is possible to consider the Lebesgue integral and assume that the functions $\{v_i\}$ are measurable, but we will stick with the Riemann integral for simplicity.

Notice this theorem can hold even if $v_i$ is *discontinuous*. In general, the computation powers $\{v_i\}$ can be even random. Indeed, we can assume that $v_i : \mathbb{R}_+ \times \Omega \to \mathbb{R}_+$ is a *stochastic computation power*, where $\Omega$ a sample space of a probability space. Then, all the following results hold when conditioned over all randomness in $\{v_i\}$, assuming that the sources of randomness are statistically independent. Without loss of generality, we continue assuming that $v_i : \mathbb{R}_+ \to \mathbb{R}_+$ for all $i \in [n]$. For all $i \in [n]$, let us define the generalized inverse function[3] $V_i^{-1} : \mathbb{R}_+^\infty \to \mathbb{R}_+^\infty$ such that

$$V_i^{-1}(S) = \min\{t \geq 0 \,:\, V_i(t) = S\} \tag{8}$$

for all $S \in \mathbb{R}_+^\infty$. If $V_i$ is strongly increasing, then $V_i^{-1}$ is the standard inverse function of $V_i$.

## 4 PRELIMINARIES

Before presenting our main results, we formalize the concepts of time and time complexity. This section is somewhat technical; readers not interested in the proof details may skip ahead to Sections 5 and 6.

Recall that in the classical approach of deriving lower bounds, we examine the following protocol (Nemirovskij & Yudin, 1983; Nesterov, 2018; Carmon et al., 2020):

---
**Protocol 1** Classical Oracle Protocol

---
1: **Input:** function $f \in \mathcal{F}$, oracle $O \in \mathcal{O}(f)$, algorithm $A \in \mathcal{A}$
2: **for** $k = 0, \ldots, \infty$ **do**
3:      $x^k = A^k(g^1, \ldots, g^k)$                         $\triangleright\ x^0 = A^0$ for $k = 0$.
4:      $g^{k+1} = O(x^k)$
5: **end for**

---

Where we want to find the worst-case oracle complexity formalized by

$$\inf_{A \in \mathcal{A}} \sup_{f \in \mathcal{F}} \sup_{O \in \mathcal{O}(f)} \inf\left\{k \in \mathbb{N} \,\middle|\, \mathbb{E}\left[\left\|\nabla f(x^k)\right\|^2\right] \leq \varepsilon\right\}.$$

Protocol 1 is a reasonable way to establish lower bounds and compare algorithms, but it is not convenient for analyzing parallel algorithms. In order to analyze parallel and asynchronous algorithms, Tyurin & Richtárik (2023) proposed to use the time multiple oracles protocol:

---
**Protocol 2** Time Multiple Oracles Protocol

---
1: **Input:** function $f$ (or functions $f_i$), oracles $\{O_i\}_{i=1}^n \in \mathcal{O}(f)$, algorithm $A = \{A^k\}_{k=0}^\infty \in \mathcal{A}$
2: $s_i^0 = 0$ for all $i \in [n]$
3: **for** $k = 0, \ldots, \infty$ **do**
4:      $(t^{k+1}, i^{k+1}, c^{k+1}, x^k) = A^k(g^1, \ldots, g^k)$             $\triangleright\ t^{k+1} \geq t^k$
5:      $(s_{i^{k+1}}^{k+1}, g^{k+1}) = O_{i^{k+1}}(t^{k+1}, x^k, s_{i^{k+1}}^k, c^{k+1})$     $\triangleright \forall j \neq i^{k+1} : s_j^{k+1} = s_j^k$
6: **end for**

---

Unlike the classical protocol where an algorithm returns a new point $x^k$ based on the current information $g^1, \ldots, g^k$, this protocol requires an algorithm to return a time $t^{k+1}$, an index of an oracle $i^{k+1}$, a control variable $c^{k+1}$, and a new point $x^k$. In the parallel setting, algorithms have access to many workers/oracles. In every iteration, an algorithm has the freedom to choose any oracle using $i^{k+1}$, and call the oracle at a point $x^k$. The role of control variables $\{c^{k+1}\}$ will be clear later.

*The main idea is that an algorithm controls time and decides when it is ready to go forward using a time sequence $\{t^{k+1}\}$.* Let us introduce an oracle that emulates the behavior of a real worker, and then we will provide clarifications. For all $i \in [n]$, we consider the mapping

---
[3]We use the standard convention $\min\{\emptyset\} = \infty$.

$$O_i : \underbrace{\mathbb{R}_+}_{\text{time}} \times \underbrace{\mathbb{R}^d}_{\text{point}} \times \underbrace{(\mathbb{R}_+ \times \mathbb{R}^d \times \{0,1\})}_{\text{input state}} \times \underbrace{\{0,1\}}_{\text{stop computation}} \to \underbrace{(\mathbb{R}_+ \times \mathbb{R}^d \times \{0,1\})}_{\text{output state}} \times \underbrace{\mathbb{R}^d}_{\text{output vector}} \quad \text{such that}$$

$$O_i(t, x, (s_t, s_x, s_q), c) = \begin{cases} ((t, x, 1), & 0), & c = 0, s_q = 0, \\ ((s_t, s_x, 1), & 0), & c = 0, s_q = 1, V_i(t) - V_i(s_t) < 1, \\ ((0, 0, 0), & g_i(s_x; \xi, t)), & c = 0, s_q = 1, V_i(t) - V_i(s_t) \geq 1, \\ ((0, 0, 0), & 0), & c = 1, \end{cases} \quad (9)$$

where $\mathcal{D}_{s_x,t,i}$ is some distribution that can depend on $s_x, t$, $\xi \sim \mathcal{D}_{s_x,t,i}$, and $g_i : \mathbb{R}^d \times \mathbb{S}_\xi \times \mathbb{R}_+ \to \mathbb{R}^d$ is a mapping. This oracle can return different outputs depending on an input it receives: i) if $c = 0, s_q = 0$, then the oracle is only starting the calculation of a stochastic gradient, and it memorizes the time when it was called in the variable $s_t$; ii) if $c = 0, s_q = 1, V_i(t) - V_i(s_t) < 1$, then the oracle is still calculating; iii) if $c = 0, s_q = 1, V_i(t) - V_i(s_t) \geq 1$, then the oracle has finished the calculation and returns $g_i(s_x; \xi, t)$ at the point $s_x$ where the calculation was initialized. The condition $V_i(t) - V_i(s_t) \geq 1$ means that "at the current time $t$, the oracle is ready to return the stochastic gradient that began to be calculated at time $s_t$." The control variable $c$ allows algorithms to stop the calculations at any time they want if they pass $c = 1$.

The oracles (9) force algorithms to increase times; otherwise, they will not get stochastic gradients and enough information to find an $\varepsilon$–stationary point. Using Protocol 2, we consider the time complexity measure

$$\inf_{A \in \mathcal{A}} \sup_{f \in \mathcal{F}} \sup_{(O,\mathcal{D}) \in \mathcal{O}(f)} \inf \left\{ t \geq 0 \,\middle|\, \mathbb{E}\left[ \inf_{k \in S_t} \left\| \nabla f(x^k) \right\|^2 \right] \leq \varepsilon \right\}, S_t := \left\{ k \in \mathbb{N}_0 \,\middle|\, t^k \leq t \right\} \quad (10)$$

where the sequences $t^k$ and $x^k$ are generated by Protocol 2. This measure takes algorithm and function classes and returns the worst-case time complexity. We refer to (Tyurin & Richtárik, 2023)[Sections 3–5] for more details.

In this work, we consider zero-respecting algorithms formalized by the definition below.

**Definition 4.1** (Algorithm Class $\mathcal{A}_{\text{zr}}$). Let us consider Protocol 2. We say that an algorithm $A = \{A^k\}_{k=0}^\infty$ is a zero-respecting algorithm, if

1. $A^k : \underbrace{\mathbb{R}^d \times \cdots \times \mathbb{R}^d}_{k \text{ times}} \to \mathbb{R}_{\geq 0} \times [n] \times \{0,1\} \times \mathbb{R}^d \quad \forall k \geq 1, A^0 \in \mathbb{R}_{\geq 0} \times [n] \times \{0,1\} \times \mathbb{R}^d,$

2. $\text{supp}\left(x^k\right) \subseteq \bigcup_{j=1}^k \text{supp}\left(g^j\right)$ for all $k \in \mathbb{N}_0$, where $\text{supp}(x) := \{i \in [d] \mid x_i \neq 0\}$,

3. for all $k \geq 1$ and $g^1, \ldots, g^k \in \mathbb{R}^d$, we have $t^{k+1} \geq t^k$, where $t^{k+1}$ and $t^k$ are defined as $(t^{k+1}, \cdot) = A^k(g^1, \ldots, g^k)$ and $(t^k, \cdot) = A^{k-1}(g^1, \ldots, g^{k-1})$.

The first condition defines the domain and range, the second condition is the definition of a zero-respecting algorithm (Carmon et al., 2020), the third condition ensures that the algorithm return a non-decreasing sequence of times (Tyurin & Richtárik, 2023).

## 5 HOMOGENEOUS SETUP

We are ready to present our lower bound in the homogeneous setup. Let us also provide a simplified and informal version of the theorem, followed by the formal one.

**Theorem** (Informal Formulation of Theorem 5.1). *Let Assumptions 1.1, 1.2, 1.3, and 3.1 hold. It is impossible to converge faster than $1/2 \times \underline{t}_{\lceil c_1 \times \frac{L\Delta}{\varepsilon} \rceil}$ seconds, where the sequence $\{\underline{t}_k\}$ is defined in (11) and $c_1$ is a universal constant.*

**Theorem 5.1.** *Consider Protocol 2. We take computation powers $\{v_i\}$ such that Assumption 3.1 holds, and fix $L, \Delta, \varepsilon > 0$ and $\sigma^2 \geq 0$ that satisfy the inequality $\varepsilon < c' L\Delta$. For any algorithm $A \in \mathcal{A}_{\text{zr}}$, there exist a function $f$, which satisfy Assumptions 1.1, 1.2 and $f(0) - f^* \leq \Delta$, and stochastic gradient mappings $\{g_i\}$ in (9), which satisfy Assumption 1.3, i.e., $\mathbb{E}_\xi [g_i(s_x; \xi, t)] = \nabla f(s_x)$ and*

| **Method 3** Rennala SGD | **Method 4** Malenia SGD |
|---|---|
| 1: **Input:** point $x^0$, stepsize $\gamma$, batch size $S$ | 1: **Input:** point $x^0$, stepsize $\gamma$, parameter $S$ |
| 2: **for** $k = 0, 1, \ldots, K - 1$ **do** | 2: **for** $k = 0, 1, \ldots, K - 1$ **do** |
| 3: Ask all workers to calculate stochastic gradients at $x^k$ | 3: Ask all workers to calculate stochastic gradients at $x^k$ |
| 4: Init $g^k = 0$ and $s = 1$ | 4: Init[a] $g^k = 0$ and $B_i = 0$ |
| 5: **while** $s \leq S$ **do** | 5: **while** $\left(\frac{1}{n}\sum_{i=1}^{n}\frac{1}{B_i}\right)^{-1} < \frac{S}{n}$ **do** |
| 6: Wait for the next worker | 6: Wait for the next worker $j$ |
| 7: Receive a calculated stochastic gradient $\nabla f(x^k; \xi_s^k)$ | 7: Update $B_j = B_j + 1$ |
| 8: $g^k = g^k + \frac{1}{S}\nabla f(x^k; \xi_s^k); \quad s = s + 1$ | 8: Receive a calculated stochastic gradient $\nabla f_j(x^k; \xi_{j,B_j}^k)$ |
| 9: Ask this worker to calculate a stochastic gradient at $x^k$ | 9: $g_j^k = g_j^k + \nabla f_j(x^k; \xi_{j,B_j}^k)$ |
| 10: **end while** | 10: Ask this worker to calculate a stochastic gradient at $x^k$ |
| 11: $x^{k+1} = x^k - \gamma g^k$ | 11: **end while** |
| 12: Stop all the workers' calculations | 12: $g^k = \frac{1}{n}\sum_{i=1}^{n}\frac{1}{B_i}g_i^k$ |
| 13: **end for** | 13: $x^{k+1} = x^k - \gamma g^k$ |
| (In practice, instead of $x^{k+1} = x^k - \gamma g^k$ (Line 11), one can use any other update technique, including ADAM (Kingma & Ba, 2015), AdaGrad (Duchi et al., 2011), and SGD with momentum (Polyak, 1964; Nesterov, 1983)) | 14: Stop all the workers' calculations |
| | 15: **end for** |
| | (a): In practice, worker $i$ can store $g_i^k$ |

$\mathbb{E}_{\xi}[\|g_i(s_x; \xi, t) - \nabla f(s_x)\|^2] \leq \sigma^2$ *for all* $s_x \in \mathbb{R}^d, i \in [n]$ *such that* $\mathbb{E}\left[\inf_{k \in S_t}\|\nabla f(x^k)\|^2\right] > \varepsilon$
*holds, where* $S_t := \{k \in \mathbb{N}_0 \,|\, t^k \leq t\}$,

$$t = \frac{1}{2} \times \underline{t}_{\lfloor c_1 \times \frac{L\Delta}{\varepsilon}\rfloor},$$

*and[4]*

$$\underline{t}_k := \min\left\{t \geq 0 \,:\, \sum_{i=1}^{n}\lfloor V_i(t) - V_i(\underline{t}_{k-1})\rfloor \geq c_2 \times \max\left\{\left\lceil\frac{\sigma^2}{\varepsilon}\right\rceil, 1\right\}\right\} \qquad (\underline{t}_0 = 0) \qquad (11)$$

*for all* $k \geq 0$. *The quantities* $c', c_1$, *and* $c_2$ *are universal constants. The sequences* $x^k$ *and* $t^k$ *are defined in Protocol* 2.

Unlike most previous works (e.g., (Nesterov, 2018; Arjevani et al., 2022; Tyurin & Richtárik, 2023)), the obtained lower bound is *implicit*. This, we believe, is expected due to the generality of our assumptions about the universal computation model. To find the lower bound, one must determine the minimum of the set in (11) one by one $(\underline{t}_1, \underline{t}_2, \underline{t}_3, \ldots)$ to get $1/2 \times \underline{t}_{\lfloor c_1 \times L\Delta/\varepsilon\rfloor}$. Computationally, this problem is not difficult since the function $\sum_{i=1}^{n}\lfloor V_i(t) - V_i(\underline{t}_{k-1})\rfloor$ is non-decreasing; thus, for instance, we can employ the bisection method.

## 5.1 OPTIMAL ALGORITHM

The natural question is whether the lower bound is tight. The answer is yes (as usual, up to constant factors). The lower bound can be matched by Rennala SGD (Method 3) (Tyurin & Richtárik, 2023). In every iteration, the method collects a batch of size $S$, and performs a gradient-like step once the batch has been collected. The following result was proved in (Tyurin & Richtárik, 2023). The proof technique is simple and follows the classical analysis of SGD (Ghadimi & Lan, 2013; Khaled & Richtárik, 2022) since the logic of Rennala SGD is equivalent to the steps $x^{k+1} = x^k - \gamma/S\sum_{i=1}^{S}\nabla f(x^k; \xi_i)$, where $\{\xi_i\}$ are i.i.d. samples.

**Theorem 5.2.** *[(Tyurin & Richtárik, 2023)] Let Assumptions 1.1, 1.2, and 1.3 hold. We take* $\gamma = 1/2L$ *and batch size* $S = \max\{\lceil\sigma^2/\varepsilon\rceil, 1\}$ *in Method 3. For all* $K \geq 24L\Delta/\varepsilon$, *we get* $\frac{1}{K}\sum_{k=0}^{K-1}\mathbb{E}\left[\|\nabla f(x^k)\|^2\right] \leq \varepsilon$.

---

[4]We use the standard convention $\min\{\emptyset\} = \infty$.

The following result is new and proves the time complexity of Rennala SGD.

**Theorem 5.3.** *Consider the assumptions and the parameters from Theorem 5.2, plus Assumption 3.1. Then Method 3 (*Rennala SGD*) converges after at most* $\bar{t}_{\lceil \frac{24L\Delta}{\varepsilon} \rceil}$ *seconds, where*

$$\bar{t}_k := \min \left\{ t \geq 0 \,:\, \sum_{i=1}^{n} \lfloor V_i(t) - V_i(\bar{t}_{k-1}) \rfloor \geq \max \left\{ \left\lceil \frac{\sigma^2}{\varepsilon} \right\rceil, 1 \right\} \right\} \qquad (\bar{t}_0 \equiv 0) \quad \forall k \geq 1. \quad (12)$$

Up to constant factors, Theorem 5.1 together with Theorem 5.3 provide the tight time complexity for the problem (1) in the homogeneous setup. As we noted in Section 5, the obtained result is implicit, we do not get a closed-form expression for $\bar{t}_{\lceil 24L\Delta/\varepsilon \rceil}$ in Theorem 5.3. That said, the sequence $\bar{t}_k$ is mathematically rigorous, and we can provide explicit formulas in some cases. Surprisingly, Rennala SGD gets the optimal complexity automatically, without prior knowledge about the computation powers $\{v_i\}$. Therefore, the absence of a closed-form expression is irrelevant for practical implementations and is only of theoretical interest.

**Example 5.4.** [Fixed Computation Model] Consider Example 3.2 with $v_i(t) = v_i \in \mathbb{R}_+$ for all $t \geq 0, i \in [n]$. Then, for all $i \in [n]$, $V_i(t) = v_i t$ and

$$\bar{t}_{\lceil \frac{24L\Delta}{\varepsilon} \rceil} = \Theta \left( \min_{m \in [n]} \left( \frac{1}{m} \sum_{i=1}^{m} v_{\pi_i} \right)^{-1} \left( \frac{L\Delta}{\varepsilon} + \frac{L\Delta\sigma^2}{m\varepsilon^2} \right) \right), \quad (13)$$

$\pi$ is a permutation such that $v_{\pi_1} \geq \cdots \geq v_{\pi_n}$. The proofs of the examples are in Section I.

In Example 3.2, we discuss that $\tau_i = 1/v_i$ is the time required to find one stochastic gradient in worker $i$. If we reparametrize (13) with $v_i = 1/\tau_i$, then we get the time complexity (4). Thus, Example 5.4 restores the optimal time complexity obtained by Tyurin & Richtárik (2023) for the fixed computation model, where the smaller the computation times $\tau_i$ (the higher the computation powers $v_i$), the smaller the complexities. Notice that if $v_j$ is small enough for some worker $j$, then it is possible that the complexity (13) will not depend on $v_j$, meaning that this worker potentially does not contribute to an optimization process because it is too slow. We can immediately derive a more general result:

**Example 5.5.** [Nonlinear Trend] Assume that $v_i(t) = v_i \times g(t)$ with $v_i > 0$ for all $i \in [n]$ and a continuous almost everywhere positive[5] function $g(t) : \mathbb{R}_+^\infty \to \mathbb{R}_+$. Then

$$\bar{t}_{\lceil \frac{24L\Delta}{\varepsilon} \rceil} = G^{-1} \left( c_1 \cdot \min_{m \in [n]} \left( \frac{1}{m} \sum_{i=1}^{m} v_{\pi_i} \right)^{-1} \left( \frac{L\Delta}{\varepsilon} + \frac{L\Delta\sigma^2}{m\varepsilon^2} \right) \right), \quad (14)$$

where $\pi$ is a permutation such that $v_{\pi_1} \geq \cdots \geq v_{\pi_n}$, $G(t) := \int_0^t g(\tau)d\tau$, and $c_1 \in [1/4, 4]$ (can depend on other parameters but is bounded).

Example 5.5 illustrates many practical cases. For example, the computation powers can vary according to the function $g(t) = 1.01 + \sin(t)$, causing them periodically increase and decrease. Then $G(t) = 1.01t - \cos t + 1$, which is invertible, and we can obtain a formula for the optimal time complexity using (14).

Let us consider an example where all workers have the same performances, but any worker can randomly shut down and, after a while, become available again. We could have chosen virtually any (even random) example, but for the sake of simplicity, let us consider the following example to gain a basic intuition.

**Example 5.6.** ["Random" Outages] Assume that

$$v_i(t) = \begin{cases} v, & t \in \bigcup_{j=1}^{\infty} [k_i(j-1), (k_i(j-1)+1)] \\ 0, & \text{otherwise}, \end{cases} \quad (15)$$

$v > 0$, $k_i \in \mathbb{N}$, and $h_i > 0$ for all $i \in [n]$. Then

$$\bar{t}_{\lceil \frac{24L\Delta}{\varepsilon} \rceil} \approx \Theta \left( \min_{m \in [n]} \left( \frac{1}{m} \sum_{i=1}^{m} \frac{v}{k_{\pi_i}} \right)^{-1} \left( \frac{L\Delta}{\varepsilon} + \frac{L\Delta\sigma^2}{m\varepsilon^2} \right) \right), \quad (16)$$

where $\pi$ is a permutation such that $k_{\pi_1} \leq \cdots \leq k_{\pi_n}$.

---

[5] We can relax these assumptions to *measurability and non-negativity*, but the proof will be more technical.

In this example, worker $i$ is active in the time intervals $[0, 1]$, $[k_j, k_j + 1]$, $[2k_j, 2k_j + 1]$, and so forth. The parameter $k_j$ characterizes how often the worker's outages occur.

The formula (16) says that the more worker $i$ is inactive (the larger $k_i$), the more time it takes to solve the problem. Due to the $\min$ operation in (16), the formula indicates that some workers can be ignored if their $k_i$ are too large. In general, we could have analyzed (15) with $\bigcup_{j=1}^{\infty} [\text{start}_{k,j}, \text{end}_{k,j}]$, where the pairs $\{\text{start}_{k,j}, \text{end}_{k,j}\}$ are arbitrarily (random) values on $\mathbb{R}_+$, but would get less interpretable formulas.

## 6 HETEROGENEOUS SETUP

We now consider the heterogeneous setup discussed in Section 1, and present our first lower bound:

**Theorem 6.1.** *Consider Protocol 2. We take computation powers $\{v_i\}$ such that Assumption 3.1 holds, and fix $L, \Delta, \varepsilon > 0$ and $\sigma^2 \geq 0$ that satisfy the inequality $\varepsilon < c' L \Delta$. For any algorithm $A \in \mathcal{A}_{\text{zr}}$, there exist functions $\{f_i\}_{i=1}^n$, where the function $f = \frac{1}{n} \sum_{i=1}^n f_i$ satisfies Assumptions 1.1, 1.2 and $f(0) - f^* \leq \Delta$, and stochastic gradient mappings $\{g_i\}_{i=1}^n$ in (9), which satisfy Assumption 1.4, i.e., $\mathbb{E}_\xi [g_i(s_x; \xi, t)] = \nabla f_i(s_x)$ and $\mathbb{E}_\xi[\|g_i(s_x; \xi, t) - \nabla f_i(s_x)\|^2] \leq \sigma^2$ for all $s_x \in \mathbb{R}^d, i \in [n]$, and $t \geq 0$, such that $\mathbb{E}\left[ \inf_{k \in S_t} \|\nabla f(x^k)\|^2 \right] > \varepsilon$ holds, where $S_t := \{k \in \mathbb{N}_0 \mid t^k \leq t\}$,*

$$t = \tfrac{1}{2} \underline{t}_{\left\lfloor \frac{c_1 \times \frac{L\Delta}{\varepsilon}}{\log \frac{L\Delta}{\varepsilon}} \right\rfloor}$$

*and*

$$\underline{t}_k := \min \left\{ t \geq 0 : \left( \frac{1}{n} \sum_{i=1}^n \left\lfloor c_3 \times \frac{V_i(t) - V_i(\underline{t}_{k-1})}{\log\left(\frac{L\Delta}{\varepsilon}\right)} \right\rfloor^{-1} \right)^{-1} \geq \max\left\{ c_2 \times \frac{\sigma^2}{n\varepsilon}, 1 \right\} \right\} \quad (17)$$

*for all $k \geq 1$ ($\underline{t}_0 \equiv 0$). The quantities $c', c_1, c_2$, and $c_3$ are universal constants. The sequences $x^k$ and $t^k$ are defined in Protocol 2.*

Unlike (12) where the dependencies on $\{V_i\}$ are *mean-like*, the dependencies in (17) are *harmonic-like*. Since the heterogeneous setting is more general and complicated, this leads to worse guarantees. Looking ahead, up to logarithmic and constants factors, the obtained lower bound is tight and attained by Malenia SGD (Tyurin & Richtárik, 2023) (see Section 6.1).

We asked ourselves if getting a tight lower bound without the logarithmic terms is possible. The answer is affirmative, but instead of taking one group of predefined worst-case deterministic functions $\{f_i\}$, the following construction samples random functions $\{f_i\}$. The fact that the functions $\{f_i\}_{i=1}^n$ are random helps to prove a tight lower bound (the main difference between the theorems is highlighted in **bold**). This lower bound is fundamental and can not be bypassed by any parallel and asynchronous method (Zheng et al., 2017; Gu et al., 2021; Mishchenko et al., 2022; Islamov et al., 2024). We provide informal and formal versions of the theorem:

**Theorem** (Informal Formulation of Theorem 6.2)**.** *Let Assumptions 1.1, 1.2, 1.4, and 3.1 hold. It is impossible to converge faster than $1/2 \times \underline{t}_{\lceil c_1 \times \frac{L\Delta}{\varepsilon} \rceil}$ seconds, where the sequence $\{\underline{t}_k\}$ is defined in (18) and $c_1$ is a universal constant.*

**Theorem 6.2.** *Consider Protocol 2. We take computation powers $\{v_i\}$ such that Assumption 3.1 holds, and fix $L, \Delta, \varepsilon > 0$ and $\sigma^2 \geq 0$ that satisfy the inequality $\varepsilon < c' L \Delta$. For any algorithm $A \in \mathcal{A}_{\text{zr}}$, **we sample** $\{f_i\}_{i=1}^n$ **from some distribution of functions**, where the function $f = \frac{1}{n} \sum_{i=1}^n f_i$ satisfies Assumptions 1.1, 1.2 and $f(0) - f^* \leq \Delta$ **deterministically**, and there exist stochastic gradient mappings $\{g_i\}_{i=1}^n$ in (9), which satisfy Assumption 1.4, i.e., $\mathbb{E}_\xi [g_i(s_x; \xi, t)] = \nabla f_i(s_x)$ and $\mathbb{E}_\xi[\|g_i(s_x; \xi, t) - \nabla f_i(s_x)\|^2] \leq \sigma^2$ for all $s_x \in \mathbb{R}^d, i \in [n]$, and $t \geq 0$, such that $\mathbb{E}\left[ \inf_{k \in S_t} \|\nabla f(x^k)\|^2 \right] > \varepsilon$ holds[6], where $S_t := \{k \in \mathbb{N}_0 \mid t^k \leq t\}$,*

$$t = \tfrac{1}{2} \underline{t}_{\left\lfloor c_1 \times \frac{L\Delta}{\varepsilon} \right\rfloor}$$

---

[6]We take the expectation over all randomness.

*and*

$$\underline{t}_k := \min \left\{ t \geq 0 \,:\, \left( \tfrac{1}{n} \sum_{i=1}^{n} \left\lfloor c_3 \times \left( V_i(t) - V_i(\underline{t}_{k-1}) \right) \right\rfloor^{-1} \right)^{-1} \geq \max \left\{ c_2 \times \tfrac{\sigma^2}{n\varepsilon}, 1 \right\} \right\} \quad (18)$$

*for all $k \geq 1$ ($\underline{t}_0 \equiv 0$). The quantities $c'$, $c_1$, $c_2$, and $c_3$ are universal constants. The sequences $x^k$ and $t^k$ are defined in Protocol 2.*

## 6.1 Optimal method

The obtained lower bound is tight since it is matched by Malenia SGD (Tyurin & Richtárik, 2023). This method is closely related to Rennala SGD with a similar structure, and mathematically, it is the vanilla SGD method with a proper batch collection strategy (see Method 3). However, essential algorithmic changes must be applied to make it work with heterogeneous functions. The following theorem was proved by Tyurin & Richtárik (2023).

**Theorem 6.3.** *[(Tyurin & Richtárik, 2023)] Let Assumptions 1.1, 1.2, and 1.4 hold. We take take $S = \max \left\{ \lceil \sigma^2 / \varepsilon \rceil, n \right\}$, and $\gamma = \min \left\{ \tfrac{1}{L}, \tfrac{\varepsilon S}{2L\sigma^2} \right\} = \Theta(1/L)$ in Method 4, then after $K \geq 24\Delta L/\varepsilon$ iterations the method guarantees that $\tfrac{1}{K} \sum_{k=0}^{K-1} \mathbb{E} \left[ \left\| \nabla f(x^k) \right\|^2 \right] \leq \varepsilon$.*

This is a new theorem analyzing Malenia SGD with the universal computation model:

**Theorem 6.4.** *Consider the assumptions and the parameters from Theorem 6.3, plus Assumption 3.1. Then Method 4 (Malenia SGD) converges after at most $\bar{t}_{\lceil \frac{24L\Delta}{\varepsilon} \rceil}$ seconds, where*

$$\bar{t}_k := \min \left\{ t \geq 0 \,:\, \left( \tfrac{1}{n} \sum_{i=1}^{n} \left\lfloor V_i(t) - V_i(\bar{t}_{k-1}) \right\rfloor^{-1} \right)^{-1} \geq \max \left\{ \tfrac{2\sigma^2}{n\varepsilon}, 1 \right\} \right\} \quad (\bar{t}_0 \equiv 0) \quad (19)$$

*for all $k \geq 1$.*

Up to constant factors, Theorem 6.2 and Theorem 6.4 provide the optimal time complexity in the heterogeneous setting. The result is implicit, which is not a problem in practice since Malenia SGD does not require $\{V_i\}$ to reach the optimality. Let us consider examples where we can get an explicit formula.

**Example 6.5.** [Fixed Computation Model in the Heterogeneous Setting] Assume that $v_i(t) = v_i$ with $v_i > 0$ for all $i \in [n]$. Then

$$\bar{t}_{\lceil \frac{24L\Delta}{\varepsilon} \rceil} = \Theta \left( \max_{i \in [n]} \tfrac{1}{v_i} + \left( \tfrac{1}{n} \sum_{i=1}^{n} \tfrac{1}{v_i} \right) \tfrac{\sigma^2}{n\varepsilon} \right). \quad (20)$$

**Example 6.6.** [Nonlinear Trend in the Heterogeneous Setting] Assume that $v_i(t) = v_i \times g(t)$ with $v_i > 0$ for all $i \in [n]$ and a continuous almost everywhere positive function $g(t) \,:\, \mathbb{R}_+^\infty \to \mathbb{R}_+$. Then

$$\bar{t}_{\lceil \frac{24L\Delta}{\varepsilon} \rceil} = G^{-1} \left( c_1 \cdot \left[ \max_{i \in [n]} \tfrac{1}{v_i} + \left( \tfrac{1}{n} \sum_{i=1}^{n} \tfrac{1}{v_i} \right) \tfrac{\sigma^2}{n\varepsilon} \right] \right), \quad (21)$$

where $G(t) := \int_0^t g(\tau) d\tau$, and $c_1 \in [1/4, 4]$ (can depend on other parameters but is bounded).

Example 6.5 shows that our result recovers the optimal complexity derived in (Tyurin & Richtárik, 2023). However, our time complexity works with virtually any computation model.

## 7 Conclusion

To the best of our knowledge, this is the first work that provides optimal time complexities under virtually arbitrary computation behavior of workers in the distributed setting. We believe that our lower bounds, Theorems 5.1 and 6.2, and upper bounds, Theorems 5.3 and 6.4, close an important problem in parallel optimization. Our approach and techniques have the potential to serve as a foundation for solving other mathematical questions from parallel and asynchronous optimization in the future. One interesting question is determining the optimal time complexities when the universal computation model is correlated with the randomness from stochastic gradients.

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

## CONTENTS

# A PROOF TECHNIQUES

## A.1 PROOF TECHNIQUES IN THE HOMOGENEOUS SETUP

The proof of the upper bound (Theorem 5.3) is relatively simple and uses only (5). The proof of the lower bound (Theorem 5.1) is standard at the beginning: we assume that the workers store the "worst-case" function from (Carmon et al., 2020) and have access to oracles that calculate the exact gradient but zero out the last non-zero coordinate with some probability (Arjevani et al., 2022). The next steps are new and can be briefly described in the following way. The workers calculate in parallel; thus, they can calculate at most $\sum_{i=1}^{n} \lfloor V_i(t) \rfloor$ stochastic gradients by a time $t$. At the same time, the oracles zero out the last coordinate with a probability $p$ using i.i.d. Bernoulli random variables. Therefore, the workers cannot get a point with a non-zero first coordinate earlier than $t_1 := \min \{t \geq 0 : \sum_{i=1}^{n} \lfloor V_i(t) \rfloor \geq \eta_1\}$ seconds, where $\{\eta_k\}_{k=1}^{T}$ are i.i.d. geometric random variables. Using the same reasoning, the workers cannot get a point with a non-zero $k^{\text{th}}$ coordinate earlier than $t_k := \min \{t \geq 0 : \sum_{i=1}^{n} \lfloor V_i(t) - V_i(t_{k-1}) \rfloor \geq \eta_k\}$ seconds, and $T \approx {}^{L\Delta}\!/\varepsilon$. With high probability, the large chunk (at least a quarter of $T \approx {}^{L\Delta}\!/\varepsilon$) of $\{\eta_k\}$ is not significantly smaller than ${}^{1}\!/p \approx \max \{\lceil {}^{\sigma^2}\!/\varepsilon \rceil, 1\}$. Finally, with high probability, at least a quarter of indices from the set $[T]$ satisfy

$$t_k \geq \min \left\{ t \geq 0 : \sum_{i=1}^{n} \lfloor V_i(t) - V_i(t_{k-1}) \rfloor \gtrsim \max \left\{ \lceil {}^{\sigma^2}\!/\varepsilon \rceil, 1 \right\} \right\}. \tag{22}$$

Note that all lower bounds in stochastic optimization ultimately reduce to the concentration analysis of the sum of random variables. Tyurin & Richtárik (2023) approach this by analyzing the sum $\sum_{i=1}^{T} \min_{j \in [n]} \tau_j \eta_{ij}$, where $\eta_j$ are i.i.d. geometric random variables. In our case, we cannot directly apply this reduction anymore because the computation powers vary over time. Therefore, we found a non-trivial modification: we have to reduce the problem to the concentration analysis of the sum of indicators: $\sum_{j=1}^{T} \mathbb{I}[\eta_j > \frac{1}{p}]$ and investigate this sum, which represents the number of indices that satisfy (22).

## A.2 PROOF TECHNIQUES IN THE HETEROGENEOUS SETUP

The proofs in the heterogeneous setup are more technical, so we suggest first understanding the idea in the homogeneous setting. Unlike the homogeneous setting, where all workers have access to stochastic gradients of the same function, the heterogeneous setting offers more freedom in designing the worst-case construction. We can allocate the worst-case functions from (Carmon et al., 2020) in almost any desired manner. We consider $S$ functions $\{h_j\}$ such that $h_j(x) : \mathbb{R}^{S \times T} \to \mathbb{R}$ and $h_j$ depends only on the $j^{\text{th}}$ block $x_j \in \mathbb{R}^T$ of $x = [x_1, \ldots, x_S] \in \mathbb{R}^{S \times T}$, where $T \approx L\Delta/\varepsilon$, and consider the optimization problem with $f(x) = {}^{1}\!/n \sum_{j=1}^{S} h_j(x)$. As usual, the designed oracles zero out the last non-zero coordinate of calculated gradients, and an algorithm cannot find an $\varepsilon$-solution before at least roughly half of the functions $\{h_j\}$ are "solved," requiring non-zero values in the last coordinates of the corresponding blocks. The main question is how to distribute the functions $\{h_j\}$ among the workers. Intuitively, the slower a worker, the more functions we want to assign to this worker. This intuition works, and in Theorem 6.1, we take the first $K$ "parts" of all the functions $\{h_j\}$, and assign the "parts" of the first $c_1/V_1(\bar{t}_1)$ functions to the first worker, the "parts" of the second $c_1/V_2(\bar{t}_1)$ functions to the second worker, and so forth, where $\bar{t}_1 \approx \underline{t}_1$ from (18), $c_1$ is a constant such that $\sum_{i=1}^{n} c_1/V_i(\bar{t}_1) = S$. The next $K$ "parts" we assign proportionally to $c_2/(V_i(\bar{t}_2) - V_i(\bar{t}_1))$, and so forth. In total, the allocation of the functions $\{h_j\}$ is dynamic since one function can be stored on many workers. For all $j \in [S]$, the "parts" of $h_j$ can be distributed among different workers according to $\{V_i\}$. By taking the appropriate values $S$, $K$, and other parameters, we can ensure the lower bound in Theorem 6.1 holds.

However, Theorem 6.1 is only tight up to logarithmic factors because the allocation of $\{h_j\}$ is *predefined*. In response, we propose a construction where the functions $\{h_j\}$ are allocated based on the randomness we receive from the oracles that calculate stochastic gradients in the proof of Theorem 6.2. The idea is to track the Bernoulli random variables, which zero out the last coordinates, and use them in the construction. We ensure that the workers still receive unbiased and $\sigma$–variance bounded stochastic gradients.

## B  PROOF OF THEOREM 5.3

**Theorem 5.3.** *Consider the assumptions and the parameters from Theorem 5.2, plus Assumption 3.1. Then Method 3 (*Rennala SGD*) converges after at most $\bar{t}_{\lceil \frac{24L\Delta}{\varepsilon} \rceil}$ seconds, where*

$$\bar{t}_k := \min\left\{ t \geq 0 \; : \; \sum_{i=1}^{n} \lfloor V_i(t) - V_i(\bar{t}_{k-1}) \rfloor \geq \max\left\{ \left\lceil \frac{\sigma^2}{\varepsilon} \right\rceil, 1 \right\} \right\} \qquad (\bar{t}_0 \equiv 0) \quad \forall k \geq 1. \quad (12)$$

*Proof.* From Theorem 5.2, we know that Method 3 converges after

$$\left\lceil \frac{24L\Delta}{\varepsilon} \right\rceil$$

iterations. The algorithm waits for $S = \max\{\lceil \sigma^2/\varepsilon \rceil, 1\}$ stochastic gradients from all the workers in every iteration. The workers work in parallel and after $t$ seconds they guarantee to calculate

$$\sum_{i=1}^{n} \lfloor V_i(t) \rfloor$$

stochastic gradients (see the discussion of the universal computation model in Section 3). It means they will calculate the first $S = \max\{\lceil \sigma^2/\varepsilon \rceil, 1\}$ stochastic gradients after at most

$$\bar{t}_1 := \min\left\{ t \geq 0 \; : \; \sum_{i=1}^{n} \lfloor V_i(t) \rfloor \geq \max\left\{ \left\lceil \frac{\sigma^2}{\varepsilon} \right\rceil, 1 \right\} \right\},$$

seconds, where the $\min$ operation is well-defined due to Lemma G.1. After at most $\bar{t}_1$ seconds, the algorithm stops the calculations in the workers and asks them to start the calculation of a new batch of $S$ stochastic gradients that will take at most

$$\bar{t}_2 := \min\left\{ t \geq 0 \; : \; \sum_{i=1}^{n} \lfloor V_i(t) - V_i(\bar{t}_1) \rfloor \geq \max\left\{ \left\lceil \frac{\sigma^2}{\varepsilon} \right\rceil, 1 \right\} \right\},$$

seconds because $\lfloor V_i(t) - V_i(\bar{t}_1) \rfloor$ is the number of stochastic gradients that worker $i$ can calculate from time $\bar{t}_1$ to time $t$ (see (5)). Using the same reasoning, it will take at most $\bar{t}_{\lceil \frac{24L\Delta}{\varepsilon} \rceil}$ seconds to finish all calculations. □

## C  PROOF OF THEOREM 6.4

**Theorem 6.4.** *Consider the assumptions and the parameters from Theorem 6.3, plus Assumption 3.1. Then Method 4 (*Malenia SGD*) converges after at most $\bar{t}_{\lceil \frac{24L\Delta}{\varepsilon} \rceil}$ seconds, where*

$$\bar{t}_k := \min\left\{ t \geq 0 \; : \; \left( \frac{1}{n} \sum_{i=1}^{n} \lfloor V_i(t) - V_i(\bar{t}_{k-1}) \rfloor^{-1} \right)^{-1} \geq \max\left\{ \frac{2\sigma^2}{n\varepsilon}, 1 \right\} \right\} \qquad (\bar{t}_0 \equiv 0) \quad (19)$$

*for all $k \geq 1$.*

*Proof.* From Theorem 6.3, we know that Method 4 converges after

$$\left\lceil \frac{24L\Delta}{\varepsilon} \right\rceil$$

iterations. In the first iteration, the algorithm waits for the moment when

$$\frac{1}{n} \sum_{i=1}^{n} \frac{1}{B_i} \leq \frac{n}{S}.$$

Since

$$S = \max\left\{ \left\lceil \frac{\sigma^2}{\varepsilon} \right\rceil, n \right\} \leq \max\left\{ \frac{2\sigma^2}{\varepsilon}, n \right\},$$

we get

$$\frac{n}{S} \geq \min\left\{\frac{n\varepsilon}{2\sigma^2}, 1\right\}.$$

According to the computation model, after $t$ seconds, worker $i$ can calculate

$$B_i = \lfloor V_i(t) \rfloor$$

stochastic gradients meaning that

$$\frac{1}{n}\sum_{i=1}^{n}\frac{1}{B_i} = \frac{1}{n}\sum_{i=1}^{n}\frac{1}{\lfloor V_i(t) \rfloor}.$$

Therefore, the algorithm exits the first iteration after at most

$$\bar{t}_1 := \min\left\{t \geq 0 \;:\; \frac{1}{n}\sum_{i=1}^{n}\frac{1}{\lfloor V_i(t)\rfloor} \leq \min\left\{\frac{n\varepsilon}{2\sigma^2}, 1\right\}\right\}$$

$$= \min\left\{t \geq 0 \;:\; \left(\frac{1}{n}\sum_{i=1}^{n}\frac{1}{\lfloor V_i(t)\rfloor}\right)^{-1} \geq \max\left\{\frac{2\sigma^2}{n\varepsilon}, 1\right\}\right\}$$

seconds, where we use Lemma G.1.

The second iteration will start at least after $\bar{t}_1$ seconds. Since, after $t$ seconds, worker $i$ can calculate at least

$$\lfloor V_i(t) - V_i(\bar{t}_1) \rfloor$$

stochastic gradients in the second iteration. Using the same reasoning as in the first iteration, the algorithm exits the second iteration after at most

$$\bar{t}_2 := \min\left\{t \geq 0 \;:\; \left(\frac{1}{n}\sum_{i=1}^{n}\frac{1}{\lfloor V_i(t) - V_i(\bar{t}_1)\rfloor}\right)^{-1} \geq \max\left\{\frac{2\sigma^2}{n\varepsilon}, 1\right\}\right\}$$

seconds. We can continue and show that it will take at most $\bar{t}_{\lceil \frac{24L\Delta}{\varepsilon}\rceil}$ seconds to finish all calculations.
$\square$

## D   PROOF OF THEOREM 5.1

In our lower bound proofs, we employ the following well-known function. Let us define

$$\mathrm{prog}(x) := \max\{i \geq 0 \,|\, x_i \neq 0\} \quad (x_0 \equiv 1).$$

For any $T \in \mathbb{N}$, Carmon et al. (2020); Arjevani et al. (2022) define a function $F_T \;:\; \mathbb{R}^T \to \mathbb{R}$ such that

$$F_T(x) = -\Psi(1)\Phi(x_1) + \sum_{i=2}^{T}\left[\Psi(-x_{i-1})\Phi(-x_i) - \Psi(x_{i-1})\Phi(x_i)\right], \tag{23}$$

where $x_i$ is the $i^{\text{th}}$ coordinate of a vector $x \in \mathbb{R}^d$ and

$$\Psi(x) = \begin{cases} 0, & x \leq 1/2, \\ \exp\left(1 - \frac{1}{(2x-1)^2}\right), & x \geq 1/2, \end{cases} \quad \text{and} \quad \Phi(x) = \sqrt{e}\int_{-\infty}^{x}e^{-\frac{1}{2}t^2}\,dt.$$

In the proofs, we will only use the results from the following lemma.

**Lemma D.1** (Carmon et al. (2020); Arjevani et al. (2022)). *The function $F_T$ satisfies:*

   *1. $F_T(0) - \inf_{x \in \mathbb{R}^T} F_T(x) \leq \Delta^0 T$, where $\Delta^0 = 12$.*

2. *The function $F_T$ is $l_1$-smooth, where $l_1 = 152$.*

3. *For all $x \in \mathbb{R}^T$, $\|\nabla F_T(x)\|_\infty \leq \gamma_\infty$, where $\gamma_\infty = 23$.*

4. *For all $x \in \mathbb{R}^T$, $\mathrm{prog}(\nabla F_T(x)) \leq \mathrm{prog}(x) + 1$.*

5. *For all $x \in \mathbb{R}^T$, if $\mathrm{prog}(x) < T$, then $\|\nabla F_T(x)\| > 1$.*

We are ready to prove our first main result.

**Theorem 5.1.** *Consider Protocol 2. We take computation powers $\{v_i\}$ such that Assumption 3.1 holds, and fix $L, \Delta, \varepsilon > 0$ and $\sigma^2 \geq 0$ that satisfy the inequality $\varepsilon < c'L\Delta$. For any algorithm $A \in \mathcal{A}_{zr}$, there exist a function $f$, which satisfy Assumptions 1.1, 1.2 and $f(0) - f^* \leq \Delta$, and stochastic gradient mappings $\{g_i\}$ in (9), which satisfy Assumption 1.3, i.e., $\mathbb{E}_\xi [g_i(s_x; \xi, t)] = \nabla f(s_x)$ and $\mathbb{E}_\xi[\|g_i(s_x; \xi, t) - \nabla f(s_x)\|^2] \leq \sigma^2$ for all $s_x \in \mathbb{R}^d, i \in [n]$ such that $\mathbb{E}\left[\inf_{k \in S_t} \|\nabla f(x^k)\|^2\right] > \varepsilon$ holds, where $S_t := \{k \in \mathbb{N}_0 \,|\, t^k \leq t\}$,*

$$t = \tfrac{1}{2} \times \underline{t}_{\left\lfloor c_1 \times \frac{L\Delta}{\varepsilon} \right\rfloor},$$

*and[7]*

$$\underline{t}_k := \min\left\{ t \geq 0 \,:\, \sum_{i=1}^n \left\lfloor V_i(t) - V_i(\underline{t}_{k-1}) \right\rfloor \geq c_2 \times \max\left\{ \left\lceil \frac{\sigma^2}{\varepsilon} \right\rceil, 1 \right\} \right\} \qquad (\underline{t}_0 = 0) \qquad (11)$$

*for all $k \geq 0$. The quantities $c', c_1,$ and $c_2$ are universal constants. The sequences $x^k$ and $t^k$ are defined in Protocol 2.*

*Proof.*
**(Part 1).** In the first part of the proof we use the same idea as in (Carmon et al., 2020; Arjevani et al., 2022; Tyurin & Richtárik, 2023; Huang et al., 2022; Lu & De Sa, 2021). We will construct a "worst-case" function. Let us take any $\lambda > 0, T \in \mathbb{N}$ and take the function

$$f(x) := \frac{L\lambda^2}{l_1} F_T\left(\frac{x}{\lambda}\right).$$

We have to show that $f$ satisfy Assumptions 1.1, 1.2 and $f(0) - f^* \leq \Delta$. Indeed,

$$\|\nabla f(x) - \nabla f(y)\| = \frac{L\lambda}{l_1} \left\| \nabla F_T\left(\frac{x}{\lambda}\right) - \nabla F_T\left(\frac{y}{\lambda}\right) \right\| \leq L\lambda \left\| \frac{x}{\lambda} - \frac{y}{\lambda} \right\| = L\|x - y\| \quad \forall x, y \in \mathbb{R}^d,$$

where $l_1$-smoothness of $F_T$ (Lemma D.1). Let us take

$$T = \left\lfloor \frac{\Delta l_1}{L\lambda^2 \Delta^0} \right\rfloor,$$

then

$$f(0) - \inf_{x \in \mathbb{R}^T} f(x) = \frac{L\lambda^2}{l_1} \left( F_T(0) - \inf_{x \in \mathbb{R}^T} F_T(x) \right) \leq \frac{L\lambda^2 \Delta^0 T}{l_1} \leq \Delta.$$

Next, we construct a stochastic gradient mapping that satisfy Assumption 1.3. As in (Arjevani et al., 2022), for all $i \in [n]$, let us take

$$[g_i(x; \xi, t)]_j := [\nabla f(x)]_j \left(1 + \mathbb{1}\left[j > \mathrm{prog}(x)\right]\left(\frac{\xi_{j,m}}{p} - 1\right)\right) \quad \forall x \in \mathbb{R}^T, \qquad (24)$$

and $\{\xi_j\}$ are i.i.d. from Bernouilli$(p)$ for all $i \in [n]$, where $p \in (0, 1]$. We denote $[v]_j$ as the $j^{\text{th}}$ index of a vector $v \in \mathbb{R}^T$. This mapping satisfy Assumption 1.3 since

$$\mathbb{E}\left[[g_i(x; \xi, t)]_j\right] = [\nabla f(x)]_j \left(1 + \mathbb{1}\left[i > \mathrm{prog}(x)\right]\left(\frac{\mathbb{E}[\xi_j]}{p} - 1\right)\right) = [\nabla f(x)]_j$$

---

[7]We use the standard convention $\min\{\emptyset\} = \infty$.

for all $j \in [T]$, and

$$\mathbb{E}\left[\|g_i(x;\xi,t) - \nabla f(x)\|^2\right] \leq \max_{j\in[T]} |[\nabla f(x)]_j|^2 \mathbb{E}\left[\left(\frac{\xi_j}{p} - 1\right)^2\right]$$

because the difference is non-zero only in one coordinate. Therefore

$$\mathbb{E}\left[\|g_i(x;\xi,t) - \nabla f(x)\|^2\right] \leq \frac{\|\nabla f(x)\|_\infty^2 (1-p)}{p} = \frac{L^2\lambda^2 \left\|\nabla F_T\left(\frac{x}{\lambda}\right)\right\|_\infty^2 (1-p)}{l_1^2 p}$$

$$\leq \frac{L^2\lambda^2\gamma_\infty^2(1-p)}{l_1^2 p} \leq \sigma^2,$$

where we take into account Lemma D.1 and choose

$$p = \min\left\{\frac{L^2\lambda^2\gamma_\infty^2}{\sigma^2 l_1^2}, 1\right\}.$$

We also choose

$$\lambda = \frac{\sqrt{2\varepsilon}l_1}{L}$$

to ensure that

$$\|\nabla f(x)\|^2 = \frac{L^2\lambda^2}{l_1^2}\left\|\nabla F_T\left(\frac{x}{\lambda}\right)\right\|^2 = 2\varepsilon\left\|\nabla F_T\left(\frac{x}{\lambda}\right)\right\|^2$$

for all $x \in \mathbb{R}^T$. Using Lemma D.1, if $\text{prog}(x) < T$, then $\|\nabla F_T(x)\| > 1$. Thus

$$\|\nabla f(x)\|^2 > 2\varepsilon \mathbb{1}\left[\text{prog}(x) < T\right] \tag{25}$$

Using the choice of $\lambda$, one can easily show that

$$T = \left\lfloor \frac{\Delta L}{2\varepsilon l_1\Delta^0} \right\rfloor \tag{26}$$

and

$$p = \min\left\{\frac{2\varepsilon\gamma_\infty^2}{\sigma^2}, 1\right\}. \tag{27}$$

The inequality (25) implies

$$\inf_{k\in S_t} \|\nabla f(x^k)\|^2 > 2\varepsilon \inf_{k\in S_t} \mathbb{1}\left[\text{prog}(x^k) < T\right], \tag{28}$$

where $\{x^k\}_{k=0}^\infty$ are defined in Protocol 2.

**(Part 2).** The last inequality in (28) says that if an algorithm wants to find an $\varepsilon$–stationary point of the function $f$, then it is necessary to return a point $x^k$ such that the last coordinate of $x^k$ is not zero. All algorithms start with the point $x^0 = 0$, and the only way to discover a new non-zero coordinate is through the oracles (9) since the family of algorithms $\mathcal{A}_{\text{zr}}$ is zero-respecting. The function $f$ is a zero-chain (Arjevani et al., 2022) meaning that $\text{prog}(\nabla F_T(x)) \leq \text{prog}(x) + 1$ for all $x \in \mathbb{R}^T$ (Lemma D.1). Therefore, the oracles can reveal the next non-zero coordinate with the probability $p$ due the construction (24).

For all $i \in [n]$, oracle $O_i$ emulates the behavior of a real computation process that can calculate at most

$$\lfloor V_i(t_1) - V_i(t_0) \rfloor$$

stochastic gradients in the time interval $[t_0, t_1]$. Effectively, the condition $V_i(t) - V_i(s_t) \geq 1$ ensures that sufficient time passes before worker $i$ receives a new stochastic gradient.

All workers work in parallel and ask the oracles to return new stochastic gradients. Thus, for all $t \geq 0$, in the interval $[0, t]$, all workers can can calculate at most

$$\sum_{i=1}^n \lfloor V_i(t) \rfloor$$

stochastic gradients. At the same time, the stochastic mapping (24) is constructed so that the last potentially non-zero coordinate is zeroed out using i.i.d. Bernoulli random variables with the parameter $p$. All workers have to wait for the moment when one of the oracles samples a Bernoulli random variables equals to 1. Therefore, the workers have to calculate at least $\eta_1$ stochastic gradients, where $\eta_1$ is a *geometric random variable* with the parameter $p$. Finally, we can conclude that the workers can progress to the first non-zero coordinate after at least

$$t_1 := \inf \left\{ t \geq 0 \ : \ \sum_{i=1}^{n} \lfloor V_i(t) \rfloor \geq \eta_1 \right\} = \min \left\{ t \geq 0 \ : \ \sum_{i=1}^{n} \lfloor V_i(t) \rfloor \geq \eta_1 \right\}$$

seconds, where we use Lemma G.1. In order to get a second non-zero coordinate, the workers should continue calculating stochastic gradients at points with the progress equals to one. Using the same reasoning, the workers can progress to the second non-zero coordinate after at least

$$t_2 := \min \left\{ t \geq 0 \ : \ \sum_{i=1}^{n} \lfloor V_i(t) - V_i(t_1) \rfloor \geq \eta_2 \right\}$$

seconds, where $\eta_2$ is a *geometric random variable* with the parameter $p$. Because worker $i$ first gets a point with the progress equals to one, which takes at least $t_1$ seconds, and then can calculate at most

$$\lfloor V_i(t) - V_i(t_1) \rfloor$$

stochastic gradients by a time $t \geq 0$. We continue: let us define

$$t_i := \min \left\{ t \geq 0 \ : \ \sum_{i=1}^{n} \lfloor V_i(t) - V_i(t_{i-1}) \rfloor \geq \eta_i \right\} \qquad (t_0 \equiv 0),$$

and take i.i.d. $\{\eta_i\}_{i=1}^{T}$ geometric random variables with the probability $p$. We can conclude that all algorithms from $\mathcal{A}_{zr}$ require at least $t_T$ seconds to get a point where the $T^{\text{th}}$ coordinate is non-zero.
**(Part 3).**
It is left to find a concentration bound for $t_T$. Using Lemma G.2 with $p_{i,\eta_1,\ldots,\eta_{i-1}} = p$ for all $i \in [T]$, we have

$$\mathbb{P} \left( \sum_{i=1}^{T} \mathbb{1} \left[ \eta_i > \frac{1}{4p} \right] \leq \frac{T}{2} + \log \delta \right) \leq \delta$$

for all $\delta \in (0, 1]$ (Note: Clearly, Lemma G.2 is too redundant for the current case when $\{\eta_i\}_{i=1}^{T}$ are i.i.d., but we will use the lemma in other proof where the generality is justified).

Since

$$\frac{T}{2} + \log \frac{1}{2} \geq \left\lfloor \frac{T-1}{2} \right\rfloor.$$

With a probability at least $1/2$, there exist $\left\lfloor \frac{T-1}{2} \right\rfloor$ indices such that $\eta_i > \frac{1}{4p}$, i.e.,

$$\left| \left\{ i \in [T] \ \middle| \ \eta_i > \frac{1}{4p} \right\} \right| \geq \left\lfloor \frac{T-1}{2} \right\rfloor.$$

With a probability at least $1/2$, there exist $1 \leq j_1 < j_2 < \cdots < j_{\lfloor \frac{T-1}{2} \rfloor} \leq T$ such that $\eta_{j_k} > \frac{1}{4p}$ for all $k \in \left[ \left\lfloor \frac{T-1}{2} \right\rfloor \right]$. Using a proof by induction, let us show that

$$t_{j_k} \geq \underline{t}_k := \min \left\{ t \geq 0 \ : \ \sum_{i=1}^{n} \lfloor V_i(t) - V_i(\underline{t}_{k-1}) \rfloor \geq \frac{1}{4p} \right\} \qquad (\underline{t}_0 = 0) \tag{29}$$

for all $k \in \left[ \left\lfloor \frac{T-1}{2} \right\rfloor \right]$.

Recall the definition of $t_{j_1}$. Using it, we have

$$t_{j_1} = \min \left\{ t \geq 0 \ : \ \sum_{i=1}^{n} \lfloor V_i(t) - V_i(t_{j_1-1}) \rfloor \geq \eta_{j_1} \right\}.$$

Using $V_i(t_{j_1 - 1}) \geq 0$ and $\eta_{j_1} \geq \frac{1}{4p}$, we get

$$\sum_{i=1}^{n} \lfloor V_i(t) \rfloor \geq \sum_{i=1}^{n} \lfloor V_i(t) - V_i(t_{j_1 - 1}) \rfloor \geq \eta_{j_1} \geq \frac{1}{4p}$$

and

$$t_{j_1} \geq \underline{t}_1 = \min \left\{ t \geq 0 : \sum_{i=1}^{n} \lfloor V_i(t) \rfloor \geq \frac{1}{4p} \right\}.$$

Thus, we have proved the base case. Assume that (29) holds for $k - 1$. Note that

$$t_{j_k} = \min \left\{ t \geq 0 : \sum_{i=1}^{n} \lfloor V_i(t) - V_i(t_{j_k - 1}) \rfloor \geq \eta_{j_k} \right\}.$$

Using $\eta_{j_k} \geq \frac{1}{4p}$, $t_{j_k - 1} \geq t_{j_{(k-1)}}$, the induction assumption $t_{j_{(k-1)}} \geq \underline{t}_{k-1}$, and the fact that the functions $\{V_i\}$ are non-decreasing, we get

$$V_i(t_{j_k - 1}) \geq V_i(t_{j_{(k-1)}}) \geq V_i(\underline{t}_{k-1})$$

and

$$\sum_{i=1}^{n} \lfloor V_i(t) - V_i(\underline{t}_{k-1}) \rfloor \geq \sum_{i=1}^{n} \lfloor V_i(t) - V_i(t_{j_k - 1}) \rfloor \geq \eta_{j_k} \geq \frac{1}{4p}.$$

Therefore

$$t_{j_k} \geq \underline{t}_k = \min \left\{ t \geq 0 : \sum_{i=1}^{n} \lfloor V_i(t) - V_i(\underline{t}_{k-1}) \rfloor \geq \frac{1}{4p} \right\}.$$

In total, with a probability at least $1/2$, due to (26) and (27), we have

$$t_T \geq t_{j_{\lfloor \frac{T-1}{2} \rfloor}} \geq \underline{t}_{\lfloor \frac{T-1}{2} \rfloor} \geq \underline{t}_{\lfloor c_1 \times \frac{L\Delta}{\varepsilon} \rfloor},$$

where

$$\underline{t}_k := \min \left\{ t \geq 0 : \sum_{i=1}^{n} \lfloor V_i(t) - V_i(\underline{t}_{k-1}) \rfloor \geq c_2 \times \max \left\{ \left\lceil \frac{\sigma^2}{\varepsilon} \right\rceil, 1 \right\} \right\} \qquad (\underline{t}_0 = 0) \qquad (30)$$

and $c_1, c_2$ are universal constants. Recall that $t_T$ is a necessary number of seconds to get a point where the $T^{\text{th}}$ coordinate is non-zero. Due to (28),

$$\mathbb{E} \left[ \inf_{k \in S_t} \left\| \nabla f(x^k) \right\|^2 \right] > 2\varepsilon \mathbb{P}(t_T > t) \geq 2\varepsilon \mathbb{P} \left( t_T \geq \underline{t}_{\lfloor c_1 \times \frac{L\Delta}{\varepsilon} \rfloor} \right) \geq \varepsilon$$

for

$$t = \frac{1}{2} \times \underline{t}_{\lfloor c_1 \times \frac{L\Delta}{\varepsilon} \rfloor}.$$

The first inequality follows from the fact that if $t_T > t$, then the set $S_t := \{ k \in \mathbb{N}_0 \,|\, t^k \leq t \}$ contains the indices of iterations from Protocol 2 where all returned by the algorithm points have $\text{prog}(\cdot)$ less than $T$. □

# E  PROOF OF THEOREM 6.1

**Theorem 6.1.** *Consider Protocol 2. We take computation powers $\{v_i\}$ such that Assumption 3.1 holds, and fix $L, \Delta, \varepsilon > 0$ and $\sigma^2 \geq 0$ that satisfy the inequality $\varepsilon < c' L\Delta$. For any algorithm $A \in \mathcal{A}_{zr}$, there exist functions $\{f_i\}_{i=1}^{n}$, where the function $f = \frac{1}{n} \sum_{i=1}^{n} f_i$ satisfies Assumptions 1.1, 1.2 and $f(0) - f^* \leq \Delta$, and stochastic gradient mappings $\{g_i\}_{i=1}^{n}$ in (9), which satisfy Assumption 1.4, i.e.,*

$\mathbb{E}_\xi \left[ g_i(s_x; \xi, t) \right] = \nabla f_i(s_x)$ *and* $\mathbb{E}_\xi [\|g_i(s_x; \xi, t) - \nabla f_i(s_x)\|^2] \leq \sigma^2$ *for all* $s_x \in \mathbb{R}^d, i \in [n]$, *and* $t \geq 0$, *such that* $\mathbb{E}\left[ \inf_{k \in S_t} \left\| \nabla f(x^k) \right\|^2 \right] > \varepsilon$ *holds, where* $S_t := \left\{ k \in \mathbb{N}_0 \,|\, t^k \leq t \right\}$,

$$ t = \tfrac{1}{2} \underline{t}_{\left\lfloor \frac{c_1 \times \frac{L\Delta}{\varepsilon}}{\log \frac{L\Delta}{\varepsilon}} \right\rfloor} $$

*and*

$$ \underline{t}_k := \min \left\{ t \geq 0 \,:\, \left( \tfrac{1}{n} \sum_{i=1}^n \left\lfloor c_3 \times \frac{V_i(t) - V_i(\underline{t}_{k-1})}{\log\left(\frac{L\Delta}{\varepsilon}\right)} \right\rfloor^{-1} \right)^{-1} \geq \max \left\{ c_2 \times \frac{\sigma^2}{n\varepsilon}, 1 \right\} \right\} \tag{17} $$

*for all* $k \geq 1$ ($\underline{t}_0 \equiv 0$). *The quantities* $c', c_1, c_2,$ *and* $c_3$ *are universal constants. The sequences* $x^k$ *and* $t^k$ *are defined in Protocol 2.*

*Proof.* As in Theorem 5.1, we base our proof on the function $F_T(x)$ from Section D. Let us fix any algorithm $A \in \mathcal{A}_{zr}$. First, we define $S \in \mathbb{N}$ functions such that, for all $j \in [S]$, we have $h_j(x) : \mathbb{R}^{S \times T} \to \mathbb{R}$ and

$$ h_j(x) = \frac{nL\lambda^2}{l_1} F_T \left( \frac{x_j}{\lambda} \right), \tag{31} $$

where $T$, $S$, and $\lambda$ are defined later. We assume that $x = [x_1, \ldots, x_S] \in \mathbb{R}^{S \times T}$. We define $x_j \in \mathbb{R}^T$ as the $j^{\text{th}}$ block of a vector $x = [x_1, \ldots, x_S] \in \mathbb{R}^{S \times T}$. The function $h_j$ depends only on a subset of variables $x_j$ from $x$. We construct a function $f : \mathbb{R}^{S \times T} \to \mathbb{R}$ such that

$$ f(x) = \frac{1}{n} \sum_{i=1}^S h_i(x). \tag{32} $$

While the final structure of the local stochastic functions $f_i$ is not yet defined, during the proof, we will ensure that (32) holds.

(**Step 1**: $f \in \mathcal{F}_{\Delta, L}$)
First, we show that $f$ satisfies Assumptions 1.1, 1.2 and $f(0) - f^* \leq \Delta$. Let us show that the function $f$ is $L$-smooth. Indeed, we have

$$ \|\nabla f(x) - \nabla f(y)\|^2 = \frac{1}{n^2} \left\| \sum_{i=1}^S (\nabla h_i(x) - \nabla h_i(y)) \right\|^2 = \frac{1}{n^2} \sum_{i=1}^S \|\nabla h_i(x) - \nabla h_i(y)\|^2 $$

$$ = \frac{1}{n^2} \sum_{i=1}^S \left\| \frac{nL\lambda}{l_1} \nabla F_T \left( \frac{x_i}{\lambda} \right) - \frac{nL\lambda}{l_1} \nabla F_T \left( \frac{y_i}{\lambda} \right) \right\|_i^2, $$

where $\|\cdot\|_j$ is the Euclidean norm w.r.t. $j^{\text{th}}$ block. Then,

$$ \|\nabla f(x) - \nabla f(y)\|^2 = \frac{L^2 \lambda^2}{l_1^2} \sum_{i=1}^S \left\| \nabla F_T \left( \frac{x_i}{\lambda} \right) - \nabla F_T \left( \frac{y_i}{\lambda} \right) \right\|_i^2 \leq L^2 \sum_{i=1}^S \|x_i - y_i\|_i^2 = L^2 \|x - y\|^2, $$

where the last inequality due to Lemma D.1. Let us take

$$ T = \left\lfloor \frac{\Delta l_1}{L\lambda^2 S \Delta^0} \right\rfloor \tag{33} $$

then

$$ f(0) - \inf_{x \in \mathbb{R}^{S \times T}} f(x) = \frac{1}{n} \sum_{i=1}^S \frac{nL\lambda^2}{l_1} \left( F_T(0) - \inf_{x \in \mathbb{R}^T} F_T(x) \right) \leq \frac{L\lambda^2 S \Delta^0 T}{l_1} \leq \Delta, $$

where the first inequality due to Lemma D.1.

(**Step 2**: Construction of local functions $f_i$)
Each function $h_i(x)$ depends only on $T$ coordinates of the $i^{\text{th}}$ block. We split these $T$ coordinates into $\bar{T} + 1$ groups with $\bar{T} := \lfloor T/K \rfloor$, where we choose $K$ later. For all $w \in [\bar{T} + 1]$, let us take any

$$s_{w,0} \equiv 1 \le s_{w,1} \le \cdots \le s_{w,n-1} \le s_{w,n} \equiv S + 1. \tag{34}$$

It is convenient to define the length of each segment: $a_{w,i} := s_{w,i} - s_{w,(i-1)}$ for all $w \in [\bar{T} + 1]$ and $i \in [n]$. Also, we take any times

$$\bar{t}_0 \equiv 0 \le \bar{t}_1 \le \cdots \le \bar{t}_{\bar{T}} \le \bar{t}_{\bar{T}+1} \equiv \infty \tag{35}$$

associated with the coordinates' groups. We construct functions in the following way. For all $i \in [n]$, we define

$$\widehat{f}_i(x) := \sum_{j=s_{1,(i-1)}}^{s_{1,i}-1} \left( -\Psi(1)\Phi(x_{j,1}) + \cdots + \Psi(-x_{j,K-1})\Phi(-x_{j,K}) - \Psi(x_{j,K-1})\Phi(x_{j,K}) \right)$$

$$+ \sum_{j=s_{2,(i-1)}}^{s_{2,i}-1} \left( \Psi(-x_{j,K})\Phi(-x_{j,K+1}) - \Psi(x_{j,K})\Phi(x_{j,K+1}) + \ldots \right.$$

$$\left. + \Psi(-x_{j,2K-1})\Phi(-x_{j,2K}) - \Psi(x_{j,2K-1})\Phi(x_{j,2K}) \right)$$

$$+ \ldots$$

$$+ \sum_{j=s_{\bar{T},(i-1)}}^{s_{\bar{T},i}-1} \left( \Psi(-x_{j,(\bar{T}-1)K})\Phi(-x_{j,(\bar{T}-1)K+1}) - \Psi(x_{j,(\bar{T}-1)K})\Phi(x_{j,(\bar{T}-1)K+1}) + \ldots \right.$$

$$\left. + \Psi(-x_{j,\bar{T}K-1})\Phi(-x_{j,\bar{T}K}) - \Psi(x_{j,\bar{T}K-1})\Phi(x_{j,\bar{T}K}) \right)$$

$$+ \sum_{j=s_{\bar{T}+1,(i-1)}}^{s_{\bar{T}+1,i}-1} \left( \Psi(-x_{j,\bar{T}K})\Phi(-x_{j,\bar{T}K+1}) - \Psi(x_{j,\bar{T}K})\Phi(x_{j,\bar{T}K+1}) + \ldots \right.$$

$$\left. + \Psi(-x_{j,T-1})\Phi(-x_{j,T}) - \Psi(x_{j,T-1})\Phi(x_{j,T}) \right)$$

and

$$f_i(x) := \frac{nL\lambda^2}{l_1} \widehat{f}_i\left(\frac{x}{\lambda}\right),$$

where $x_{j,i}$ is the $i^{\text{th}}$ coordinate of $x_j \in \mathbb{R}^T$.

Let us explain the idea. For all $j \in [S]$, we take the function $h_j$ from (31), which consists of $T$ parts with the structure $\Psi(-x_{j,\cdot})\Phi(-x_{j,\cdot}) - \Psi(x_{j,\cdot})\Phi(x_{j,\cdot})$, and distribute these parts between the workers according to the predefined segments $\{s_{w,i}\}$. The first $K$ parts of the function $h_j$ will be stored in worker $i_1$ such that $s_{1,(i_1-1)} \le j < s_{1,i_1}$, the second $K$ parts will be stored in worker $i_2$ such that $s_{2,(i_2-1)} \le j < s_{2,i_2}$, and so on. One can easily show that $\sum_{j=1}^{S} h_j = \sum_{i=1}^{n} f_i$.
(**Step 3**: Time-dependent stochastic oracles)
We now construct a stochastic oracle. Let us take

$$p_{w,i} := \min\left\{ \frac{a_{w,i}n^2L^2\lambda^2\gamma_\infty^2}{\sigma^2 l_1^2}, 1 \right\} \tag{36}$$

for all $w \in [\bar{T} + 1], i \in [n]$. The stochastic mapping takes a point $x$, a random variable $\xi$, a time $t$, and returns

$$[g_i(x;\xi,t)]_{j,m} := [\nabla f_i(x)]_{j,m} \times$$

$$\times \left( 1 + \mathbb{1}\left[ m > \text{prog}(x_j) \wedge \left\lfloor \frac{m-1}{K} \right\rfloor + 1 = w(t) \right] \left( \frac{\xi_{j,m}}{p_{w(t),i}} - 1 \right) \right) \quad \forall x \in \mathbb{R}^{S \times T}, \tag{37}$$

where $[x]_{j,m}$ is the $m^{\text{th}}$ coordinate of the $j^{\text{th}}$ block of $x \in \mathbb{R}^{S \times T}$, $\{\xi_{j,m}\}_{j,m}$ are i.i.d. from Bernoulli$(p_{w(t),i})$, and $w(t) \in [\bar{T}]$ is the index such that $\bar{t}_{(w(t)-1)} \le t < \bar{t}_{w(t)}$.

The idea is almost the same as in (Arjevani et al., 2022): we also zero out the last potentially non-zero coordinate with the probability $p_{w(t),i}$. However, we only zero out a coordinate if it belongs to the

parts from the set $\{(K-1)w(t)+1, \ldots, Kw(t)\}$, where $w(t)$ is associated with the current time interval $[\bar{t}_{(w(t)-1)}, \bar{t}_{w(t)})$ where the current time $t$ belongs to.

Then, $g_i(x; \xi, t)$ is unbiased because $\mathbb{E}_\xi \left[ \left( \frac{\xi_{j,m}}{p_{w(t),i}} - 1 \right) \right] = 0$ and

$$\mathbb{E}_\xi \left[ \|g_i(x; \xi, t) - \nabla f_i(x)\|^2 \right] \leq \sum_{j=s_{w(t),(i-1)}}^{s_{w(t),i}-1} \frac{n^2 L^2 \lambda^2 \left\| \nabla F_T \left( \frac{x_j}{\lambda} \right) \right\|_\infty^2 (1 - p_{w(t),i})}{l_1^2 p_{w(t),i}}$$

because, due the condition $\left\lfloor \frac{m-1}{K} \right\rfloor + 1 = w(t)$, we only consider the blocks from the sum $\sum_{j=s_{w(t),(i-1)}}^{s_{w(t),i}-1}$. Using Lemma D.1, we have $\|\nabla F_T(x)\|_\infty^2 \leq \gamma_\infty^2$ for all $x \in \mathbb{R}^T$ and

$$\mathbb{E}_\xi \left[ \|g_i(x; \xi, t) - \nabla f_i(x)\|^2 \right] \leq \frac{a_{w(t),i} n^2 L^2 \lambda^2 \gamma_\infty^2 (1 - p_{w(t),i})}{l_1^2 p_{w(t),i}} \leq \sigma^2,$$

where the last inequality follows from the choice of $p_{w,i}$ in (36).

(**Step 4**: Analysis of Protocol)

Using the definition of $f$, we get

$$\|\nabla f(x)\|^2 = \frac{1}{n^2} \sum_{i=1}^{S} \|\nabla h_i(x)\|^2 = \sum_{i=1}^{S} \frac{L^2 \lambda^2}{l_1^2} \left\| \nabla F_T \left( \frac{x_i}{\lambda} \right) \right\|^2$$

$$> \frac{L^2 \lambda^2}{l_1^2} \sum_{i=1}^{S} \mathbb{1}[\text{prog}(x_i) < T] \tag{38}$$

for all $x = [x_1, \ldots, x_n] \in \mathbb{R}^T$. In the last inequality, we use Lemma D.1. Let us take

$$\lambda = \sqrt{\frac{4\varepsilon l_1^2}{L^2 S}}. \tag{39}$$

to ensure that

$$\inf_{k \in S_t} \left\| \nabla f(x^k) \right\|^2 > \inf_{k \in S_t} \frac{4\varepsilon}{S} \sum_{i=1}^{S} \mathbb{1}[\text{prog}(x_i^k) < T] \geq \frac{4\varepsilon}{S} \sum_{i=1}^{S} \inf_{k \in S_t} \mathbb{1}[\text{prog}(x_i^k) < T], \tag{40}$$

where $x^k$ are points defined in Protocol 2. Using Markov's inequality, we get

$$\begin{aligned}
&\mathbb{P} \left( \frac{4\varepsilon}{S} \sum_{i=1}^{S} \inf_{k \in S_t} \mathbb{1}[\text{prog}(x_i^k) < T] \leq 2\varepsilon \right) \\
&= \mathbb{P} \left( \frac{1}{S} \sum_{i=1}^{S} \inf_{k \in S_t} \mathbb{1}[\text{prog}(x_i^k) < T] \leq \frac{1}{2} \right) \\
&= \mathbb{P} \left( \frac{1}{S} \sum_{i=1}^{S} \sup_{k \in S_t} \mathbb{1}[\text{prog}(x_i^k) \geq T] \geq \frac{1}{2} \right) \\
&\leq 2\mathbb{E} \left[ \frac{1}{S} \sum_{i=1}^{S} \sup_{k \in S_t} \mathbb{1}[\text{prog}(x_i^k) \geq T] \right] = \frac{2}{S} \sum_{i=1}^{S} \mathbb{E} \left[ \sup_{k \in S_t} \mathbb{1}[\text{prog}(x_i^k) \geq T] \right].
\end{aligned} \tag{41}$$

(**Step 5**: Bound on the expectations)

In this step of the proof, we fix $j \in [S]$, and consider the function $h_j$ from (31).

Recall that worker $i$ can calculate $\lfloor V_i(t) \rfloor$ stochastic gradients by a time $t$. Therefore, it takes at least

$$\min \{t \geq 0 : \lfloor V_i(t) \rfloor \geq \eta\} \geq V_i^{-1}(\eta)$$

seconds to calculate $\eta \in \mathbb{N}$ stochastic gradients, where we use the definition (8).

By the construction of the functions $\{f_i\}$, the first $K$ parts of $h_j$ belong to worker $i_{1,j}$ such that $s_{1,i_{1,j}-1} \leq j < s_{1,i_{1,j}}$. While the algorithm $A$ is returning times $t^k$ such that $\bar{t}_0 \equiv 0 \leq t^k < \bar{t}_1$, by the construction of the stochastic mapping (37), whenever the algorithm calls oracle $O_{i_{1,j}}$, the oracle zeros out the last potentially non-zero coordinate with the probability $p_{1,i_{1,j}}$. However, due to the condition $\lfloor \frac{m-1}{K} \rfloor + 1 = w(t)$, the stochastic mapping will zero out the first $K$ coordinates only if $t^k < \bar{t}_1$. Thus, the time required to progress to the $K^{\text{th}}$ coordinate in the block $x_j$ is at least

$$\hat{t}_{1,j} := \min\left\{\bar{t}_1, V_{i_{1,j}}^{-1}\left(\sum_{v=1}^{K} \eta_{1,i_{1,j},v}\right)\right\}$$

seconds, where $\{\eta_{1,i_{1,j},v}\}$ are i.i.d. geometric random variables with the probability $p_{1,i_{1,j}}$. Because either the algorithm returns $t^k \geq \bar{t}_1$, or it keeps returning $t^k < \bar{t}_1$, but then $A$ should calculate at least $\sum_{v=1}^{K} \eta_{1,i_{1,j},v}$ stochastic gradients since the stochastic mapping zeros out the last potentially non-zero coordinate, and $A$ should wait $K$ times for the "lucky" ($\xi = 1$) draws of Bernoulli random variables.

Using the same reasoning, for all $w \in [\bar{T}]$, it will take at least

$$\hat{t}_{w,j} := \min\left\{\bar{t}_w, V_{i_{w,j}}^{-1}\left(\sum_{v=1}^{K} \eta_{w,i_{w,j},v} + V_{i_{w,j}}(\hat{t}_{w-1,j})\right)\right\} \qquad (\hat{t}_{0,j} \equiv 0) \qquad (42)$$

seconds to progress to the $w \times K^{\text{th}}$ coordinate in the block $x_j$, where $\{\eta_{w,i_{w,j},v}\}$ are i.i.d. geometric random variables with the probability $p_{w,i_{w,j}}$, and $i_{w,j}$ is the index of the worker such that $s_{w,i_{w,j}-1} \leq j < s_{w,i_{w,j}}$. Because either the algorithm returns $t^k \geq \bar{t}_w$, or it keeps returning $t^k < \bar{t}_w$, but then $A$ should first progress to the $(w-1) \times K^{\text{th}}$ coordinate, which takes at least $\hat{t}_{w-1,j}$ seconds, and then should calculate at least $\sum_{v=1}^{K} \eta_{w,i_{w,j},v}$ stochastic gradients. This will take at least

$$\min\left\{t \geq 0 : \lfloor V_{i_{w,j}}(t) - V_{i_{w,j}}(\hat{t}_{w-1,j}) \rfloor \geq \sum_{v=1}^{K} \eta_{w,i_{w,j},v}\right\}$$

$$\overset{\text{(cont. of } V_{i_{w,j}})}{=} \min\left\{t \geq 0 : V_{i_{w,j}}(t) - V_{i_{w,j}}(\hat{t}_{w-1,j}) = \sum_{v=1}^{K} \eta_{w,i_{w,j},v}\right\}$$

$$\overset{(8)}{=} V_{i_{w,j}}^{-1}\left(\sum_{v=1}^{K} \eta_{w,i_{w,j},v} + V_{i_{w,j}}(\hat{t}_{w-1,j})\right)$$

seconds.

Using Lemma G.3, with a probability at least $1 - \bar{T}e^{-K/2}$, we have

$$\sum_{v=1}^{K} \eta_{w,i_{w,j},v} \geq \frac{K}{8p_{w,i_{w,j}}}$$

for all $w \in [\bar{T}]$. Using these inequalities and (42), we get

$$\hat{t}_{w,j} \geq \min\left\{\bar{t}_w, V_{i_{w,j}}^{-1}\left(\frac{K}{8p_{w,i_{w,j}}} + V_{i_{w,j}}(\hat{t}_{w-1,j})\right)\right\}, \qquad (43)$$

for all $w \in [\bar{T}]$.

Note that $\{\bar{t}_w\}_{w=1}^{\bar{T}}, \{s_{w,i}\}_{w\in[\bar{T}+1],i\in[n]}$, and $S$ are free parameters with the conditions (34) and (35). Due to (36) and (39), we have

$$p_{w,i} = \min\left\{\frac{4\varepsilon\gamma_\infty^2 a_{w,i} n^2}{\sigma^2 S}, 1\right\}.$$

Therefore, we can use Lemma G.4: we can find $\{\bar{t}_w\}_{w=1}^{\bar{T}}, \{s_{w,i}\}_{w\in[\bar{T}+1],i\in[n]}$, and $S$ such that $V_i^{-1}\left(\frac{K}{8p_{1,i}}\right) \geq \bar{t}_1$ for all $i \in [n]$. With a probability at least $1 - \bar{T}e^{-K/2}$, with the chosen parameters, we have

$$\hat{t}_{1,j} \geq \min\left\{\bar{t}_1, V_{i_{1,j}}^{-1}\left(\frac{K}{8p_{1,i_{1,j}}}\right)\right\} \geq \min\{\bar{t}_1, \bar{t}_1\} = \bar{t}_1.$$

Using a proof by induction, with a probability at least $1 - \bar{T}e^{-K/2}$, let us prove that $\hat{t}_{w,j} \geq \bar{t}_w$ for all $w \in [\bar{T}]$ with the parameters from Lemma G.4. The base case has been proved. Assume that

$$\hat{t}_{w-1,j} \geq \bar{t}_{w-1},$$

then, using (43), we get

$$\hat{t}_{w,j} \geq \min\left\{\bar{t}_w, V_{i_{w,j}}^{-1}\left(\frac{K}{8p_{w,i_{w,j}}} + V_{i_{w,j}}(\hat{t}_{w-1,j})\right)\right\} \geq \min\left\{\bar{t}_w, V_{i_{w,j}}^{-1}\left(\frac{K}{8p_{w,i_{w,j}}} + V_{i_{w,j}}(\bar{t}_{w-1})\right)\right\}.$$

In Lemma G.4, we show that $V_i^{-1}\left(\frac{K}{8p_{w,i}} + V_i(\bar{t}_{w-1})\right) \geq \bar{t}_w$ for all $i \in [n]$. Thus

$$\hat{t}_{w,j} \geq \min\{\bar{t}_w, \bar{t}_w\} = \bar{t}_w.$$

(**Step 6**: Choose a parameter $K$)
In the previous step, we prove that, with a probability at least $1 - \bar{T}e^{-K/2}$, the algorithm requires at least $\bar{t}_{\bar{T}}$ seconds to progress to the $K \times \bar{T}^{\text{th}}$ coordinate ($K \times \bar{T} \leq T$), where $\bar{t}_{\bar{T}}$ is defined in in the proof of Lemma G.4.

Let us take

$$K = \lfloor 2\log 4T \rfloor,$$

then, with a probability at least $3/4$, the algorithm will require at least $\bar{t}_{\bar{T}}$ seconds to get a non-zero last coordinate in the block $x_j$. Thus

$$\mathbb{E}\left[\sup_{k \in S_t} \mathbb{1}[\text{prog}(x_j^k) \geq T]\right] \leq \frac{1}{4}$$

for all $j \in [n]$ and for all $t \leq \frac{1}{2}\bar{t}_{\bar{T}}$. We substitute these inequalities to (41) and (40), and get

$$\mathbb{P}\left(\frac{4\varepsilon}{S}\sum_{i=1}^S \inf_{k \in S_t} \mathbb{1}[\text{prog}(x_i^k) < T] \leq 2\varepsilon\right) \leq \frac{1}{2}$$

and

$$\mathbb{E}\left[\inf_{k \in S_t} \left\|\nabla f(x^k)\right\|^2\right] > \mathbb{E}\left[\frac{4\varepsilon}{S}\sum_{i=1}^S \inf_{k \in S_t} \mathbb{1}[\text{prog}(x_i^k) < T]\right] > 2\varepsilon\mathbb{P}\left(\frac{4\varepsilon}{S}\sum_{i=1}^S \inf_{k \in S_t} \mathbb{1}[\text{prog}(x_i^k) < T] > 2\varepsilon\right) \geq \varepsilon$$

for all

$$t \leq \frac{1}{2}\widetilde{t}_{\bar{T}} \overset{(60)}{\leq} \frac{1}{2}\bar{t}_{\bar{T}},$$

where

$$\bar{T} = \left\lfloor \frac{T}{\lfloor 2\log 4T \rfloor} \right\rfloor \overset{(33),(39)}{\geq} \left\lfloor \frac{c_1 \times \frac{L\Delta}{\varepsilon}}{\log \frac{L\Delta}{\varepsilon}} \right\rfloor$$

for some universal constant $c_1$, where we use the assumption $\varepsilon < c'L\Delta$ of the theorem. Finally, since $\widetilde{t}_w \geq \underline{t}_w$ for all $w \in [\bar{T}]$ with the chosen $K$, where the later sequence is defined in (17), we can take

$$t = \frac{1}{2}\underline{t}_{\left\lfloor \frac{c_1 \times \frac{L\Delta}{\varepsilon}}{\log \frac{L\Delta}{\varepsilon}} \right\rfloor}.$$

$\square$

# F    PROOF OF THEOREM 6.2

**Theorem 6.2.** *Consider Protocol 2. We take computation powers $\{v_i\}$ such that Assumption 3.1 holds, and fix $L, \Delta, \varepsilon > 0$ and $\sigma^2 \geq 0$ that satisfy the inequality $\varepsilon < c'L\Delta$. For any algorithm $A \in \mathcal{A}_{\text{zr}}$, **we sample** $\{f_i\}_{i=1}^n$ **from some distribution of functions**, where the function $f = \frac{1}{n}\sum_{i=1}^n f_i$*

*satisfies Assumptions 1.1, 1.2 and $f(0) - f^* \leq \Delta$ deterministically, and there exist stochastic gradient mappings $\{g_i\}_{i=1}^n$ in (9), which satisfy Assumption 1.4, i.e., $\mathbb{E}_\xi [g_i(s_x; \xi, t)] = \nabla f_i(s_x)$ and $\mathbb{E}_\xi [\|g_i(s_x; \xi, t) - \nabla f_i(s_x)\|^2] \leq \sigma^2$ for all $s_x \in \mathbb{R}^d, i \in [n]$, and $t \geq 0$, such that $\mathbb{E}\left[\inf_{k \in S_t} \|\nabla f(x^k)\|^2\right] > \varepsilon$ holds[8], where $S_t := \{k \in \mathbb{N}_0 \,|\, t^k \leq t\}$,*

$$t = \tfrac{1}{2} \underline{t}_{\left\lfloor c_1 \times \frac{L\Delta}{\varepsilon} \right\rfloor}$$

*and*

$$\underline{t}_k := \min \left\{ t \geq 0 \,:\, \left( \tfrac{1}{n} \sum_{i=1}^n \left\lfloor c_3 \times (V_i(t) - V_i(\underline{t}_{k-1})) \right\rfloor^{-1} \right)^{-1} \geq \max \left\{ c_2 \times \tfrac{\sigma^2}{n\varepsilon}, 1 \right\} \right\} \quad (18)$$

*for all $k \geq 1$ ($\underline{t}_0 \equiv 0$). The quantities $c', c_1, c_2,$ and $c_3$ are universal constants. The sequences $x^k$ and $t^k$ are defined in Protocol 2.*

*Proof.* We base our proof on the function $F_T(x)$ from Section D. Let us fix any algorithm $A \in \mathcal{A}_{zr}$. First, we define $S \in \mathbb{N}$ functions such that, for all $j \in [S]$, we have $h_j(x) : \mathbb{R}^{S \times T} \to \mathbb{R}$ and

$$h_j(x) = \frac{nL\lambda^2}{l_1} F_T\left(\frac{x_j}{\lambda}\right), \quad (44)$$

where $T, S,$ and $\lambda$ are defined later. We assume that $x = [x_1, \dots, x_S] \in \mathbb{R}^{S \times T}$. We define $x_j \in \mathbb{R}^T$ as the $j^{\text{th}}$ block of a vector $x = [x_1, \dots, x_S] \in \mathbb{R}^{S \times T}$. The function $h_j$ depends only on a subset of variables $x_j$ from $x$. We construct a function $f : \mathbb{R}^{S \times T} \to \mathbb{R}$ such that

$$f(x) = \frac{1}{n} \sum_{i=1}^S h_i(x). \quad (45)$$

While the final structure of the local stochastic functions $f_i$ is not yet defined, during the proof, we will ensure that (45) holds.

(**Step 1**: $f \in \mathcal{F}_{\Delta,L}$)
We have to show that $f$ satisfies Assumptions 1.1, 1.2 and $f(0) - f^* \leq \Delta$. This step is exactly the same as in the proof of Theorem 6.1. It is sufficient to take

$$T = \left\lfloor \frac{\Delta l_1}{L\lambda^2 S\Delta^0} \right\rfloor. \quad (46)$$

(**Step 2**: Construction of local functions $f_i$ and stochastic mappings)
Unlike the proof of Theorem 6.1 where the functions are predetermined, in this construction the functions $\{f_i\}$ depend on sequences of random variables and are constructed algorithmically in the following way.

For all $w \in [T + 1]$, let us take any

$$s_{w,0} \equiv 1 \leq s_{w,1} \leq \cdots \leq s_{w,n-1} \leq s_{w,n} \equiv S + 1, \quad (47)$$

and define $a_{w,i} := s_{w,i} - s_{w,(i-1)}$. We also take any $p_{w,i} > 0$ for all $w \in [T], i \in [n]$, and any times

$$\bar{t}_0 \equiv 0 \leq \bar{t}_1 \leq \cdots \leq \bar{t}_T \leq \bar{t}_{T+1} \equiv \infty \quad (48)$$

associated with $\{s_{w,\cdot}\}_{w \in [T+1]}$ and $\{p_{w,\cdot}\}_{w \in [T]}$. We construct the functions $\{f_i\}$ using the algorithm below.

---

[8]We take the expectation over all randomness.

---

**Algorithm 5** "Resisting allocation" of the functions $\{h_j\}$

---

1: $f_i(x) \leftarrow 0$ for all $i \in [n]$
2: **for** $j = 1, \ldots, S$ **do**
3:      Current time window $w = 1$
4:      **for** $m = 1, \ldots, T$ **do**
5:          Find $i_{w,j}$ such that $s_{w,i_{w,j}-1} \leq j < s_{w,i_{w,j}}$
6:          Set $b_{j,m} \leftarrow (w, i_{w,j})$
7:          Update $f_{i_{w,j}}(x) \leftarrow f_{i_{w,j}}(x) + \frac{nL\lambda^2}{l_1} \left( \Psi\left(-\frac{x_{j,m-1}}{\lambda}\right) \Phi\left(-\frac{x_{j,m}}{\lambda}\right) - \Psi\left(\frac{x_{j,m-1}}{\lambda}\right) \Phi\left(\frac{x_{j,m}}{\lambda}\right) \right)$
            ($x_{j,0} \equiv 0$ for all $j \in [S]$)
8:          Draw an infinite i.i.d. sequence $\{\xi_{j,m,s}\}_{s=1}^{\infty}$ from Bernoulli$(p_{w,i_{w,j}})$
9:          Find the first moment when $\xi_{j,m,s} = 1$, i.e., $\eta_{j,m} = \inf\{s \geq 1 : \xi_{j,m,s} = 1\}$
10:         **if** $V_{i_{w,j}}^{-1}\left(\eta_{j,m} + V_{i_{w,j}}(\bar{t}_{w-1})\right) \geq \bar{t}_w$ **then**
11:            $w \leftarrow w + 1$
12:         **end if**
13:      **end for**
14: **end for**

---

As in the proof of Theorem 6.1, the stochastic mapping takes a point $x$, a random variable $\xi$, a time $t$, and returns

$$[g_i(x; \bar{\xi}, t)]_{j,m} := \begin{cases} [\nabla f_i(x)]_{j,m} \times \frac{\bar{\xi}_j}{p_{w(t),i}}, & m = \text{prog}(x_j) + 1 \wedge b_{j,m} = (w(t), i), \\ [\nabla f_i(x)]_{j,m}, & \text{otherwise,} \end{cases} \quad (49)$$

where $[x]_{j,m}$ is the $m^{\text{th}}$ coordinate of the $j^{\text{th}}$ block of $x \in \mathbb{R}^{S \times T}$, $\bar{\xi} \equiv (\bar{\xi}_1, \ldots, \bar{\xi}_S)$, $\bar{\xi}_j$ is the "next" random variable from $\{\xi_{j,(\text{prog}(x_j)+1),s}\}_{s=1}^{\infty}$ (see Alg. 5), and $w(t) \in [T]$ is the index such that $\bar{t}_{(w(t)-1)} \leq t < \bar{t}_{w(t)}$.

Let us clarify what we mean by the "next" random variable. In Line 8 of Alg. 5, we draw an infinite i.i.d. sequence $\{\xi_{j,m,s}\}_{s=1}^{\infty}$ from Bernoulli$(p_{w,i_{w,j}})$. For the first time when the mapping $g_i$ has to take the "next" random variable from $\{\xi_{j,(\text{prog}(x_j)+1),s}\}_{s=1}^{\infty}$, it takes $\xi_{j,(\text{prog}(x_j)+1),1}$. For the second time, it takes $\xi_{j,(\text{prog}(x_j)+1),2}$, and so forth.

For all $i \in [n]$, the mapping $g_i$ in the oracle $O_i$ zeroes out the coordinate $m$ of the gradient $\nabla f_i(x)$ corresponding to $m = \text{prog}(x_j) + 1$ (idea is the same as in (Arjevani et al., 2022)). However, we only zero out this coordinate if the corresponding part of the function $h_j$ is stored on worker $i$ at the time $t$ (condition $b_{j,m} = (w(t), i)$).

We now explain the idea. For all $i \in [n]$, deterministically, we have

$$f_i(x) = \frac{nL\lambda^2}{l_1} \sum_{j=s_{1,i-1}+1}^{s_{1,i}} -\Psi(1)\Phi\left(\frac{x_{j,1}}{\lambda}\right) + \ldots.$$

where $x_{j,i}$ is the $i^{\text{th}}$ coordinate of $x_j \in \mathbb{R}^T$. Thus, the first part $-\Psi(1)\Phi(x_1)$ of the functions (44) (see (23)) are allocated according to the values $\{s_{1,i}\}$.

As always, we rely on the fact that the algorithm is zero-respecting, meaning that at the beginning, it starts with the point $x^0 = 0$. While $x^k = 0$, it does not matter where we allocate the other parts of the functions $h_j$. The main idea is to decide the allocation based on the random variables $\{\xi_{j,m,s}\}_{s=1}^{\infty}$ from Alg. 5. By the construction of the functions $\{f_i\}$, the first part of $h_j$ belongs to worker $i_{1,j}$ such that $s_{1,i_{1,j}-1} \leq j < s_{1,i_{1,j}}$. The oracle $O_{i_{1,j}}$ zeroes out the first coordinate of the $j^{\text{th}}$ block of gradients with the probability $p_{1,i_{1,j}}$ (see (49)).

In Alg. 5, we consider two cases:
If $V_{i_{1,j}}^{-1}(\eta_{j,1}) < \bar{t}_1$, then in the next iteration of Alg. 5, we allocate the second part of the function $h_j$ to the same worker $i_{1,j}$, i.e.,

$$f_{i_{1,j}}(x) \leftarrow f_{i_{1,j}}(x) + \frac{nL\lambda^2}{l_1} \left( \Psi\left(-\frac{x_{j,1}}{\lambda}\right) \Phi\left(-\frac{x_{j,2}}{\lambda}\right) - \Psi\left(\frac{x_{j,1}}{\lambda}\right) \Phi\left(\frac{x_{j,2}}{\lambda}\right) \right).$$

Otherwise, if $V_{i_{1,j}}^{-1}(\eta_{j,1}) \geq \bar{t}_1$, then we increment the parameter $w$ in Alg. 5 and allocate the second part to worker $i_{2,j}$ :

$$f_{i_{2,j}}(x) \leftarrow f_{i_{2,j}}(x) + \frac{nL\lambda^2}{l_1}\left(\Psi\left(-\frac{x_{j,1}}{\lambda}\right)\Phi\left(-\frac{x_{j,2}}{\lambda}\right) - \Psi\left(\frac{x_{j,1}}{\lambda}\right)\Phi\left(\frac{x_{j,2}}{\lambda}\right)\right),$$

where $i_{2,j}$ such that $s_{2,i,j-1} \leq j < s_{2,i_{2,j}}$.

The mapping $g_i$ is unbiased and $\sigma^2$–variance bounded. If $m > \text{prog}(x_j) + 1$, then $[\nabla f_i(x)]_{j,m} = 0$ deterministically due to Lemma D.1. If $m = \text{prog}(x_j) + 1$ and $b_{j,m} = (w(t), i)$, then we have to show that

$$\mathbb{E}_{\bar{\xi}}\left[[\nabla f_i(x)]_{j,m} \times \frac{\bar{\xi}_j}{p_{w(t),i}}\right] = [\nabla f_i(x)]_{j,m}$$

for all $j \in [S], m \in [T]$. Notice that due the construction in Alg. 5, the gradient $\nabla f_i(x)$ can depend on $\{\xi_{j,m,s}\}_{s=1}^{\infty}$ since the random variables affect the allocation of parts starting from

$$\Psi\left(-\frac{x_{j,m}}{\lambda}\right)\Phi\left(-\frac{x_{j,m+1}}{\lambda}\right) - \Psi\left(\frac{x_{j,m}}{\lambda}\right)\Phi\left(\frac{x_{j,m+1}}{\lambda}\right),$$
$$\Psi\left(-\frac{x_{j,m+1}}{\lambda}\right)\Phi\left(-\frac{x_{j,m+2}}{\lambda}\right) - \Psi\left(\frac{x_{j,m+1}}{\lambda}\right)\Phi\left(\frac{x_{j,m+2}}{\lambda}\right), \tag{50}$$
$$\cdots$$
$$\Psi\left(-\frac{x_{j,T-1}}{\lambda}\right)\Phi\left(-\frac{x_{j,T}}{\lambda}\right) - \Psi\left(\frac{x_{j,T-1}}{\lambda}\right)\Phi\left(\frac{x_{j,T}}{\lambda}\right).$$

However, i) the $j^{\text{th}}$ block does no depend on $\xi_{j',m',s'}$ with $j' \neq j, m' \in [T], s' \geq 1$ ii) all the previous parts in the $j^{\text{th}}$ block depend only on $\{\xi_{j,m',s}\}_{m'<m,s\geq 1}$, and iii) all the parts from (50) are zero because $m = \text{prog}(x_j) + 1$. Therefore

$$\mathbb{E}_{\bar{\xi}}\left[[\nabla f_i(x)]_{j,m} \times \frac{\bar{\xi}_j}{p_{w(t),i}}\right] \overset{\text{i)}}{=} \mathbb{E}_{\bar{\xi}_j}\left[[\nabla f_i(x)]_{j,m} \times \frac{\bar{\xi}_j}{p_{w(t),i}}\right]$$
$$\overset{\text{ii), iii)}}{=} [\nabla f_i(x)]_{j,m}\mathbb{E}_{\bar{\xi}_j}\left[\frac{\bar{\xi}_j}{p_{w(t),i}}\right] = [\nabla f_i(x)]_{j,m}.$$

For $m = \text{prog}(x_j) + 1$ and $b_{j,m} \neq (w(t), i)$, we have

$$\mathbb{E}_{\bar{\xi}}\left[[g_i(x; \bar{\xi}, t)]_{j,m}\right] = \mathbb{E}_{\bar{\xi}}\left[[\nabla f_i(x)]_{j,m}\right] \overset{\text{i)}}{=} \mathbb{E}_{\bar{\xi}_j}\left[[\nabla f_i(x)]_{j,m}\right] \overset{\text{ii), iii)}}{=} [\nabla f_i(x)]_{j,m}.$$

For $m < \text{prog}(x_j) + 1$, we have

$$\mathbb{E}_{\bar{\xi}}\left[[g_i(x; \bar{\xi}, t)]_{j,m}\right] = \mathbb{E}_{\bar{\xi}}\left[[\nabla f_i(x)]_{j,m}\right] \overset{\text{i)}}{=} \mathbb{E}_{\bar{\xi}_j}\left[[\nabla f_i(x)]_{j,m}\right] \overset{\text{ii)}}{=} [\nabla f_i(x)]_{j,m}.$$

Using the same reasoning, we have

$$\mathbb{E}_{\bar{\xi}}\left[\left\|g_i(x; \bar{\xi}, t) - \nabla f_i(x)\right\|^2\right]$$

$$= \mathbb{E}_{\bar{\xi}}\left[\sum_{\substack{j,m : b_{j,m}=(w(t),i),\\ m=\text{prog}(x_j)+1}} ([\nabla f_i(x)]_{j,m})^2\left(\frac{\bar{\xi}_j}{p_{w(t),i}} - 1\right)^2\right]$$

$$= \sum_{\substack{j,m : b_{j,m}=(w(t),i),\\ m=\text{prog}(x_j)+1}} ([\nabla f_i(x)]_{j,m})^2\mathbb{E}_{\bar{\xi}}\left[\left(\frac{\bar{\xi}_j}{p_{w(t),i}} - 1\right)^2\right]$$

$$= \sum_{\substack{j,m : b_{j,m}=(w(t),i),\\ m=\text{prog}(x_j)+1}} ([\nabla f_i(x)]_{j,m})^2\frac{\left(1 - p_{w(t),i}\right)}{p_{w(t),i}}$$

$$= \sum_{\substack{j,m : b_{j,m}=(w(t),i),\\ m=\text{prog}(x_j)+1}} \left(\left[\nabla F_T\left(\frac{x_j}{\lambda}\right)\right]_m\right)^2\frac{n^2L^2\lambda^2\left(1 - p_{w(t),i}\right)}{l_1^2 p_{w(t),i}}.$$

Due to Lemma D.1, we get $\|\nabla F_T(x)\|_\infty^2 \le \gamma_\infty^2$ for all $x \in \mathbb{R}^T$ and

$$
\mathbb{E}_{\bar{\xi}}\left[\left\|g_i(x;\bar{\xi},t) - \nabla f_i(x)\right\|^2\right] \le \sum_{\substack{j,m\,:\,b_{j,m}=(w(t),i),\\ m=\text{prog}(x_j)+1}} \frac{n^2 L^2 \lambda^2 \gamma_\infty^2 \left(1 - p_{w(t),i}\right)}{l_1^2 p_{w(t),i}}
$$

$$
= \frac{a_{w(t),i} n^2 L^2 \lambda^2 \gamma_\infty^2 \left(1 - p_{w(t),i}\right)}{l_1^2 p_{w(t),i}}
$$

because, due the condition $b_{j,m} = (w(t),i)$, we only consider the blocks from the set $\{s_{w(t),(i-1)}, \ldots, s_{w(t),i} - 1\}$ (see Alg. 5), take one coordinate from each block, and recall that $a_{w(t),i} := s_{w(t),i} - s_{w(t),(i-1)}$. Using the choice

$$
p_{w,i} := \min\left\{\frac{a_{w,i} n^2 L^2 \lambda^2 \gamma_\infty^2}{\sigma^2 l_1^2}, 1\right\} \tag{51}
$$

for all $w \in [T], i \in [n]$, we get

$$
\mathbb{E}_{\bar{\xi}}\left[\left\|g_i(x;\bar{\xi},t) - \nabla f_i(x)\right\|^2\right] \le \sigma^2.
$$

(**Step 3**: Analysis of Protocol)

Mirroring the proof of Theorem 6.1, using

$$
\lambda = \sqrt{\frac{4\varepsilon l_1^2}{L^2 S}}, \tag{52}
$$

one can show

$$
\inf_{k \in S_t}\left\|\nabla f(x^k)\right\|^2 > \frac{4\varepsilon}{S}\sum_{i=1}^{S}\inf_{k \in S_t}\mathbb{1}[\text{prog}(x_i^k) < T], \tag{53}
$$

where $x^k$ are points defined in Protocol 2, and

$$
\mathbb{P}\left(\frac{4\varepsilon}{S}\sum_{i=1}^{S}\inf_{k \in S_t}\mathbb{1}[\text{prog}(x_i^k) < T] \le 2\varepsilon\right) \le \frac{2}{S}\sum_{i=1}^{S}\mathbb{E}\left[\sup_{k \in S_t}\mathbb{1}[\text{prog}(x_i^k) \ge T]\right]. \tag{54}
$$

(**Step 4**: Bound on the expectations)

The time required to progress (get a non-zero value) to the $1^{\text{th}}$ coordinate in the block $x_j$ is at least

$$
\min\left\{\bar{t}_1, V_{i_{1,j}}^{-1}(\eta_{j,1})\right\}
$$

seconds, where $V_i^{-1}$ is defined in (8) and $\eta_{j,1}$ is a geometric random variable with the probability $p_{1,i_{1,j}}$. Because, due the condition $b_{j,1} \equiv (1, i_{1,j}) = (w(t),i)$, the mapping (49) zeroes out the first coordinate only if the algorithm returns $t^k < \bar{t}_1$ and $i_{1,j} = i$. Therefore, either the algorithm returns $t^k \ge \bar{t}_1$ and (49) does not zero out the coordinate of gradients, or it keeps returning $t^k < \bar{t}_1$, but then $A$ should calculate at least $\eta_{j,1}$ stochastic gradients in worker $i_{1,j}$ since the stochastic mapping zeros out the potentially non-zero coordinate in (49).

Recall that the "resisting" allocator (Alg. 5) tracks the random variable $\eta_{j,1}$.
**Opt. 1:** If $V_{i_{1,j}}^{-1}(\eta_{j,1}) < \bar{t}_1$, then Alg. 5 allocates the second part of the function $h_j$ to the same worker $i_{1,j}$, meaning that the time required to progress (get a non-zero value) to the $2^{\text{th}}$ coordinate in the block $x_j$ is at least

$$
\min\left\{\bar{t}_1, V_{i_{1,j}}^{-1}(\eta_{j,1} + \eta_{j,2})\right\} \ge \min\left\{\bar{t}_1, V_{i_{1,j}}^{-1}(\eta_{j,2})\right\}
$$

seconds, where $\eta_{j,2}$ is a geometric random variable with the probability $p_{1,i_{1,j}}$.
**Opt. 2:** If $V_{i_{1,j}}^{-1}(\eta_{j,1}) \ge \bar{t}_1$, then we allocate the second part to worker $i_{2,j}$, where $i_{2,j}$ such that

$s_{1,i_{2,j}-1} \leq j < s_{1,i_{2,j}}$, meaning that the time required to progress (get a non-zero value) to the $2^{\text{th}}$ coordinate in the block $x_j$ is at least

$$\min\left\{\bar{t}_2, V_{i_{2,j}}^{-1}\left(\eta_{j,2} + V_{i_{2,j}}(\bar{t}_1)\right)\right\}$$

seconds, where $\eta_{j,2}$ is a geometric random variable with the probability $p_{2,i_{2,j}}$, because either the algorithm returns $t^k \geq \bar{t}_2$, or it keeps returning $t^k < \bar{t}_2$, but then $A$ should first progress to the $1^{\text{th}}$ coordinate, which takes at least $\bar{t}_1$ seconds, and then should calculate at least $\eta_{j,2}$ stochastic gradients. This will take at least

$$\min\left\{t \geq 0 \,:\, \left\lfloor V_{i_{2,j}}(t) - V_{i_{2,j}}(\bar{t}_1) \right\rfloor \geq \eta_{j,2}\right\}$$
$$\overset{\text{(cont. of } V_{i_{2,j}})}{=} \min\left\{t \geq 0 \,:\, V_{i_{2,j}}(t) - V_{i_{2,j}}(\bar{t}_1) = \eta_{j,2}\right\}$$
$$\overset{(8)}{=} V_{i_{2,j}}^{-1}\left(\eta_{j,2} + V_{i_{2,j}}(\bar{t}_1)\right)$$

seconds. Notice that the parameter of the geometric random variable $\eta_{j,2}$ depends on the previous randomness.

In the case **Opt. 1**, we can only conclude that the algorithm $A$ will require at least $\bar{t}_0 \equiv 0$ seconds to get a non-zero value in the $T^{\text{th}}$ coordinate. In the case **Opt. 2**, we have better guarantees and can infer that the algorithm $A$ will require at least $\bar{t}_1 \geq \bar{t}_1$ seconds to get a non-zero value in the $T^{\text{th}}$ coordinate because the inequality $V_{i_{1,j}}^{-1}(\eta_{j,1}) \geq \bar{t}_1$ holds. Hence, the condition in Line 10 of Alg. 5 determines a necessary time to get the $T^{\text{th}}$ with a non-zero value.

For all $j \in [m]$, we have the following Markov process that generalizes our previous discussion.

---

**Algorithm 6** Markov process in the $j^{\text{th}}$ block

---

1: Current time window $w_m = 1$
2: **for** $m = 1, \ldots, T$ **do**
3:     Find $i_{w_m,j}$ such that $s_{w_m,i_{w_m,j}-1} \leq j < s_{w_m,i_{w_m,j}}$
4:     Draw an infinite i.i.d. sequence $\{\xi_{j,m,s}\}_{s=1}^{\infty}$ from Bernoulli$(p_{w_m,i_{w_m,j}})$
5:     Find the first moment when $\xi_{j,m,s} = 1$, i.e., $\eta_{j,m} = \inf\{s \geq 1 \,:\, \xi_{j,m,s} = 1\}$
6:     **if** $V_{i_{w_m,j}}^{-1}\left(\eta_{j,m} + V_{i_{w_m,j}}(\bar{t}_{w_m-1})\right) \geq \bar{t}_{w_m}$ **then**
7:         $w_{m+1} \leftarrow w_m + 1$
8:     **else**
9:         $w_{m+1} \leftarrow w_m$
10:     **end if**
11: **end for**
12: **Return:** $\bar{t}_{(w_T-1)}$ is a necessary time to get the $T^{\text{th}}$ non-zero coordinate in the $j^{\text{th}}$ block

---

The provided random Markov process determines the time $\bar{t}_{(w_T-1)}$ required to get the $T^{\text{th}}$ non-zero coordinate in the $j^{\text{th}}$ block.

For all $j \in [m], m \in [T]$, $\eta_{j,m}$ has the geometric distribution with the parameter $p_{w_m,i_{w_m,j}}$, which depends only on the previous random variables $\eta_{j,1}, \ldots, \eta_{j,m-1}$. Therefore, we can use Lemma G.2 and get

$$\mathbb{P}\left(\sum_{m=1}^{T} \mathbb{1}\left[\eta_{j,m} > \frac{1}{4p_{w_m,i_{w_m,j}}}\right] \leq \frac{T}{2} + \log\delta\right) \leq \delta$$

for all $\delta \in (0,1]$. The last inequality means that with a probability at least $^3/_4$, there exist $1 \leq m_1 < m_2 < \cdots < m_{\lfloor \frac{T-2}{2} \rfloor} \leq T$ such that

$$\eta_{j,m_k} > \frac{1}{4p_{w_{m_k},i_{w_{m_k},j}}} \geq \frac{1}{8p_{w_{m_k},i_{w_{m_k},j}}} \tag{55}$$

for all $k \in \left[\left\lfloor \frac{T-2}{2} \right\rfloor\right]$.

Due to (51) and (52), we have

$$p_{w,i} := \min\left\{\frac{4\varepsilon\gamma_\infty^2 a_{w,i} n^2}{\sigma^2 S}, 1\right\}.$$

Recall that $\{\bar{t}_w\}_{w\in[\bar{T}]}, \{s_{w,i}\}_{w\in[\bar{T}+1], i\in[n]}$, and $S \in \mathbb{N}$ are free parameters with the only conditions (47) and (48). Therefore, we can use Lemma G.4 with $K = 1$ and ensure that there exist parameters such that

$$V_i^{-1}\left(\frac{1}{8p_{w,i}} + V_i(\bar{t}_{w-1})\right) \geq \bar{t}_w.$$

for all $w \in [T], i \in [n]$. Putting the last inequality, (55), and Line 6 of Alg. 6 together, we can conclude that, with a probability at least $3/4$, the value of $w_T - 1$ is greater or equal to $\lfloor\frac{T-2}{2}\rfloor$ since the condition in Line 6 of Alg. 6 will hold at least $\lfloor\frac{T-2}{2}\rfloor$ times.

(**Step 5**: Endgame)
In the previous step, we prove that, with a probability at least $3/4$, the algorithm requires at least $\bar{t}_{\lfloor\frac{T-2}{2}\rfloor}$ seconds to progress to the $T^{\text{th}}$ coordinate of the $j^{\text{th}}$ block, where $\bar{t}_{\bar{T}}$ is defined in the proof of Lemma G.4.

Thus

$$\mathbb{E}\left[\sup_{k\in S_t} \mathbb{1}[\text{prog}(x_j^k) \geq T]\right] \leq \frac{1}{4}$$

for all $j \in [n]$ and for all $t \leq \frac{1}{2}\bar{t}_{\lfloor\frac{T-2}{2}\rfloor}$. We substitute these inequalities to (54) and (53), and get

$$\mathbb{P}\left(\frac{4\varepsilon}{S}\sum_{i=1}^{S}\inf_{k\in S_t}\mathbb{1}[\text{prog}(x_i^k) < T] \leq 2\varepsilon\right) \leq \frac{1}{2}$$

and

$$\mathbb{E}\left[\inf_{k\in S_t}\left\|\nabla f(x^k)\right\|^2\right] > \mathbb{E}\left[\frac{4\varepsilon}{S}\sum_{i=1}^{S}\inf_{k\in S_t}\mathbb{1}[\text{prog}(x_i^k) < T]\right] > 2\varepsilon\mathbb{P}\left(\frac{4\varepsilon}{S}\sum_{i=1}^{S}\inf_{k\in S_t}\mathbb{1}[\text{prog}(x_i^k) < T] > 2\varepsilon\right) \geq \varepsilon$$

for all

$$t \leq \frac{1}{2}\widetilde{t}_{\lfloor\frac{T-2}{2}\rfloor} \overset{(60)}{\leq} \frac{1}{2}\bar{t}_{\lfloor\frac{T-2}{2}\rfloor}.$$

Using the assumption $\varepsilon < c'L\Delta$ of the theorem, (46), and (52), we get

$$\left\lfloor\frac{T-2}{2}\right\rfloor \geq \left\lfloor c_1 \times \frac{L\Delta}{\varepsilon}\right\rfloor$$

for some universal constant $c_1$.

Finally, since $\widetilde{t}_w \geq \underline{t}_w$ for all $w \in [T]$, where the later sequence is defined in (18), we can take

$$t = \frac{1}{2}\underline{t}_{\lfloor c_1 \times \frac{L\Delta}{\varepsilon}\rfloor}.$$

$\square$

# G AUXILIARY LEMMAS

**Lemma G.1.** *Let $V_i : \mathbb{R}_+^\infty \to \mathbb{R}_+^\infty$ is a continuous and non-decreasing function for all $i \in [n]$. For all $\eta, b_1, \ldots, b_n \in \mathbb{R}_+^\infty$, the minimums of the sets*

$$\left\{t \geq 0 : \sum_{i=1}^{n}\lfloor V_i(t) - b_i\rfloor \geq \eta\right\}$$

*and*

$$\left\{ t \geq 0 : \left( \frac{1}{n} \sum_{i=1}^{n} \frac{1}{\lfloor V_i(t) - b_i \rfloor} \right)^{-1} \geq \eta \right\}$$

*exist, considering the convention* $\min\{\emptyset\} = \infty$.

*Proof.* We now focus on the first set. If the set is empty, then the minimum is $\infty$ by the convention. Otherwise, let us define

$$\underline{t} := \inf \left\{ t \geq 0 : \sum_{i=1}^{n} \lfloor V_i(t) - b_i \rfloor \geq \eta \right\} < \infty.$$

If the minimum does not exist, then

$$\sum_{i=1}^{n} \lfloor V_i(\underline{t}) - b_i \rfloor < \eta.$$

For all $i \in [n]$, the functions $V_i$ is continuous and non-decreasing meaning that there exists $\delta_i > 0$ such that

$$\lfloor V_i(\underline{t}) - b_i \rfloor = \lfloor V_i(\underline{t} + \delta) - b_i \rfloor.$$

for all $0 \leq \delta \leq \delta_i$. Let us take $\delta = \min_{i \in [n]} \delta_i > 0$, then

$$\sum_{i=1}^{n} \lfloor V_i(\underline{t} + \delta) - b_i \rfloor = \sum_{i=1}^{n} \lfloor V_i(\underline{t}) - b_i \rfloor < \eta.$$

This contradicts the fact that $\underline{t}$ is the infimum. The reasoning for the second set is the same. $\square$

**Lemma G.2.** *Let $T \geq 1$ and $\{\eta_i\}_{i=1}^{T}$ are geometric random variables such that given $\eta_1, \ldots, \eta_{i-1}$, $\eta_i \sim \text{Geometric}(p_{i,\eta_1,\ldots,\eta_{i-1}})$ and the probability $p_{i,\eta_1,\ldots,\eta_{i-1}} \in (0,1]$ depends only on $\eta_1, \ldots, \eta_{i-1}$ for all $i \in [T]$. Then*

$$\mathbb{P} \left( \sum_{i=1}^{T} \mathbb{1}\left[ \eta_i > \frac{1}{4 p_{i,\eta_1,\ldots,\eta_{i-1}}} \right] \leq \frac{T}{2} + \log \delta \right) \leq \delta$$

*for all $\delta \in (0,1]$.*

*Proof.* Let us consider the simplified notation $p_i \equiv p_{i,\eta_1,\ldots,\eta_{i-1}}$ and take any $\bar{T}, s > 0$. Using Chernoff's method, we get

$$\mathbb{P} \left( \sum_{i=1}^{T} \mathbb{1}\left[ \eta_i > \frac{1}{4p_i} \right] \leq \bar{T} \right) = \mathbb{P} \left( -s \sum_{i=1}^{T} \mathbb{1}\left[ \eta_i > \frac{1}{4p_i} \right] \geq -s\bar{T} \right)$$

$$\leq e^{s\bar{T}} \mathbb{E} \left[ e^{-s \sum_{i=1}^{T} \mathbb{1}\left[ \eta_i > \frac{1}{4p_i} \right]} \right] \qquad (56)$$

$$= e^{s\bar{T}} \mathbb{E} \left[ \prod_{i=1}^{T} \mathbb{E} \left[ e^{-s \mathbb{1}\left[ \eta_i > \frac{1}{4p_i} \right]} \middle| \eta_1, \ldots, \eta_{i-1} \right] \right],$$

where we use the definition of conditional expectation.

For all $i \in [T]$, we now consider the $i^{\text{th}}$ expectation separately:

$$\mathbb{E} \left[ e^{-s \mathbb{1}\left[ \eta_i > \frac{1}{4p_i} \right]} \middle| \eta_1, \ldots, \eta_{i-1} \right]$$

$$= \mathbb{P} \left( \eta_i \leq \frac{1}{4p_i} \middle| \eta_1, \ldots, \eta_{i-1} \right) + e^{-s} \mathbb{P} \left( \eta_i > \frac{1}{4p_i} \middle| \eta_1, \ldots, \eta_{i-1} \right)$$

$$= (1 - e^{-s})\mathbb{P}\left(\eta_i \leq \frac{1}{4p_i}\,\Big|\,\eta_1, \ldots, \eta_{i-1}\right) + e^{-s}.$$

Due the assumption of our theorem, given $\eta_1, \ldots, \eta_{i-1}$, $\eta_i$ is a geometric random variable with the parameter $p_i \equiv p_{i,\eta_1,\ldots,\eta_{i-1}}$. Therefore

$$\mathbb{E}\left[e^{-s\mathbb{1}\left[\eta_i > \frac{1}{4p_i}\right]}\,\Big|\,\eta_1, \ldots, \eta_{i-1}\right] = (1 - e^{-s})\left(1 - (1-p_i)^{\left\lfloor\frac{1}{4p_i}\right\rfloor}\right) + e^{-s},$$

where $1 - (1-p_i)^{\left\lfloor\frac{1}{4p_i}\right\rfloor} = 0$ if $p_i = 1$ and $\left\lfloor\frac{1}{4p_i}\right\rfloor = 0$. Since $1 - (1-p)^{\lfloor S\rfloor} \leq p\lfloor S\rfloor$ for all $p \in (0,1]$ and $S \geq 0$, we get

$$\mathbb{E}\left[e^{-s\mathbb{1}\left[\eta_i > \frac{1}{4p_i}\right]}\,\Big|\,\eta_1, \ldots, \eta_{i-1}\right] \leq (1 - e^{-s})p_i\left\lfloor\frac{1}{4p_i}\right\rfloor + e^{-s}$$

$$\leq (1 - e^{-s})p_i \times \frac{1}{4p_i} + e^{-s} = (1 - e^{-s})\frac{1}{4} + e^{-s}.$$

Let us take $s = 1$, then

$$\mathbb{E}\left[e^{-s\mathbb{1}\left[\eta_i > \frac{1}{4p_i}\right]}\,\Big|\,\eta_1, \ldots, \eta_{i-1}\right] \leq (1 - e^{-1})\frac{1}{4} + e^{-1} \leq e^{-1/2}.$$

We substitute the last inequality and the chosen value of $s$ to (56) and get

$$\mathbb{P}\left(\sum_{i=1}^{T} \mathbb{1}\left[\eta_i > \frac{1}{4p_i}\right] \leq \bar{T}\right) \leq e^{\bar{T} - \frac{T}{2}}.$$

For all $\delta \in (0, 1]$, we can take

$$\bar{T} = \frac{T}{2} + \log\delta$$

to ensure that

$$\mathbb{P}\left(\sum_{i=1}^{T} \mathbb{1}\left[\eta_i > \frac{1}{4p_i}\right] \leq \frac{T}{2} + \log\delta\right) \leq \delta.$$

$\square$

**Lemma G.3.** *Let* $\eta_{1,1}, \ldots, \eta_{1,K} \sim \text{Geometric}(p_1)$, $\eta_{2,1}, \ldots, \eta_{2,K} \sim \text{Geometric}(p_2)$, $\ldots$, $\eta_{\bar{T},1}, \ldots, \eta_{\bar{T},K} \sim \text{Geometric}(p_{\bar{T}})$ *are mutually independent geometric random variables. Then*

$$\mathbb{P}\left(\bigcup_{k \in [\bar{T}]} \left\{\sum_{j=1}^{K} \eta_{k,j} \leq \frac{K}{8p_k}\right\}\right) \leq \bar{T}e^{-K/2}.$$

*Proof.* Let us fix any $a_k \geq 0$ for all $k \in [\bar{T}]$. Using the union bound, we get

$$\mathbb{P}\left(\bigcup_{k \in [\bar{T}]} \left\{\sum_{j=1}^{K} \eta_{k,j} \leq a_k\right\}\right) \leq \sum_{k=1}^{\bar{T}} \mathbb{P}\left(\sum_{j=1}^{K} \eta_{k,j} \leq a_k\right). \tag{57}$$

For all $k \in [\bar{T}]$ and $s > 0$, we obtain the following series of inequalities:

$$\mathbb{P}\left(\sum_{j=1}^{K} \eta_{k,j} \leq a_k\right) = \mathbb{P}\left(-s\sum_{j=1}^{K} \eta_{k,j} \geq -sa_k\right) = \mathbb{P}\left(e^{-s\sum_{j=1}^{K}\eta_{k,j}} \geq e^{-sa_k}\right)$$

$$\leq e^{sa_k}\mathbb{E}\left[e^{-s\sum_{j=1}^K \eta_{k,j}}\right],$$

where we use Markov's inequality. Since the random variables are mutually independent, we have

$$\mathbb{P}\left(\sum_{j=1}^K \eta_{k,j} \leq a_k\right) \leq e^{sa_k}\left(\mathbb{E}\left[e^{-s\eta_{k,1}}\right]\right)^K = e^{sa_k}\left(\frac{p_k}{e^s - (1-p_k)}\right)^K,$$

where we use the moment-generating function of the geometric random variables. Since $p_k \geq 0$ and $e^s \geq 1 + s$ for all $s \in \mathbb{R}$, we get

$$\mathbb{P}\left(\sum_{j=1}^K \eta_{k,j} \leq a_k\right) \leq e^{sa_k}\left(\frac{p_k}{s}\right)^K.$$

Let us take $s = 4p_k$ to ensure that

$$\mathbb{P}\left(\sum_{j=1}^K \eta_{k,j} \leq a_k\right) \leq e^{4p_k a_k - K}.$$

We can take $a_k = \frac{K}{8p_k}$ to get

$$\mathbb{P}\left(\sum_{j=1}^K \eta_{k,j} \leq \frac{K}{8p_k}\right) \leq e^{-K/2}.$$

It is left to substitute the last inequality to (57). $\qquad \square$

**Lemma G.4.** *For all* $\bar{T}, K \in \mathbb{N}$, *there exist non-negative parameters* $\{\bar{t}_w\}_{w\in[\bar{T}]}, \{s_{w,i}\}_{w\in[\bar{T}+1], i\in[n]}$, *and* $S \in \mathbb{N}$ *such that*

*1.*

$$s_{w,0} \equiv 1 \leq s_{w,1} \leq \cdots \leq s_{w,n-1} \leq s_{w,n} \equiv S+1 \quad \forall w \in [\bar{T}+1], \tag{58}$$
$$\bar{t}_0 \equiv 0 \leq \bar{t}_1 \leq \cdots \leq \bar{t}_{\bar{T}} \leq \bar{t}_{\bar{T}+1} \equiv \infty.$$

*2.*

$$V_i^{-1}\left(\frac{K}{8p_{w,i}} + V_i(\bar{t}_{w-1})\right) \geq \bar{t}_w. \tag{59}$$

*for all* $w \in [\bar{T}], i \in [n]$,

*3.*

$$\bar{t}_w \geq \tilde{t}_w := \min\left\{t \geq 0 \ : \ \left(\frac{1}{n}\sum_{i=1}^n \frac{1}{\left\lceil\frac{16(V_i(t)-V_i(\tilde{t}_{w-1}))}{K}\right\rceil}\right)^{-1} \geq \max\left\{\frac{\sigma^2}{32\gamma_\infty^2 \varepsilon n}, 1\right\}\right\} \quad (\tilde{t}_0 \equiv 0) \tag{60}$$

*for all* $w \in [\bar{T}]$,

*where*

$$p_{w,i} := \min\left\{\frac{4\varepsilon\gamma_\infty^2 a_{w,i} n^2}{\sigma^2 S}, 1\right\}, \tag{61}$$

$a_{w,i} := s_{w,i} - s_{w,(i-1)}$, *and* $\varepsilon, \gamma_\infty^2, \sigma^2$ *are arbitrarily non-negative constants.*

*Proof.* We have the free parameters $\{\bar{t}_w\}_{w\in[\bar{T}]}, \{s_{w,i}\}_{w\in[\bar{T}+1], i\in[n]}$, and $S \in \mathbb{N}$ with the only condition

$$s_{w,0} \equiv 1 \leq s_{w,1} \leq \cdots \leq s_{w,n-1} \leq s_{w,n} \equiv S+1 \quad \forall w \in [\bar{T}+1], \tag{62}$$
$$\bar{t}_0 \equiv 0 \leq \bar{t}_1 \leq \cdots \leq \bar{t}_{\bar{T}} \leq \bar{t}_{\bar{T}+1} \equiv \infty.$$

We now choose values of these parameters to ensure that (59) holds. Instead of $\{s_{w,i}\}$, we will work with $\{a_{w,i}\}$, then one can restore $\{s_{w,i}\}$ using the definition $a_{w,i} := s_{w,i} - s_{w,(i-1)}$. We have to ensure that

$$\sum_{i=1}^{n} a_{w,i} = S \tag{63}$$

holds for all $w \in [\bar{T}+1]$ to get (62). It is sufficient to validate that

$$p_{w,i} < \frac{K}{8(V_i(\bar{t}_w) - V_i(\bar{t}_{w-1}))} \qquad (\bar{t}_0 \equiv 0)$$

for all $w \in [\bar{T}]$ to guarantee that

$$V_i^{-1}\left(\frac{K}{8p_{w,i}} + V_i(\bar{t}_{w-1})\right) \geq \bar{t}_w$$

for all $w \in [\bar{T}]$. Due to (61), it is sufficient to find $\{a_{w,i}\}$, $\{\bar{t}_w\}$, and $S$ such that (63) holds and

$$\min\left\{\frac{a_{w,i}n^2 4\varepsilon\gamma_\infty^2}{S\sigma^2}, 1\right\}$$
$$\overset{(63)}{=} \min\left\{\frac{a_{w,i}n^2 4\varepsilon\gamma_\infty^2}{\sum_{i=1}^{n} a_{w,i}\sigma^2}, 1\right\} < \frac{K}{8(V_i(\bar{t}_w) - V_i(\bar{t}_{w-1}))} \tag{64}$$

for all $i \in [n]$ and for all $w \in [\bar{T}]$.
Assume that $\bar{t}_{w-1}$ is defined ($\bar{t}_0 \equiv 0$), and let us now consider $w \in [\bar{T}]$.
Let us define

$$\bar{t}_w^1 := \max_{j \in [n]} V_j^{-1}\left(\frac{K}{16} + V_j(\bar{t}_{w-1})\right) \tag{65}$$

and

$$\bar{t}_w^2 := \min\left\{t \geq 0 : \sum_{i=1}^{n} \frac{K\sigma^2}{n^2 4\varepsilon\gamma_\infty^2 \left(V_i(t) - V_i(\bar{t}_{w-1})\right)} = 64\right\}. \tag{66}$$

**Opt. 1:** If $\bar{t}_w^1 > \bar{t}_w^2$, then we take $\bar{t}_w = \bar{t}_w^1$, $\bar{a}_{w,i} := 0$ for all $i \neq j^*$, and $\bar{a}_{w,j^*} := 1$, where

$$j^* = \arg\max_{j \in [n]} V_j^{-1}\left(\frac{K}{16} + V_j(\bar{t}_{w-1})\right). \tag{67}$$

**Opt. 2:** If $\bar{t}_w^1 \leq \bar{t}_w^2$, then we take $\bar{t}_w = \bar{t}_w^2$, and [9]

$$\bar{a}_{w,i} := \left\lfloor \frac{\max_{i \in [n]}\{V_i(\bar{t}_w) - V_i(\bar{t}_{w-1})\}}{V_i(\bar{t}_w) - V_i(\bar{t}_{w-1})} \right\rfloor \tag{68}$$

for all $i \in [n]$.

For $w = \bar{T} + 1$, we take $\bar{a}_{w,i} := 0$ for all $i \neq 1$, and $\bar{a}_{w,1} := 1$.

We choose the following $S$ :

$$S = \max_{w \in [\bar{T}+1]}\left(\sum_{i=1}^{n} \bar{a}_{w,i}\right),$$

and for all $w \in \text{Arg}\max_{w \in [\bar{T}+1]}\left(\sum_{i=1}^{n} \bar{a}_{w,i}\right)$, we take $a_{w,i} := \bar{a}_{w,i}$. Let us take any $w$ such that $\sum_{i=1}^{n} \bar{a}_{w,i} < S$, then, for all $w \in [\bar{T}+1]$, there exists the smallest $k_w \geq 2$ that yields

$$\sum_{i=1}^{n} k_w \times \bar{a}_{w,i} \geq S.$$

---

[9] $V_i(\bar{t}_w) > V_i(\bar{t}_{w-1})$ for all $i \in [n]$ since $\bar{t}_w \geq \bar{t}_w^1$ and $V_i(\bar{t}_w) \geq \frac{K}{16} + V_i(\bar{t}_{w-1})$ for all $i \in [n]$, due to (65) and the definition (8).

**Opt. 1:** If $\bar{t}_w^1 > \bar{t}_w^2$, then we take $a_{w,j^*} := k_w \times \bar{a}_{w,j^*} = k_w$ and $a_{w,i} := k_w \times \bar{a}_{w,i} = 0$ for all $i \neq j^*$ ($j^*$ from (67)) to ensure that $\sum_{i=1}^n a_{w,i} = S$.

**Opt. 2:** If $\bar{t}_w^1 \leq \bar{t}_w^2$, there exist $r_{w,i} \in \{0, \ldots, \bar{a}_{w,i}\}$ that if we take

$$a_{w,i} := (k_w - 1) \times \bar{a}_{w,i} + r_{w,i},$$

then we guarantee the equality $\sum_{i=1}^n a_{w,i} = S$.

It is left to ensure that (64) holds.

**Opt. 1:** If $\bar{t}_w^1 > \bar{t}_w^2$, then (64) holds since

$$\min\left\{\frac{a_{w,i} n^2 4\varepsilon\gamma_\infty^2}{\sum_{i=1}^n a_{w,i}\sigma^2}, 1\right\} = 0 \quad \forall i \neq j^*$$

and

$$\min\left\{\frac{a_{w,i} n^2 4\varepsilon\gamma_\infty^2}{\sum_{i=1}^n a_{w,i}\sigma^2}, 1\right\} \leq 1 < \frac{K}{8(V_i(\bar{t}_w) - V_i(\bar{t}_{w-1}))} \text{ for } i = j^*,$$

where the last inequality due to the definition (8) and the choice of $\bar{t}_w^1$:

$$V_{j^*}(\bar{t}_w^1) = \frac{K}{16} + V_{j^*}(\bar{t}_{w-1}) < \frac{K}{8} + V_{j^*}(\bar{t}_{w-1}).$$

**Opt. 2:** If $\bar{t}_w^1 \leq \bar{t}_w^2$, then (64) holds since

$$\frac{a_{w,i}}{\sum_{i=1}^n a_{w,i}} \leq \frac{k_w \bar{a}_{w,i}}{(k_w - 1)\sum_{i=1}^n \bar{a}_{w,i}} \leq 2\frac{\bar{a}_{w,i}}{\sum_{i=1}^n \bar{a}_{w,i}}$$

because $k_w \geq 2$. Using (68) and $x \geq \lfloor x \rfloor \geq \frac{x}{2}$ for all $x \geq 1$, we get

$$\frac{a_{w,i}}{\sum_{i=1}^n a_{w,i}} \leq 4\frac{\frac{1}{V_i(\bar{t}_w) - V_i(\bar{t}_{w-1})}}{\sum_{i=1}^n \frac{1}{V_i(\bar{t}_w) - V_i(\bar{t}_{w-1})}} \overset{(66)}{\leq} \frac{K\sigma^2}{64 n^2 \varepsilon\gamma_\infty^2 \left(V_i(\bar{t}_w) - V_i(\bar{t}_{w-1})\right)}.$$

The last inequality ensures that (64) holds. In total, we have $\bar{t}_w = \max\{\bar{t}_w^1, \bar{t}_w^2\}$.

It is left to prove (60). Let us define

$$\widetilde{t}_w := \min\left\{t \geq 0 : \left(\frac{1}{n}\sum_{i=1}^n \frac{1}{\left\lfloor \frac{16(V_i(t) - V_i(\widetilde{t}_{w-1}))}{K}\right\rfloor}\right)^{-1} \geq \max\left\{\frac{\sigma^2}{32\gamma_\infty^2 \varepsilon n}, 1\right\}\right\} \qquad (\widetilde{t}_0 \equiv 0). \tag{69}$$

We know that $\widetilde{t}_0 \equiv 0 \leq \bar{t}_0 \equiv 0$. Using a proof by induction and assuming $\widetilde{t}_{w-1} \leq \bar{t}_{w-1}$, we have

$$\widetilde{t}_w = \min\left\{t \geq \max_{j \in [n]} V_j^{-1}\left(\frac{K}{16} + V_j(\widetilde{t}_{w-1})\right) : \left(\frac{1}{n}\sum_{i=1}^n \frac{1}{\left\lfloor \frac{16(V_i(t) - V_i(\widetilde{t}_{w-1}))}{K}\right\rfloor}\right)^{-1} \geq \max\left\{\frac{\sigma^2}{32\gamma_\infty^2 \varepsilon n}, 1\right\}\right\}$$

$$= \min\left\{t \geq \max_{j \in [n]} V_j^{-1}\left(\frac{K}{16} + V_j(\widetilde{t}_{w-1})\right) : \frac{1}{n}\sum_{i=1}^n \frac{1}{\left\lfloor \frac{16(V_i(t) - V_i(\widetilde{t}_{w-1}))}{K}\right\rfloor} \leq \min\left\{\frac{32\gamma_\infty^2 \varepsilon n}{\sigma^2}, 1\right\}\right\}$$

because if $\left(\frac{1}{n}\sum_{i=1}^n \frac{1}{\left\lfloor \frac{16(V_i(t) - V_i(\widetilde{t}_{w-1}))}{K}\right\rfloor}\right)^{-1} \geq \max\left\{\frac{\sigma^2}{32\gamma_\infty^2 \varepsilon n}, 1\right\}$, then, necessarily, $16(V_i(t) -$

$V_i(\widetilde{t}_{w-1})) \geq K$ for all $i \in [n]$. Then

$$\widetilde{t}_w = \min\left\{t \geq \max_{j \in [n]} V_j^{-1}\left(\frac{K}{16} + V_j(\widetilde{t}_{w-1})\right) : \frac{1}{n}\sum_{i=1}^n \frac{1}{\left\lfloor \frac{16(V_i(t) - V_i(\widetilde{t}_{w-1}))}{K}\right\rfloor} \leq \frac{32\gamma_\infty^2 \varepsilon n}{\sigma^2}\right\}$$

because $\frac{1}{n} \sum_{i=1}^{n} \frac{1}{\left\lceil \frac{16(V_i(t) - V_i(\tilde{t}_{w-1}))}{K} \right\rceil} \leq 1$ for all $t \geq \max_{j \in [n]} V_j^{-1} \left( \frac{K}{16} + V_j(\tilde{t}_{w-1}) \right)$. Further

$$\tilde{t}_w \leq \min \left\{ t \geq \max_{j \in [n]} V_j^{-1} \left( \frac{K}{16} + V_j(\bar{t}_{w-1}) \right) \; : \; \frac{1}{n} \sum_{i=1}^{n} \frac{1}{\left\lceil \frac{16(V_i(t) - V_i(\bar{t}_{w-1}))}{K} \right\rceil} \leq \frac{32 \gamma_\infty^2 \varepsilon n}{\sigma^2} \right\}$$

$$\leq \min \left\{ t \geq \max_{j \in [n]} V_j^{-1} \left( \frac{K}{16} + V_j(\bar{t}_{w-1}) \right) \; : \; \sum_{i=1}^{n} \frac{\sigma^2}{n^2 4 \varepsilon \gamma_\infty^2 \frac{(V_i(t) - V_i(\bar{t}_{w-1}))}{K}} \leq 64 \right\}$$

because $\tilde{t}_{w-1} \leq \bar{t}_{w-1}$, $\{V_i\}$ are non-decreasing, and $\lfloor x \rfloor \geq \frac{x}{2}$ for all $x \geq 1$. Then

$$\tilde{t}_w \leq \min \left\{ t \geq \max_{j \in [n]} V_j^{-1} \left( \frac{K}{16} + V_j(\bar{t}_{w-1}) \right) \; : \; \sum_{i=1}^{n} \frac{K \sigma^2}{n^2 4 \varepsilon \gamma_\infty^2 (V_i(t) - V_i(\bar{t}_{w-1}))} \leq 64 \right\} \leq \max\{\bar{t}_w^1, \bar{t}_w^2\} = \bar{t}_w$$

due to the definitions (65), (66) of $\bar{t}_w^1, \bar{t}_w^2$, and $\bar{t}_w$. $\qquad\qquad\square$

**Lemma G.5.** *(Tyurin et al., 2024a)[Section K] Consider a sequence $\infty > v_1 \geq \ldots \geq v_n$ and fix some $S > 0$. For all $j \in [n]$, define*

$$g(j) := \left( \sum_{i=1}^{j} v_i \right)^{-1} (S + j).$$

1. *Let $j_{\max}^*$ be the largest index such that $\min_{j \in [n]} g(j) = g(j_{\max}^*)$. For $j_{\max}^* < n$, we have*

$$\min_{j \in [n]} g(j) < \frac{1}{v_{(j_{\max}^* + 1)}}.$$

2. *Let $j^*$ be any index such that $\min_{j \in [n]} g(j) = g(j^*)$. For $j^* < n$, we have*

$$\min_{j \in [n]} g(j) \leq \frac{1}{v_{(j^* + 1)}}.$$

3. *Let $j_{\min}^*$ be the smallest index such that $\min_{j \in [n]} g(j) = g(j_{\min}^*)$. Then*

$$\frac{1}{v_{j_{\min}^*}} < \min_{j \in [n]} g(j).$$

4. *Let $j^*$ be any index such that $\min_{j \in [n]} g(j) = g(j^*)$. Then*

$$\frac{1}{v_{j^*}} \leq \min_{j \in [n]} g(j).$$

## H  CONVEX SETTING

In the convex setting, the time complexities do not change significantly in a conceptual sense compared to the nonconvex case. Therefore, we will provide a somewhat less detailed description in this section. The obtained time complexities also hinge on the sequences (12) and (19). The only difference is the number of iterations that the methods do in each particular setup.

Using the same reasoning as in other sections and (Tyurin & Richtárik, 2023)[Section B], we conjecture that the following results are optimal up to constant factors. It is sufficient to use an appropriate "difficult" function (Guzmán & Nemirovski, 2015; Nesterov, 2018; Woodworth et al., 2018) designed for the convex domain instead of (23).

We use the following assumptions:

**Assumption H.1.** The function $f$ is convex and attains a minimum at some point $x^* \in \mathbb{R}^d$.

**Assumption H.2.** The function $f$ is $M$–Lipschitz, i.e.,

$$|f(x) - f(y)| \le M \|x - y\|, \quad \forall x, y \in \mathbb{R}^d$$

for some $M \in (0, \infty]$.

**Assumption H.3** (Homogeneous setup)**.** For all $i \in [n]$, worker $i$ can only calculate $\nabla f(x; \xi)$. For all $x \in \mathbb{R}^d$, stochastic (sub)gradients $\nabla f(x; \xi)$ are unbiased and are $\sigma^2$-variance-bounded, i.e., $\mathbb{E}_\xi [\nabla f(x; \xi)] \in \partial f(x)$ and $\mathbb{E}_\xi \left[ \|\nabla f(x; \xi) - \mathbb{E}_\xi [\nabla f(x; \xi)]\|^2 \right] \le \sigma^2$, where $\sigma^2 \ge 0$.

**Assumption H.4** (Heterogeneous setup)**.** For all $i \in [n]$, worker $i$ can only calculate $\nabla f_i(x; \xi_i)$. For all $x \in \mathbb{R}^d$, $i \in [n]$ stochastic (sub)gradients $\nabla f_i(x; \xi_i)$ are unbiased and are $\sigma^2$-variance-bounded, i.e., $\mathbb{E}_{\xi_i} [\nabla f_i(x; \xi_i)] \in \partial f_i(x)$ and $\mathbb{E}_{\xi_i} \left[ \|\nabla f_i(x; \xi_i) - \mathbb{E}_{\xi_i} [\nabla f_i(x; \xi_i)]\|^2 \right] \le \sigma^2$, where $\sigma^2 \ge 0$.

We consider *four* cases.

### H.1 HOMOGENEOUS SETUP AND NONSMOOTH CASE

**Theorem H.5.** *[(Tyurin & Richtárik, 2023)] Let Assumptions H.1, H.2 and H.3 hold. Choose any $\varepsilon > 0$. Let us take the batch size $S = \max \left\{ \lceil \sigma^2/M^2 \rceil, 1 \right\}$, stepsize $\gamma = \frac{\varepsilon}{M^2 + \sigma^2/S} \in \left[ \frac{\varepsilon}{2M^2}, \frac{\varepsilon}{M^2} \right]$ in Method 3. Then after $K \ge 2M^2R^2/\varepsilon^2$ iterations the method guarantees $\mathbb{E} \left[ f(\widehat{x}^K) \right] - f(x^*) \le \varepsilon$, where $\widehat{x}^K = \frac{1}{K} \sum_{k=0}^{K-1} x^k$ and $R = \|x^* - x^0\|$.*

**Theorem H.6.** *Consider the assumptions and the parameters from Theorem H.5, plus Assumption 3.1. Then Method 3 (*Rennala SGD*) converges after at most $\bar{t}_{\left\lceil \frac{2M^2R^2}{\varepsilon^2} \right\rceil}$ seconds, where the sequence $\bar{t}_k$ is defined in (12).*

*Proof.* The proof is identical to the proof of Theorem 5.3. $\qquad\square$

### H.2 HOMOGENEOUS SETUP AND SMOOTH CASE

In the homogeneous and smooth case, we can use an accelerated technique (Nesterov, 1983; Lan, 2020). Instead of Line 11 of Method 3, we use the following steps suggested by Lan (2020):

$$
\begin{aligned}
&\gamma_{k+1} = \gamma \cdot (k+1), \quad \alpha_{k+1} = 2/(k+2) \\
&y^{k+1} = (1 - \alpha_{k+1})x^k + \alpha_{k+1}u^k, \qquad (u^0 = x^0) \\
&u^{k+1} = u^k - \frac{\gamma_{k+1}}{s^k} g^k, \\
&x^{k+1} = (1 - \alpha_{k+1})x^k + \alpha_{k+1}u^{k+1}.
\end{aligned}
\tag{70}
$$

A new method with these steps is called the Accelerated Rennala SGD method (Tyurin & Richtárik, 2023).

**Theorem H.7.** *[(Tyurin & Richtárik, 2023)] Let Assumptions H.1, 1.1 and 1.3 hold. Choose any $\varepsilon > 0$. Let us take the batch size $S = \max \left\{ \lceil (\sigma^2 R)/(\varepsilon^{3/2}\sqrt{L}) \rceil, 1 \right\}$, and $\gamma = \min \left\{ \frac{1}{4L}, \left[ \frac{3R^2S}{4\sigma^2(K+1)(K+2)^2} \right]^{1/2} \right\}$ in Accelerated Method 3 (*Accelerated Rennala SGD*), then after $K \ge \frac{8\sqrt{L}R}{\sqrt{\varepsilon}}$ iterations the method guarantees that $\mathbb{E} \left[ f(x^K) \right] - f(x^*) \le \varepsilon$, where $R = \|x^* - x^0\|$.*

**Theorem H.8.** *Consider the assumptions and the parameters from Theorem H.7, plus Assumption 3.1. Then Accelerated Method 3 (*Accelerated Rennala SGD*) converges after at most $\bar{t}_{\left\lceil \frac{8\sqrt{L}R}{\sqrt{\varepsilon}} \right\rceil}$ seconds, where the sequence $\bar{t}_k$ is defined in (12).*

*Proof.* The proof is identical to the proof of Theorem 5.3. $\qquad\square$

### H.3 HETEROGENEOUS SETUP AND NONSMOOTH CASE

Consider the heterogeneous setup discussed in Section 1.

**Theorem H.9.** *Let Assumptions H.1, H.2, and H.4 hold. Choose any $\varepsilon > 0$. Let us take $S = \max\left\{\left\lceil \sigma^2/M^2 \right\rceil, n\right\}$, and $\gamma = \frac{\varepsilon}{M^2 + \sigma^2/S} \in \left[\frac{\varepsilon}{2M^2}, \frac{\varepsilon}{M^2}\right]$ in Method 4, then after $K \geq 2M^2R^2/\varepsilon^2$ iterations the method guarantees that $\mathbb{E}\left[f(\widehat{x}^K)\right] - f(x^*) \leq \varepsilon$, where $\widehat{x}^K = \frac{1}{K}\sum_{k=0}^{K-1} x^k$ and $R = \left\|x^* - x^0\right\|$.*

*Proof.* Notice that Malenia SGD is equivalent to the classical SGD method with the step

$$x^{k+1} = x^k - \gamma \frac{1}{n}\sum_{j=1}^{n}\frac{1}{B_j}\sum_{i=1}^{B_j}\nabla f_j(x^k; \xi_{j,i}^k),$$

where the variance of the unbiased gradient estimator can be bounded in the following way:

$$\mathbb{E}_{\xi^k}\left[\left\|\frac{1}{n}\sum_{j=1}^{n}\frac{1}{B_j}\sum_{i=1}^{B_j}\nabla f_j(x^k; \xi_{j,i}^k) - \nabla f(x^k)\right\|^2\right] \leq \frac{\sigma^2}{n^2}\left(\sum_{j=1}^{n}\frac{1}{B_j}\right).$$

In Method 4, we ensure that $\left(\sum_{j=1}^{n}\frac{1}{B_j}\right) \leq \frac{n^2}{S}$. Therefore

$$\mathbb{E}_{\xi^k}\left[\left\|\frac{1}{n}\sum_{j=1}^{n}\frac{1}{B_j}\sum_{i=1}^{B_j}\nabla f_j(x^k; \xi_{j,i}^k) - \nabla f(x^k)\right\|^2\right] \leq \frac{\sigma^2}{S}.$$

Thus, we can use the classical result from the literature (e.g. (Lan, 2020)). We get

$$\mathbb{E}\left[f(\widehat{x}^K)\right] - f(x^*) \leq \varepsilon$$

if

$$K \geq \frac{2M^2\left\|x^* - x^0\right\|^2}{\varepsilon^2} \geq \frac{(M^2 + \frac{\sigma^2}{S})\left\|x^* - x^0\right\|^2}{\varepsilon^2}$$

for the stepsize

$$\gamma = \frac{\varepsilon}{M^2 + \frac{\sigma^2}{S}} \in \left[\frac{\varepsilon}{2M^2}, \frac{\varepsilon}{M^2}\right],$$

where we use the fact that $S \geq \sigma^2/M^2$. $\qquad\square$

**Theorem H.10.** *Consider the assumptions and the parameters from Theorem H.9, plus Assumption 3.1. Then Method 4 (Malenia SGD) converges after at most $\bar{t}_{\left\lceil \frac{2M^2R^2}{\varepsilon^2} \right\rceil}$ seconds, where the sequence $\bar{t}_k$ is defined in (19).*

*Proof.* The proof is identical to the proof of Theorem 6.4. $\qquad\square$

### H.4 HETEROGENEOUS SETUP AND SMOOTH CASE

Using the same idea as in Section H.2, we will modify Malenia SGD and, instead of Line 13 from Algorithm 4, we use the lines (70). Such a method is called Accelerated Malenia SGD.

**Theorem H.11.** *Let Assumptions H.1 and 1.1, and 1.4 hold. Choose any $\varepsilon > 0$. Let us take $S = \max\left\{\left\lceil (\sigma^2 R)/(\varepsilon^{3/2}\sqrt{L}) \right\rceil, n\right\}$, and $\gamma = \min\left\{\frac{1}{4L}, \left[\frac{3R^2 S}{4\sigma^2(K+1)(K+2)^2}\right]^{1/2}\right\}$ in Accelerated Method 4 (Accelerated Malenia SGD), then after $K \geq \frac{8\sqrt{L}R}{\sqrt{\varepsilon}}$ iterations the method guarantees that $\mathbb{E}\left[f(x^K)\right] - f(x^*) \leq \varepsilon$, where $R = \left\|x^* - x^0\right\|$.*

*Proof.* Accelerated Malenia SGD is equivalent to the classical accelerated stochastic gradient method with a mini-batch from (Lan, 2020). The proof repeats the proofs of Theorem H.7 and Theorem H.9. $\qquad\square$

**Theorem H.12.** *Consider the assumptions and the parameters from Theorem H.11, plus Assumption 3.1. Then Accelerated Method 4 (*Accelerated Malenia SGD*) converges after at most $\bar{t}_{\lceil \frac{8\sqrt{L}R}{\sqrt{\varepsilon}} \rceil}$ seconds, where the sequence $\bar{t}_k$ is defined in (19).*

*Proof.* The proof is identical to the proof of Theorem 6.4. □

## I PROOF OF EXAMPLES

### I.1 HOMOGENEOUS SETUP

**Example 5.4.** [Fixed Computation Model] Consider Example 3.2 with $v_i(t) = v_i \in \mathbb{R}_+$ for all $t \geq 0, i \in [n]$. Then, for all $i \in [n]$, $V_i(t) = v_i t$ and

$$\bar{t}_{\lceil \frac{24L\Delta}{\varepsilon} \rceil} = \Theta \left( \min_{m \in [n]} \left( \frac{1}{m} \sum_{i=1}^{m} v_{\pi_i} \right)^{-1} \left( \frac{L\Delta}{\varepsilon} + \frac{L\Delta\sigma^2}{m\varepsilon^2} \right) \right), \tag{13}$$

$\pi$ is a permutation such that $v_{\pi_1} \geq \cdots \geq v_{\pi_n}$. The proofs of the examples are in Section I.

*Proof.* Clearly, $V_i(t) = \int_0^t v_i d\tau = v_i t$. Substituting this to (12), we get $\bar{t}_k = \min \left\{ t \geq 0 : \sum_{i=1}^{n} \lfloor v_i(t - \bar{t}_{k-1}) \rfloor \geq \max \left\{ \lceil \sigma^2/\varepsilon \rceil, 1 \right\} \right\} (\bar{t}_0 \equiv 0)$. Equivalently, we have to find $\bar{\delta}$ such that

$$\bar{\delta} = \min \left\{ \delta \geq 0 : \sum_{i=1}^{n} \lfloor v_i \delta \rfloor \geq \max \left\{ \lceil \sigma^2/\varepsilon \rceil, 1 \right\} \right\} \tag{71}$$

and obtain $\bar{t}_k = \bar{\delta} + \bar{t}_{k-1} = k\bar{\delta}$. Let us show that

$$\bar{\delta}_{1/4} \leq \bar{\delta} \leq \bar{\delta}_4 \tag{72}$$

where $\pi$ is a permutation such that $v_{\pi_1} \geq \cdots \geq v_{\pi_n}$, and the $\bar{\delta}$. is defined as

$$\bar{\delta}_c := c \times \min_{j \in [n]} \left( \sum_{i=1}^{j} v_{\pi_i} \right)^{-1} \left( \frac{\sigma^2}{\varepsilon} + j \right) = c \times \left( \sum_{i=1}^{j^*} v_{\pi_i} \right)^{-1} \left( \frac{\sigma^2}{\varepsilon} + j^* \right), \tag{73}$$

where $j^*$ is the largest index that minimizes the formula, and $c > 0$ is a constant. Using Lemma G.5, we obtain $1/(4v_{\pi_j^*}) \leq \bar{\delta}_{1/4} < 1/(4v_{\pi_{(j^*+1)}})$, where we define $1/v_{n+1} \equiv \infty$ for convenience. Thus

$$\sum_{i=1}^{n} \lfloor v_i \bar{\delta}_{1/4} \rfloor \overset{\bar{\delta}_{1/4} < 1/v_{j^*+1}}{=} \sum_{i=1}^{j^*} \lfloor v_{\pi_i} \bar{\delta}_{1/4} \rfloor \leq \sum_{i=1}^{j^*} \left( 2v_{\pi_i} \bar{\delta}_{1/2} - \frac{1}{2} \right) \overset{(73)}{=} \frac{\sigma^2}{2\varepsilon} < \max \left\{ \lceil \frac{\sigma^2}{\varepsilon} \rceil, 1 \right\},$$

where the first inequality due to $\lfloor x \rfloor \leq 2x - \frac{1}{2}$ for all $x \geq 1/4$. Therefore $\bar{\delta} \geq \bar{\delta}_{1/4}$. On the other hand

$$\sum_{i=1}^{n} \lfloor v_i \bar{\delta}_4 \rfloor \geq \sum_{i=1}^{j^*} \lfloor v_{\pi_i} \bar{\delta}_4 \rfloor \overset{L.G.5, \bar{\delta}_4 \geq 1/v_{j^*}}{\geq} \frac{1}{2} \sum_{i=1}^{j^*} v_{\pi_i} \bar{\delta}_4 \overset{(73)}{=} 2 \left( \frac{\sigma^2}{\varepsilon} + j^* \right) \geq \max \left\{ \lceil \frac{\sigma^2}{\varepsilon} \rceil, 1 \right\}.$$

We can conclude that $\bar{\delta} \leq \bar{\delta}_4$, and using the inequality $\bar{\delta} \geq \bar{\delta}_{1/4}$, we have proved (72). It is left to use the equality $\bar{t}_k = k\bar{\delta}$. □

**Example 5.5.** [Nonlinear Trend] Assume that $v_i(t) = v_i \times g(t)$ with $v_i > 0$ for all $i \in [n]$ and a continuous almost everywhere positive[10] function $g(t) : \mathbb{R}_+^\infty \to \mathbb{R}_+$. Then

$$\bar{t}_{\lceil \frac{24L\Delta}{\varepsilon} \rceil} = G^{-1} \left( c_1 \cdot \min_{m \in [n]} \left( \frac{1}{m} \sum_{i=1}^{m} v_{\pi_i} \right)^{-1} \left( \frac{L\Delta}{\varepsilon} + \frac{L\Delta\sigma^2}{m\varepsilon^2} \right) \right), \tag{14}$$

where $\pi$ is a permutation such that $v_{\pi_1} \geq \cdots \geq v_{\pi_n}$, $G(t) := \int_0^t g(\tau)d\tau$, and $c_1 \in [1/4, 4]$ (can depend on other parameters but is bounded).

---

[10]We can relax these assumptions to *measurability and non-negativity*, but the proof will be more technical.

*Proof.* We have $V_i(t) = \int_0^t v_i g(\tau) d\tau = v_i G(t)$. We substitute this equality to (12) and get $\bar{t}_k = \min\left\{t \geq 0 : \sum_{i=1}^n \lfloor v_i(G(t) - G(\bar{t}_{k-1})) \rfloor \geq \max\left\{\lceil \sigma^2/\varepsilon \rceil, 1\right\}\right\}$. Since $G(t)$ is continuous and increasing, one can show that $G(\bar{t}_k) = \min\left\{t \geq 0 : \sum_{i=1}^n \lfloor v_i(t - G(\bar{t}_{k-1})) \rfloor \geq \max\left\{\lceil \sigma^2/\varepsilon \rceil, 1\right\}\right\}$. Therefore $G(\bar{t}_k) = \bar{\delta} + G(\bar{t}_{k-1}) = k\bar{\delta}$ and $\bar{t}_k = G^{-1}(k\bar{\delta})$, where $\bar{\delta}$ is defined in (71). Using (72), we get (14). $\qquad\square$

**Example 5.6.** ["Random" Outages] Assume that

$$v_i(t) = \begin{cases} v, & t \in \bigcup_{j=1}^{\infty} [k_i(j-1), (k_i(j-1)+1)] \\ 0, & \text{otherwise,} \end{cases}, \tag{15}$$

$v > 0$, $k_i \in \mathbb{N}$, and $h_i > 0$ for all $i \in [n]$. Then

$$\bar{t}_{\lceil \frac{24L\Delta}{\varepsilon} \rceil} \approx \Theta\left(\min_{m \in [n]} \left(\frac{1}{m} \sum_{i=1}^m \frac{v}{k_{\pi_i}}\right)^{-1} \left(\frac{L\Delta}{\varepsilon} + \frac{L\Delta\sigma^2}{m\varepsilon^2}\right)\right), \tag{16}$$

where $\pi$ is a permutation such that $k_{\pi_1} \leq \cdots \leq k_{\pi_n}$.

*Proof Sketch.* We have $V_i(t) = \int_0^t v_i(\tau)d = v \times \mathbb{1}[\text{measure of intervals before the time } t] \approx vt/k_i$. We substitute it to (12) and get $\bar{t}_k := \min\left\{t \geq 0 : \sum_{i=1}^n \lfloor vt/k_i - v\bar{t}_{k-1}/k_i \rfloor \geq \max\left\{\lceil \sigma^2/\varepsilon \rceil, 1\right\}\right\}$. Using the same reasoning as in Example 5.4, one can easily get (16). $\qquad\square$

## I.2 HETEROGENEOUS SETUP

**Example 6.5.** [Fixed Computation Model in the Heterogeneous Setting] Assume that $v_i(t) = v_i$ with $v_i > 0$ for all $i \in [n]$. Then

$$\bar{t}_{\lceil \frac{24L\Delta}{\varepsilon} \rceil} = \Theta\left(\max_{i \in [n]} \frac{1}{v_i} + \left(\frac{1}{n} \sum_{i=1}^n \frac{1}{v_i}\right) \frac{\sigma^2}{n\varepsilon}\right). \tag{20}$$

*Proof.* Clearly, $V_i(t) = \int_0^t v_i d\tau = v_i t$. Substituting this to (19), we get $\bar{t}_k := \min\left\{t \geq 0 : (1/n \sum_{i=1}^n \lfloor v_i(t - \bar{t}_{k-1}) \rfloor^{-1})^{-1} \geq \max\left\{2\sigma^2/n\varepsilon, 1\right\}\right\}$. Equivalently, we have to find $\bar{\delta}$ such that

$$\bar{\delta} = \min\left\{\delta \geq 0 : 1/n \sum_{i=1}^n \lfloor v_i \delta \rfloor^{-1} \leq \min\left\{n\varepsilon/2\sigma^2, 1\right\}\right\} \tag{74}$$

and calculate $\bar{t}_k = \delta + \bar{t}_{k-1} = k\bar{\delta}$. Let us show that

$$\bar{\delta}_{1/4} \leq \bar{\delta} \leq \bar{\delta}_4 \tag{75}$$

where the $\bar{\delta}_.$ is defined as

$$\bar{\delta}_c := c \times \left(\max_{i \in [n]} \frac{1}{v_i} + \left(\frac{1}{n} \sum_{i=1}^n \frac{1}{v_i}\right) \frac{\sigma^2}{n\varepsilon}\right) \tag{76}$$

and $c > 0$ is a constant. Since $\bar{\delta}_4 \geq \max_{i \in [n]} \frac{1}{v_i}$, we have $\lfloor v_i \delta_4 \rfloor \geq \frac{v_i \delta_4}{2}$ for all $i \in [n]$ and

$$\frac{1}{n} \sum_{i=1}^n \lfloor v_i \delta_4 \rfloor^{-1} \leq \frac{1}{n} \sum_{i=1}^n \frac{2}{v_i \delta_4} \overset{(76)}{\leq} \frac{1}{n} \sum_{i=1}^n \frac{1}{1 + 2v_i \left(\frac{1}{n} \sum_{i=1}^n \frac{1}{v_i}\right) \frac{\sigma^2}{n\varepsilon}}$$

$$\leq \min\left\{\frac{1}{n} \sum_{i=1}^n \frac{1}{2v_i \left(\frac{1}{n} \sum_{i=1}^n \frac{1}{v_i}\right) \frac{\sigma^2}{n\varepsilon}}, 1\right\} = \min\left\{\frac{n\varepsilon}{2\sigma^2}, 1\right\}.$$

Thus $\bar{\delta} \leq \bar{\delta}_4$. On the other hand, let us show that $\frac{1}{n} \sum_{i=1}^n \lfloor v_i \delta_{1/4} \rfloor^{-1} > \min\left\{n\varepsilon/2\sigma^2, 1\right\}$. If $\max_{i \in [n]} 1/v_i > \left(\frac{1}{n} \sum_{i=1}^n 1/v_i\right) \sigma^2/n\varepsilon$, then $\delta_{1/4} < 1/2 \max_{i \in [n]} 1/v_i$, $\lfloor v_j \delta_{1/4} \rfloor = 0$ for $j =$

$\arg\max_{i\in[n]} 1/v_i$, and $\frac{1}{n}\sum_{i=1}^{n}\lfloor v_i\delta_{1/4}\rfloor^{-1} = \infty \geq n\varepsilon/2\sigma^2$. If $\max_{i\in[n]} 1/v_i \leq \left(\frac{1}{n}\sum_{i=1}^{n} 1/v_i\right)\sigma^2/n\varepsilon$, then since $\lfloor x\rfloor \leq x$ for all $x \geq 0$, we get

$$\frac{1}{n}\sum_{i=1}^{n}\lfloor v_i\delta_{1/4}\rfloor^{-1} \geq \frac{1}{n}\sum_{i=1}^{n}(v_i\delta_{1/4})^{-1} \overset{(76)}{\geq} \frac{1}{n}\sum_{i=1}^{n}\frac{2}{v_i\left(\frac{1}{n}\sum_{i=1}^{n}\frac{1}{v_i}\right)\frac{\sigma^2}{n\varepsilon}} > \frac{n\varepsilon}{2\sigma^2}.$$

Therefore $\frac{1}{n}\sum_{i=1}^{n}\lfloor v_i\delta_{1/4}\rfloor^{-1} > \min\{n\varepsilon/2\sigma^2, 1\}$, meaning $\bar\delta \geq \delta_{1/4}$. $\square$

**Example 6.6.** [Nonlinear Trend in the Heterogeneous Setting] Assume that $v_i(t) = v_i \times g(t)$ with $v_i > 0$ for all $i \in [n]$ and a continuous almost everywhere positive function $g(t) : \mathbb{R}_+^\infty \to \mathbb{R}_+$. Then

$$\bar{t}_{\lceil\frac{24L\Delta}{\varepsilon}\rceil} = G^{-1}\left(c_1 \cdot \left[\max_{i\in[n]}\frac{1}{v_i} + \left(\frac{1}{n}\sum_{i=1}^{n}\frac{1}{v_i}\right)\frac{\sigma^2}{n\varepsilon}\right]\right), \tag{21}$$

where $G(t) := \int_0^t g(\tau)d\tau$, and $c_1 \in [1/4, 4]$ (can depend on other parameters but is bounded).

*Proof.* We have $V_i(t) = \int_0^t v_i g(\tau)d\tau = v_i G(t)$. We substitute this equality to (19) and get $\bar{t}_k = \min\left\{t \geq 0 : \left(\frac{1}{n}\sum_{i=1}^{n}\lfloor v_i(G(t)-G(\bar{t}_{k-1}))\rfloor^{-1}\right)^{-1} \geq \max\left\{\frac{2\sigma^2}{n\varepsilon}, 1\right\}\right\}$. Since $G(t)$ is continuous and increasing, one can show that $G(\bar{t}_k) = \min\left\{t \geq 0 : \left(\frac{1}{n}\sum_{i=1}^{n}\lfloor v_i(t-G(\bar{t}_{k-1}))\rfloor^{-1}\right)^{-1} \geq \max\left\{\frac{2\sigma^2}{n\varepsilon}, 1\right\}\right\}$. Therefore $G(\bar{t}_k) = \bar\delta + G(\bar{t}_{k-1}) = k\bar\delta$ and $\bar{t}_k = G^{-1}(k\bar\delta)$, where $\bar\delta$ is defined in (74). Using (75), we get (21). $\square$

