# OpenReview forum: "Tight Time Complexities in Parallel Stochastic Optimization with Arbitrary Computation Dynamics"
_ICLR.cc/2025/Conference — ICLR 2025 Poster_

### Official Review · Reviewer_EWQJ · 2024-11-03

**Soundness:** 3
**Presentation:** 3
**Contribution:** 3
**Rating:** 5
**Confidence:** 3

**Summary:**

The paper addresses the challenges and establishes optimal time complexities for distributed stochastic optimization methods that utilize parallel and asynchronous computation. The authors introduce a universal computation model that captures the real-world scenarios of fluctuating computation speeds, hardware and network delays, and other irregularities often encountered in parallel computing environments. They apply this model to demonstrate tight lower bounds for both synchronous and asynchronous optimization methods, and highlight that these bounds are closely matched by the Rennala SGD and Malenia SGD methods.

**Strengths:**

1. The paper proposes a new computation model that more realistically simulates the computational irregularities of distributed systems, which advances over previous models that assumed stable and uniform computation speeds.
2. The analysis encompasses a broad range of stochastic optimization methods and considers both homogeneous and heterogeneous computing environments.

**Weaknesses:**

1. The theoretical results, while comprehensive, may be too complex to be applied to practical algorithms. This complexity could limit their usability for practitioners who require simpler and more accessible tools for system design and analysis. Consequently, the tight lower bounds for time complexities are primarily of theoretical interest and provide limited insight into the potential performance improvements achievable with parallel stochastic optimization methods.
2. The paper does not provide sufficient discussion on the limitations of the universal computation model. Are there any properties in parallel or distributed systems that fall outside the scope of this universal model?

**Questions:**

1. Does the computation power characterize server’s computation speed?
2. In Methods 3 and 4, is the communication time between the workers and the server taken into account? In practical systems, it may take a significantly longer amount of time for a worker to transmit a gradient than to compute it. How does the universal computation model account for this?
3. Line 209: What does $\mathcal{D}$ mean?

---

> ### Author Response · Authors · 2024-11-15
> **Official Comment by Authors**
>
> Thank you for the questions and comments!
>
> > The theoretical results, while comprehensive, may be too complex to be applied to practical algorithms. This complexity could limit their usability for practitioners who require simpler and more accessible tools for system design and analysis. Consequently, the tight lower bounds for time complexities are primarily of theoretical interest and provide limited insight into the potential performance improvements achievable with parallel stochastic optimization methods.
>
> Actually, our results have very important applications for practitioners. First of all, these theoretical results say that Rennala SGD and Malenia SGD are optimal methods under arbitrary computation dynamics. Before that, it was only known for the fixed computation model [1]. There was some skepticism in the literature (see “Other delay models,” p. 3 in [3], “Related Works,” p.2 in [2]) that Rennala SGD and Malenia SGD are optimal in other regimes, and it was an open problem that we solved. Next, as we noted in Lines 352-354 and Lines 473-476, these methods achieve the optimal time complexity without knowing $\{V_i\}.$ They automatically work in the optimal regimes, and practitioners can safely use Rennala SGD-like and Malenia SGD-like algorithms regardless of the theory.
>
> Moreover, in the supplementary material, we've added an implementation that computes the optimal time complexities numerically, which may help practitioners.
>
> Regarding the complexity of the time complexities, it might simply be fundamentally infeasible to derive simpler formulas.
>
> > The paper does not provide sufficient discussion on the limitations of the universal computation model. Are there any properties in parallel or distributed systems that fall outside the scope of this universal model?
>
> This is a good question. To be honest, we truly believe that our computation model covers almost all possible scenarios, and there are no obvious limitations right now. We've added new plots to the paper: please see Figure 2, where one can see that our new model is versatile. However, one possible limitation is that our model assumes that computation performances are statistically independent of the randomness that we get when calculating stochastic gradients. We've just added this note to the "Conclusion" section (please check the revision).
>
> > Does the computation power characterize server’s computation speed?
>
> > In Methods 3 and 4, is the communication time between the workers and the server taken into account? In practical systems, it may take a significantly longer amount of time for a worker to transmit a gradient than to compute it. How does the universal computation model account for this?
>
> Thank you for the thoughtful questions.
>
> We work with $n$ workers/nodes/servers, which have different computation powers. If you are asking about a central server that aggregates and orchestrates the workers, then we do not take its performance into account in this work because, in many scenarios, the computation of stochastic gradients is the main bottleneck.
>
> Taking into account the communication time is undoubtedly a logical direction for future research that deserves separate investigation. We feel that even our current findings are complex and require notable effort and a separate paper. In the case of the communication bottleneck, one can introduce a "universal *communication* model" using our construction and investigate this scenario also, but it is a non-trivial next research step.
>
> > Line 209:
>
> There is a typo. Thank you. We have to fix this part to $O \in \mathcal{O}(f)$ without $\mathcal{D}.$ We've just fixed it (please check the revision).
>
> ---
>
> We hope we have addressed all potential concerns and questions raised by the reviewer and added the required changes to the paper (please check the revision). If you feel this is the case, we kindly ask you to consider reevaluating our work.
>
> ---
>
> [1]: Optimal Time Complexities of Parallel Stochastic Optimization Methods Under a Fixed Computation Model, A. Tyurin et al.
>
> [2]: AsGrad: A sharp unified analysis of asynchronous-SGD algorithms, R. Islamov et al.
>
> [3]: Faster Stochastic Optimization with Arbitrary Delays via Asynchronous Mini-Batching, A. Attia et al.

---

### Official Review · Reviewer_FEY6 · 2024-11-04

**Soundness:** 3
**Presentation:** 3
**Contribution:** 2
**Rating:** 5
**Confidence:** 3

**Summary:**

This paper analyzes the time and oracle complexities of distrusted stochastic gradient descent. To approach this problem, the authors proposed a general computation model where the stochastic gradient is accessed through a set of workers that have different computational speed. In contrast to the classical oracle complexity results for SGD, this work considers an additional level of complexity that the time spent for each oracle access may be different. Then, the authors proved lower bound for the time complexity under the proposed setting and then provided two algorithms that match the lower bound.

**Strengths:**

This paper offers a very detailed analysis for distributed SGD
- I agree with the author's claim that their proposed computation model is in some sense "universal" because it greatly extends the model considered by the previous works such as [1].
- For the proposed setting, this paper offers matching lower and upper bound, thus provides a complete narrative for this problem.

I appreciate that the author does a good job explaining all of the mathematically technically and despite the verbosity required for the very general setting, the definitions in this paper do not take major effort to follow.

The introduction is direct and concise and I immediately know what kind of results I should expect to see in the technical sections.

[1] Tyurin A, Richtárik P. Optimal time complexities of parallel stochastic optimization methods under a fixed computation model. Advances in Neural Information Processing Systems. 2024.

**Weaknesses:**

In terms of organization, I suggest that the examples should be moved forward. Currently, the statements of Theorem 5.1 and 6.1 are very long and hard to digest. Having the examples immediately following these theorems would be appreciated. Also, since Them 5.1 and 6.1 are implicit due to the generality of the setting, it could be useful to make some plots for a specific problem instance (say for the fixed computation model) so it is easier to parse the implications of these results.

For equations (12) (13) and (15), I think the minimum is always at $m=1$ and therefore redundant?

The proof sketch in Section 7 is not helpful at all. In particular, I want to see the "worst case function" being spelled out. From reading the full proof, this step is highly nontrivial and I would like to see some intuition. Also, on line 486, I think you meant "first coordinate." The random variables $\eta_k$ should be defined earlier in that paragraph. Lastly, I don't understand Section 7.2 at all even after multiple re-reads, in particular, I am left with a feeling with that something important was left out between lines 517 and 522.

Lastly and most importantly, **I do not think this paper has sufficient delta from earlier work [1].**
- From what I can tell, Theorems 6.4 and A.2 from [1] correspond exactly to Examples 5.4 and 6.5. Since the problem setup is quite complicated, it is important for the authors to justify that the added generality actually leads to some meaningful implications. Not having any further examples beyond 6.5 for the heterogeneous setting does not help the cause.
- And there seem to be a major reuse of proof techniques between the two papers. The proof of Theorem 5.1 up to equation (25) is almost a line-by-line reproduction of the existing analysis in [1]. While I am in no way suggesting any ill intent from the authors, I want to hear from the authors what are the main technical difficulties going from [1] to their proofs.

In light of these observations, I am hesitant to recommend  an accept because this paper seems to a straight extension of an earlier work [1] that added a lot of mathematical complexity but without too much of new insights. I think the amount of new contribution is insufficient for a new conference paper but could be worthwhile as a journal submission. So I will give score of **5** for the time-being.

**Questions:**

I am happy to hear from the author's response to my main concern that how this work adds significant value over [1]. If the authors convince me to change my mind during the rebuttal process, then I expect the authors to incorporate those discussion into the final version of the paper.

---

> ### Author Response · Authors · 2024-11-15
> **Official Comment by Authors (Part 1)**
>
> Thank you for the suggestions and comments!
>
> > In terms of organization, I suggest that the examples should be moved forward. Currently, the statements of Theorem 5.1 and 6.1 are very long and hard to digest.
>
> Note that before Theorem 5.1, we provided an informal version exactly to facilitate understanding of the main results.
>
> > Having the examples immediately following these theorems would be appreciated.
>
> Indeed, we do not provide examples just after Theorem 5.1, but on the next page, after the non-less important discussion explaining that Rennala SGD is an optimal method, one can find our examples.
>
> > Also, since Them 5.1 and 6.1 are implicit due to the generality of the setting, it could be useful to make some plots for a specific problem instance ...
>
> We agree! Please see the revision of our paper. We've added Figures 1 and 2.
>
> > For equations (12), (13) and (15), I think the minimum is always at $m = 1$ and therefore redundant?
>
> This is not true in almost all cases. Let us explain this part and consider (12):
> $$\bar{t} = \min\limits_{m \in [n]} \left(\frac{1}{m}\sum\limits_{i=1}^m v_{\pi_i}\right)^{-1} \left(\frac{L \Delta}{\varepsilon} + \frac{L \Delta \sigma^2}{m \varepsilon^2}\right).$$
> Assume that the performances $v_i = v$ are equal. Then
> $$ \bar{t} = \min\limits_{m \in [n]} \left(\frac{1}{m}\sum\limits_{i=1}^m v\right)^{-1} \left(\frac{L \Delta}{\varepsilon} + \frac{L \Delta \sigma^2}{m \varepsilon^2}\right) = \frac{1}{v} \min\limits_{m \in [n]} \left(\frac{L \Delta}{\varepsilon} + \frac{L \Delta \sigma^2}{m \varepsilon^2}\right) = \frac{1}{v} \left(\frac{L \Delta}{\varepsilon} + \frac{L \Delta \sigma^2}{n \varepsilon^2}\right),$$
> where the minimum is attained with $m = n,$ which is reasonable because $m$ is an ''effective number'' of workers that contribute to the optimization process [1].
>
> Next, let computation performances vary according to $v_i = \frac{1}{\sqrt{i}},$ then
> $$ \bar{t} = \min\limits_{m \in [n]} \left(\frac{1}{m}\sum\limits_{i=1}^m \frac{1}{\sqrt{i}}\right)^{-1} \left(\frac{L \Delta}{\varepsilon} + \frac{L \Delta \sigma^2}{m \varepsilon^2}\right) = \Theta \left(\min\limits_{m \in [n]} \sqrt{m} \left(\frac{L \Delta}{\varepsilon} + \frac{L \Delta \sigma^2}{m \varepsilon^2}\right) \right) = \Theta \left(\min\limits_{m \in [n]} \left(\frac{L \Delta \sqrt{m}}{\varepsilon} + \frac{L \Delta \sigma^2}{\sqrt{m} \varepsilon^2}\right)\right) = \Theta \left( \max[\frac{\sigma L \Delta}{\varepsilon^{3/2}},\frac{L \Delta \sigma^2}{n \varepsilon^2}] \right)$$ where the minimum is attained with $m = \min[ \left\lceil \frac{\sigma^2}{\varepsilon} \right\rceil, n ]$.
>
> In total, we have demonstrated at least two cases where $m \neq 1$ in practical scenarios.
>
> > The proof sketch in Section 7 is not helpful at all. In particular, I want to see the "worst case function" being spelled out. ...
>
> We use the standard "worst case function" from the nonconvex world [2], which was used in numerous papers that investigate lower bounds. See [3,4,5,1]. Following all these papers, we hide the construction in the appendix (Section D) because it is somewhat well-known and, unfortunately, takes a lot of place. We believe repeating the construction of this function in the main parts of every paper is redundant. The best description one can find in the original paper [2]. But while the construction has a non-trivial structure, it should be noted that the worst-case function is essentially a variation of Nesterov's chain function (see the celebrated "Lectures on Convex Optimization" book by Yu. Nesterov).
>
> > Also, on line 486, I think you meant "first coordinate." The random variables should be defined earlier in that paragraph.
>
> We fixed this part. Thank you. Please check the changes in blue. Note that we moved "Proof Techniques" to the appendix to give space for Figures 1 and 2.
>
> ---
>
> [1]: A. Tyurin, P. Richtarik. Optimal Time Complexities of Parallel Stochastic Optimization Methods Under a Fixed Computation Model,
>
> [2]: Yair Carmon, John C Duchi, Oliver Hinder, and Aaron Sidford. Lower bounds for finding stationary points i. Mathematical Programming, 184(1):71–120, 2020.
>
> [3]: Xinmeng Huang, Yiming Chen, Wotao Yin, and Kun Yuan. Lower bounds and nearly optimal algorithms in distributed learning with communication compression. Advances in Neural Information Processing Systems, 2022.
>
> [4]: Yucheng Lu and Christopher De Sa. Optimal complexity in decentralized training. In International Conference on Machine Learning, pp. 7111–7123. PMLR, 2021.
>
> [5]: Kumar Kshitij Patel, Lingxiao Wang, Blake Woodworth, Brian Bullins, Nathan Srebro. Towards Optimal Communication Complexity in Distributed Non-Convex Optimization. Advances in Neural Information Processing Systems, 2022.
>
> [6]: R. Islamov et al. AsGrad: A sharp unified analysis of asynchronous-SGD algorithms AISTATS 2024
>
> [7]: A. Attia et al. Faster Stochastic Optimization with Arbitrary Delays via Asynchronous Mini-Batching

---

> ### Author Response · Authors · 2024-11-15
> **Official Comment by Authors (Part 2)**
>
> We now respond to the main concern "Lastly and most importantly, I do not think this paper has sufficient delta from earlier work":
>
> > Not having any further examples beyond 6.5 for the heterogeneous setting does not help the cause.
>
> We agree, so we’ve added Example 6.6 to the paper for the heterogeneous setting.
>
> > Since the problem setup is quite complicated, it is important for the authors to justify that the added generality actually leads to some meaningful implications.
>
> In total, we have five distinct examples in both homogeneous and heterogeneous settings, through which we can derive explicit formulas for the optimal time complexity. Examples 5.4 and 6.5 are included to demonstrate the recovery of the previous result [1]. The purpose of three additional scenarios, Examples 5.5, 5.6, and 6.6 (new), is to offer justification and insights into the obtained time complexities.
>
> The primary challenge is that explicit and elegant formulas (which the reviewer expects) can only be obtained for a limited set of scenarios. However, using Theorems 5.3 and 6.4, we can find the optimal time complexities numerically! Note that we’ve added Figures 1 and 2 that explain the difference between the fixed computation model and the universal computation model. The fixed computation model is very limited and captures constant computation powers (see Fig 1). The universal computation model is rich and includes virtually all possible computation scenarios (see Fig. 2 (a), Fig. 2 (b), Fig. 2 (c)). The delta between the computation models is huge. We also provide numerical computations of the time complexities and attach a jupyter notebook that reproduces the plots and the calculations (see supplementary material).
>
> Note that there was some skepticism in the literature (see “Other delay models,” p. 3 in [6], “Related Works,” p.2 in [7]) that Rennala SGD and Malenia SGD are optimal in other regimes, and it was an open problem that we solved. The previous works [6,7] are on the same page that the fixed computation model is limited.
>
> > And there seem to be a major reuse of proof techniques between the two papers. The proof of Theorem 5.1 up to equation (25) is almost a line-by-line reproduction of the existing analysis in [1]. While I am in no way suggesting any ill intent from the authors, I want to hear from the authors what are the main technical difficulties going from [1] to their proofs.
>
> Note that this is the standard "proof opening" used in all papers that develop lower bounds in the nonconvex setting (see [1,2,3,4,5] and many more other papers). However, after the opening, the proof is completely different.
> 1) Note that all lower bounds in stochastic optimization ultimately reduce to the concentration analysis of the sum of random variables. [1] approach this by analyzing the sum $\sum_{i=1}^T \min_{j \in [n]} \tau_j \eta_{ij},$ where $\eta_j$ are i.i.d. geometric random variables. In our case, we cannot directly apply this reduction anymore because the computation powers vary over time. Therefore, we found a non-trivial modification: we have to reduce the problem to the concentration analysis of the sum of indicators: $\sum_{j=1}^T \mathbb{I} [\eta_j > \frac{1}{p}]$ and investigate this sum. (we added this clarifications to the revision)
> 2) In the heterogeneous setting, the construction and allocation of copies of the "bad" functions are ''dynamic" and change through time. We have never seen such a construction in the literature, including [1-5].
> 3) The universal computation model itself is an independent and non-trivial contribution that significantly generalizes the fixed computation model [1]. Introducing the new computation model, developing new proof techniques, and integrating the new computation model with these proof techniques are all non-trivial tasks.
>
> Hence, we believe the discrepancy between the proof techniques is substantial and non-trivial.
>
> ---
>
> Overall, we agree with many comments by the reviewer and have already added the corresponding fixes/clarifications/plots to the paper and added an additional example in the heterogeneous. Unfortunately, it is very difficult to get exact formulas in most cases, but we explain that it is always possible numerically. We added new figures and clarified the proof techniques that highlight a huge gap between the previous and the new approach. We kindly ask the reviewer to read our comments and changes in the paper and reevaluate our paper if the reviewer believes we have resolved all concerns. Please ask us more questions.
>
> ---

---

> > ### Comment · Reviewer_FEY6 · 2024-11-25
> >
> > I would like to thank the authors for their detailed response. Figures 1 and 2 are certainly helpful and thank you for clarifying equations (12-15).
> >
> > However, I am still not convinced by the other changes proposed by the authors.
> > - Example 6.6 is rather trivial and does little to assist the readers.
> > - The author's claim that
> >
> > > The primary challenge is that explicit and elegant formulas (which the reviewer expects) can only be obtained for a limited set of scenarios.
> >
> > is not an acceptable excuse. It seems that reviewers uBjW and EWQJ also agree that the results are too dense to parse. As currently written, it is very difficult to appreciate the contributions of this paper without first reading the earlier work [1].
> >
> > - Lastly, I was hoping that the author could add more substantial details to their proof sketch once it is moved to the appendix. In the revision, the proof sketch for the heterogeneous setup is completely unchanged. As for the homogeneous case, I am not sure if "concentration analysis of the sum of indicators" has enough technical novelty to warrant an entirely separate paper.
> >
> > My main concern continues to be the exact amount of novelty over the previous work [1], which is also echoed by reviewer uBjW:
> > >  It also seems that much notational/computational heavy loading was already done in this published paper [1]
> >
> > Since the author do not have a satisfactory answer to this question, I am maintain my score of **5**.
> >
> > However, I do feel there is something interesting about this universal computation model, even though its setup heavily overlaps with the earlier work [1]. I believe that a journal submission (e.g. JMLR) that covers both [1] and this manuscript would be a solid option.

---

> ### Author Response · Authors · 2024-11-25
> **Official Comment by Authors**
>
> Thank you.
>
> > The primary challenge is that explicit and elegant formulas (which the reviewer expects) can only be obtained for a limited set of scenarios.
> "is not an acceptable excuse."
>
> It is not an excuse at all. We firmly believe that the general computation scenarios are so complex that it is infeasible to find a simple formula. In mathematics, there are numerous objects that lack elegant formulas, yet remain essential in many applications. For instance, consider the gamma function from mathematical analysis. Although it does not have an explicit closed-form expression, it is still a very useful function in many fields. Looking at Figure 2, it becomes evident that deriving theoretical formulas for most practical computational scenarios is simply not feasible, and it is only possible numerically.
>
> > the results are too dense to parse.
>
> The first main result is summarized in Theorem (Informal Formulation of the Lower Bound) (Lines 303–306, 319) and spans just *four lines* in the paper. It is sufficient to understand the construction of the sequence ${\underline{t}_k}.$
>
> > As currently written, it is very difficult to appreciate the contributions of this paper without first reading the earlier work [1].
>
> We have a detailed overview of [1] in Section 1.2. The gap and difference are also clear in Figures 1 and 2.
>
> > Lastly, I was hoping that the author could add more substantial details to their proof sketch once it is moved to the appendix. In the revision, the proof sketch for the heterogeneous setup is completely unchanged.
>
> The reviewer can find more substantial details in the full proofs of the corresponding results. We assure you that every step in the full proof is detailed and accompanied by comments to enhance clarity and make it easier to follow.
>
> As a reference, the full proof of the homogeneous case takes approximately four pages, while the corresponding proof sketch in Section A takes only half a page. We believe this level of detail is more than sufficient for summarizing a proof of this length.
>
> > As for the homogeneous case, I am not sure if "concentration analysis of the sum of indicators" has enough technical novelty to warrant an entirely separate paper.
>
> The "concentration analysis of the sum of indicators" is just one piece of a much larger puzzle. There are many mathematically non-trivial steps, which the reviewer can find in Section D. Additionally, introducing a new computational model and integrating this model with novel proof techniques constitute substantial and non-trivial challenges.
>
> > My main concern continues to be the exact amount of novelty over the previous work
>
> The novelty and the gap between our paper and [1] are substantial, as highlighted in i) Figures 1 and 2 of the paper, ii) "Official Comment by Authors (Part 2)" (our main rebuttal), and iii) Section A. Let us repeat the main differences again:
>
> 1. We consider a completely new computation model that encompasses a far broader range of computation scenarios compared to the previous model (see Figures 1 and 2). This is an independent and important contribution.
> 2. The previous proofs are not compatible with the new model. As a result, we develop novel proof techniques that address the concentration of entirely different sequences of random variables (see details in Section A). These results are compelling on their own and can be utilized by researchers working on developing lower bounds for optimization methods.
> 3. Developing a new computation model and discovering new proof techniques are not independent tasks. Their integration requires significant effort. This is an independent, important theoretical work.
> 4. In the heterogeneous setting, the construction and allocation of copies of the "bad" functions are ''dynamic" and change through time, an element that has not been explored in previous works.
> 5. In light of the new computation model, our proofs are self-contained and distinct, as they necessitate a completely new proof trajectory (Sections D, E, and F. Proof sketch in Section A).
>
> Considering these aspects together, we firmly believe that our contributions are non-trivial and hold significant importance for the ML, Optimization, and ICLR communities. **We provide the first optimal time complexities in distributed optimization that account for virtually any computational behavior of workers. This includes handling potential disconnections due to hardware and network delays, time-varying computational power, and any fluctuations or trends in computation speeds.** This is a very practical and theoretical question that we have successfully addressed with **new constructions, new computation model, and new proof techniques.**
>
> Thank you again for the response.

---

> > ### Comment · Reviewer_FEY6 · 2024-11-27
> >
> > Thank you for the detailed response.
> >
> > Just to clarify, I think this paper is overall well-written and technically sophisticated. **However** I believe that a stand-alone conference paper is *not* the suitable type of publication for this manuscript because too much its discussion relies on the previous paper [1].
> >
> > Regarding some of the author's rebuttal:
> >
> > > We have a detailed overview of [1] in Section 1.2. The gap and difference are also clear in Figures 1 and 2.
> >
> > The comparison made in Section 1.2 is in my opinion insufficient because this work is a superset of [1]. But such connection is not made clear in Section 1.2, and I personally had to also carefully read [1] to understand where the differences lie. While Figures 1 and 2 are certainly nice to have, I don't think they justify the overly cumbersome notation of this work. Can there be a simpler framework that still covers all of the practical examples given in this paper? I think this question is very important because the proposed "universal computation model" is way overkill for the given examples.
> >
> > > The "concentration analysis of the sum of indicators" is just one piece of a much larger puzzle.
> >
> > Then your revision to the proof sketch is misleading.
> >
> > > The novelty and the gap between our paper and [1] are substantial, as highlighted in i) Figures 1 and 2 of the paper, ii) "Official Comment by Authors (Part 2)" (our main rebuttal), and iii) Section A. Let us repeat the main differences again: [...]
> >
> > Let me re-iterate, I feel strongly that a reader cannot arrive at these conclusions without closely reading [1]. So, I do not think publishing this manuscript separately is a good idea.
> >
> > Overall, I think the authors only made the minimally-viable edits to address my concerns. Therefore, I maintain my original assessment that bundling this manuscript with [1] as a journal (e.g. JMLR) submission is the better course of action (and I think it could be a very solid journal paper).

---

> > > ### Comment · Reviewer_FEY6 · 2024-11-27
> > >
> > > Just to add, given that this paper is technically sound, I am not strongly against an acceptance. But, in my opinion, there is a much more appropriate venue.

---

> ### Author Response · Authors · 2024-11-27
> **Official Comment by Authors (Part 1)**
>
> Thank you for your participation in the discussion.
>
> > Just to clarify, I think this paper is overall well-written and technically sophisticated.
>
> Thank you!
>
> > However I believe that a stand-alone conference paper is not the suitable type of publication for this manuscript because too much its discussion relies on the previous paper [1].
>
> > The comparison made in Section 1.2 is in my opinion insufficient because this work is a superset of [1].
>
> > Let me re-iterate, I feel strongly that a reader cannot arrive at these conclusions without closely reading [1]
>
> Let us give some bits of history: The fixed computation model was proposed in [8], not in [1]. Therefore, [1] relies on [8]. In particular, [8] used the fixed computation model to justify a new analysis of Asynchronous SGD. Asynchronous SGD was proposed earlier by at least [9,10]. Thus, [8] relies on [9,10]. Asyncrounous SGD is based on the classical SGD method [11]. And so on. This iterative process demonstrates how research evolves: new work builds on previous contributions. However, every work should add non-trivial improvements. In our case, we have demonstrated in detail that our contributions are non-trivial, as we propose a new computation model, new constructions, and new proof techniques.
>
> That said, we acknowledge that our work builds upon the results of previous studies [1–11]. However, it is important to emphasize that our work is fully self-contained. **We provide all necessary notations, assumptions, definitions, and proofs independently within our framework, and these changes are sufficient for a new paper.**
>
> The reviewer says "too much its discussion relies on the previous paper [1]." Could the reviewer provide specific instances where significant reliance is evident? The discussion of [1] is only in Section 1.2, where we also discuss [8], an important work in the development of this paper. We use Theorem 5.2 and Theorem 6.3 from [1], both of which are straightforward and non-technical (they are as simple as the analysis of the classical SGD method [11], which has been conducted countless times before). However, the proofs of Theorem 5.1, Theorem 5.3, Theorem 6.1, Theorem 6.2, and Theorem 6.4, the main results of this paper, are entirely standalone. **This is completely independent work. We do not rely on any (mathematical) results from [1] when we prove the optimal time complexities. We only refer to it in Section 1.2 when discussing previous work and use two simple theorems proved countless times before—nothing more. The difference between [1] and our paper is huge, as shown in Section 3 and Figures 1 and 2.**
>
> > While Figures 1 and 2 are certainly nice to have, I don't think they justify the overly cumbersome notation of this work. Can there be a simpler framework that still covers all of the practical examples given in this paper? I think this question is very important because the proposed "universal computation model" is way overkill for the given examples.
>
> If we were aware of another, more natural framework, we would have used it. Nevertheless, we strongly believe that our new framework is already highly natural, as it is motivated by physics (see our discussion in Lines 179–182). The concepts of computation powers $\{v_i\}$ and computation work $V_i$ are intuitive and well-suited choices for describing a computation environment.
>
> > The "concentration analysis of the sum of indicators" is just one piece of a much larger puzzle. "Then your revision to the proof sketch is misleading."
>
> Why is it misleading? We do not say in Section A.1 that *the only change in the proof* is "concentration analysis of the sum of indicators." We discuss other details of the proof before formula (21), and there is no discussion about indicators. Once again, we believe half of a page is more than sufficient for summarizing a proof of 4 pages in length.
>
> ---
>
> [1]: A. Tyurin, P. Richtarik. Optimal Time Complexities of Parallel Stochastic Optimization Methods Under a Fixed Computation Model. NeurIPS 2023
>
> [6]: R. Islamov et al. AsGrad: A sharp unified analysis of asynchronous-SGD algorithms AISTATS 2024
>
> [7]: A. Attia et al. Faster Stochastic Optimization with Arbitrary Delays via Asynchronous Mini-Batching
>
> [8]: K. Mishchenko, F. Bach, M. Even, and B. Woodworth. Asynchronous SGD beats minibatch SGD under arbitrary delays. NeurIPS 2022
>
> [9]: B. Recht, C. Re, S. Wright, and F. Niu. Hogwild!: A lock-free approach to parallelizing stochastic gradient descent. NIPS 2011
>
> [10]: J. Dean, G. Corrado, R. Monga, K. Chen, M. Devin, M. Mao, M. Ranzato, A. Senior, P. Tucker, K. Yang, et al. Large scale distributed deep networks. NIPS 2012\
>
> [11]: H. Robbins and S. Monro. A stochastic approximation method. The Annals of Mathematical Statistics

---

> ### Author Response · Authors · 2024-11-27
> **Official Comment by Authors (Part 2)**
>
> Overall, we are happy that the reviewer thinks the paper is "overall well-written and technically sophisticated" and recommends it as "a very solid journal paper." We are confident that this paper is a standalone work with significant new results that justify its status as an independent contribution. We thank the reviewer for the recommendation to add Figures 1 and 2, which greatly emphasize the importance of the universal computation model.
>
> While the reviewer recommends this paper as "a very solid journal paper," we strongly believe that this is also a strong paper for the ICLR conference because the community includes a significant number of optimizers. There is a huge trend in the development of parallel and asynchronous methods due to the massive computational resources required to train modern ML models. The community will gain substantial insights from seeing the optimal time complexities. There was notable skepticism in the literature (e.g., see “Other delay models,” p. 3 in [7], and “Related Works,” p. 2 in [6]) that Rennala SGD and Malenia SGD are optimal in other regimes than the fixed computation model, and addressing this open problem is a key achievement of our work.
>
> Thank you again for the comments!

---

### Official Review · Reviewer_H1r6 · 2024-11-05

**Soundness:** 3
**Presentation:** 3
**Contribution:** 3
**Rating:** 8
**Confidence:** 3

**Summary:**

This work studies distributed stochastic optimization in a parallel and asynchronous setting. In the literature, there is an assumption that all workers operate at a stable and uniform speed, which is unrealistic in practice. This work extends the framework to account for arbitrary computational capacities of workers, capturing their instability and time-varying nature. It then analyzes this generalized framework by deriving both lower and upper bounds.

**Strengths:**

-The paper considers a realistic scenario for analyzing distributed stochastic gradient descent by introducing a universal computation model. This model is general and can capture unstable, time-varying random workers.

- It defines a class of algorithms within this new framework and derives tight lower bounds achievable by optimal algorithms.

- The paper also connects previously developed algorithms for distributed stochastic gradient descent, showing that they remain optimal in this new setting.

**Weaknesses:**

The paper lacks numerical results and experiments, which would strengthen its contributions.

**Questions:**

Do you also consider a model for communication bandwidth? From what I understand, when a worker computes a stochastic gradient, it is broadcast to all other workers. Do you account for situations where there is a communication bandwidth bottleneck or where, due to stragglers, communication may completely fail, preventing the message from being broadcast?

---

> ### Author Response · Authors · 2024-11-15
> **Official Comment by Authors**
>
> Thank you! Let us respond to the questions:
>
> > The paper lacks numerical results and experiments, which would strengthen its contributions.
>
> Our main contribution is related to lower bounds. We do not develop new methods. The optimal Rennala SGD and Malenia SGD methods have already been successfully tested in [1]. However, we've just added numerical illustrations and computation of the optimal time complexities to the paper in Figure 2.
>
> > Do you also consider a model for communication bandwidth? From what I understand, when a worker computes a stochastic gradient, it is broadcast to all other workers. Do you account for situations where there is a communication bandwidth bottleneck or where, due to stragglers, communication may completely fail, preventing the message from being broadcast?
>
> Thank you for the good question. Indeed, this is a natural future research question that requires independent work. We believe that even the current results are non-trivial and need a lot of effort and a full paper to investigate. One can use our universal computation model and develop a "universal *communication* model."
>
> \[1\]: Alexander Tyurin and Peter Richtarik. Optimal time complexities of parallel stochastic optimization ´
> methods under a fixed computation model. Advances in Neural Information Processing Systems,
> 2023

---

### Official Review · Reviewer_uBjW · 2024-11-06

**Soundness:** 4
**Presentation:** 3
**Contribution:** 3
**Rating:** 8
**Confidence:** 4

**Summary:**

The paper provides new lower bounds for parallel optimization where the workers can have arbitrary delays. While this setting has been studied in many previous works, and there are known optimal analyses of algorithms such as asynchronous SGD, the paper studies a more general computation model where the delays are not only arbitrary but can also evolve in a structured or unstructured manner over time. Under this model, the paper provides tight lower bounds for many problems of interest, such as periodic delay patterns (common in cross-device federated learning), random device outages, etc. The paper studies homogeneous and heterogeneous distributed setups, matching the best-known upper bounds (up to log factors) in both settings. Overall, the paper closes important gaps in parallel optimization and I support accepting the paper.

**Strengths:**

1. **Closes significant gaps**: The paper closes critical gaps in parallel optimization by providing lower bounds that match the convergence rates for existing algorithms, meaning they are tight. I like that the paper makes these connections explicit, providing relevant corollaries for upper bounds where needed.
2. **Good writing**: The paper's writing is clear and rigorously describes all results. In notationally heavy parts, such as while introducing the computation model, the paper provides the rough intuition for each term/unit/state, which is very helpful.
3. **Exhaustive coverage**: Another good thing is that the paper considers both homogeneous and heterogeneous settings relevant to applications like federated learning, which is uncommon in much of the literature on asynchronous optimization, which focuses on data center settings. Overall, the paper's coverage is exhaustive, with both non-convex and convex results.

**Weaknesses:**

There are no significant weaknesses in this paper, but I think the following may help improve the writing and exposition:
1. See my question about [[3]](https://arxiv.org/abs/2305.12387) below.
2. See my question about the graph-oracle framework [[4]](https://arxiv.org/abs/1805.10222). Adding a comparison in the appendix would make the paper even more exhaustive.
3. Since the paper does not consider data heterogeneity, it can not recover the homogeneous lower bound. For instance, in the fixed computational power model, (19) can not recover (12). From reading the proof, I raise this issue because the heterogeneity across the workers (in the lower bound) looks pretty adversarial, and it is unclear to me what practical settings will have such heterogeneity. Theoretically, I understand there is no reason the lower bound would not use the full power of the adversary. This is why, theoretically, I think the fully heterogeneous setting is a bit too pessimistic. For instance, if we restrict the heterogeneity to disallow arbitrary division of data blocks ($h_j$'s), the lower bound should be worse (i.e., go down).

**Questions:**

1. I am unsure if I understand the difference between Theorems 6.1 and 6.2. Why does it matter if the functions are randomized v/s not? The inner/max player can always put all their weight on the worst functions for a given algorithm.
2. Can the authors comment on what Theorem 6.2 (or 6.1) implies for the partial participation setting in Federated learning (as studied in papers such as [[1]](https://arxiv.org/abs/2008.03606), [[2]](https://openreview.net/forum?id=SNElc7QmMDe)), where sampling from a meta distribution of users is standard?
3. Could the authors provide a detailed comparison against [[3]](https://arxiv.org/abs/2305.12387) regarding the main difference in the lower bound proof techniques while highlighting which delayed feedback settings can not be captured by their lower bound? It also seems that much notational/computational heavy loading was already done in this published paper.
4. One benefit of the graph oracle framework [[4]](https://arxiv.org/abs/1805.10222) is that it is easier to describe the information flow between different oracle queries between different time steps and agents, compared to using the states like the authors do in their computational model. Could the authors describe what can be captured in their model that can not be captured in the graph oracle model? What do their lower bounds say about existing gaps in the graph oracle framework?
5. Can the authors highlight in their proof sketches or at least in the appendix why the techniques used in the heterogeneous setting can not recover the homogeneous results under some notion of data heterogeneity? For instance, second or first-order heterogeneity notions combined [[2]](https://openreview.net/forum?id=SNElc7QmMDe) could help interpolate between these two settings (given the hard instances are similar to the usual zero-distributed lower bounds). Or is there any other restriction on the adversary that would make the lower-bound construction less pessimistic?

---

> ### Author Response · Authors · 2024-11-15
> **Official Comment by Authors**
>
> Thank you for the very positive review! We now respond to the weaknesses and questions:
>
> > See my question about [3] below.
>
> Their lower bound can not capture simple scenarios when, for instance, worker 1 is the fastest in the first 10 seconds, then worker 1 is the slowest in the second 10 seconds. Their scenario can not capture fluctuation in computation times. Note that we've added Figures 1 and 2 to the paper to visualize the difference.
>
> There are many places where the proofs are different:
> 1) Note that all lower bounds in stochastic optimization ultimately reduce to the concentration analysis of the sum of random variables. [1] approach this by analyzing the sum $\sum_{i=1}^T \min_{j \in [n]} \tau_j \eta_{ij},$ where $\eta_j$ are i.i.d. geometric random variables. In our case, we cannot directly apply this reduction anymore because the computation powers vary over time. Therefore, we found a non-trivial modification: we have to reduce the problem to the concentration analysis of the sum of indicators: $\sum_{j=1}^T \mathbb{I} [\eta_j > \frac{1}{p}]$ and investigate this sum.
> 2) In the heterogeneous setting, the construction and allocation of copies of the "bad" functions are ''dynamic" and change through time.
> 3) The universal computation model itself is an independent and non-trivial contribution that significantly generalizes the fixed computation model [1]. Introducing the new computation model, developing new proof techniques, and integrating the new computation model with these proof techniques are all non-trivial tasks.
>
>
> > See my question about the graph-oracle framework [4]. Adding a comparison in the appendix would make the paper even more exhaustive.
>
> [3] have already shown that the graph oracle framework gets weaker lower bound compared to our approach and approach by [3]. See Section M and Table 3 in [3].
>
> > Since the paper does not consider data heterogeneity, it can not recover the homogeneous lower bound...
>
> That is right. We consider the homogeneous setting ($f_i = f$) and the heterogeneous setting ($f_i \neq f$), which are clearly far apart. There should be some intermediate regimes between the two settings. One interesting future research question is too choose the right similarity measure (first-order, second-order, ...) and obtain a lower bound.
>
> > I am unsure if I understand the difference between Theorems 6.1 and 6.2...
>
> Because the random functions in Theorems 6.2 depend on the randomness of the stochastic oracles ($\xi$ in (8)). This inner/max player can not choose the worst function because the inner/max player does not control the randomness in (8).
>
> > Can the authors comment on what Theorem 6.2 (or 6.1) implies for the partial participation setting in Federated learning (as studied in papers such as [1], [2]), where sampling from a meta distribution of users is standard?
>
> Good question! We haven't investigated it deeply, but one can get results for the meta-learning by taking $n \to \infty.$ Under some reasonable assumptions, every continuous measure is the limit of discrete measures. So, if you solve $\min_x E_{\eta}{[f_{\eta}(x)]},$ you can find a sequence $\eta_n$ with discrete distribution such that $\eta_n \to \eta.$ Then apply our result for $\eta_n,$ take $n \to \infty$ in (18), and get the result. We've tried to provide a non-strict explanation.
>
> > Can the authors highlight in their proof sketches or at least in the appendix why the techniques used in the heterogeneous setting can not recover the homogeneous results...
>
> By the construction in the heterogeneous setting $f = \frac{1}{n} \sum_{i=1}^S h_i$ (29), where $h_i$ depends only on a subblock $x_i$ of $x = [x_1, x_2, \dots, x_S]$ (see (28)). At every time $t$, worker 1 has access only to $h_1, \dots, h_{k_t},$ worker 2 has access only to $h_{k_t + 1}, \dots,$ and so on. It means that the workers optimize independent problems because $\{h_i\}$ depend on independent subblocks $x_i.$ So, roughly, the second-order heter of some worker $j$ is $\|\| \nabla^2 f(x) - \nabla^2 f_j(x) \|\|^2 \approx \|\| \nabla^2 f(x) \|\|^2$ because $\sum_{i} h_i$ has non-zero values only in a small number of coordinates of the vector $x.$

---

> > ### Comment · Reviewer_uBjW · 2024-12-02
> >
> > I thank the authors for responding to my questions carefully. I maintain my opinion of accepting this paper.
> >
> > I would also request the authors to discuss the graph oracle framework and the lower bounds to make the paper self-contained (that this discussion appears in [3] is not a good excuse). Please also discuss the homogeneous v/s heterogeneous issues brought up during the rebuttal, mentioning the current limitations and future directions carefully.

---

### Author Response · Authors · 2024-11-23
**Official Global Comment by Authors**

Dear Reviewers,

Thank you for taking the time to review our rebuttals. We hope they have addressed your questions and concerns. With only three days remaining in the discussion period, we would greatly appreciate your feedback on our responses.

We have carefully reviewed all comments and incorporated the necessary modifications into the revised version of our paper (see the changes in blue): 1) We have clarified our proof techniques, emphasizing their novelty (Section A); 2) We have included an additional example in the heterogeneous setting (Example 6.6); 3) In the supplementary material, we've added a code that computes the optimal time complexities numerically, which takes virtually any computations powers as input; 4) We have added Figures 1 and 2, providing a comprehensive visualization of our new computation model and theoretical results; These figures illustrate the large difference between the fixed computation model (previous model) and the universal computation model (our new model). These minor changes do not alter the original flow of the paper but rather enhance its presentation.

Please do not hesitate to reach out with any additional questions.

Thank you for your time,

Authors

---

### Meta-Review · Area_Chair_q13L · 2024-12-20

**Metareview:**

The paper provides new lower bounds for parallel optimization in scenarios where workers can experience arbitrary delays. The proposed framework accounts for arbitrary computational capacities of workers, capturing their instability and time-varying nature. The paper derives tight lower bounds for synchronous and asynchronous optimization methods in homogeneous and heterogeneous settings and highlights that these bounds are matched by Rennala SGD and Malenia SGD methods (up to logarithmic factors in some cases).

The results extend prior work [1], which focused on a "fixed computation model" and derived analogous tightness results for Rennala SGD and Malenia SGD in homogeneous and heterogeneous settings.

While all reviewers agree that this paper is technically sophisticated and sound, they debated whether the contributions beyond [1] provide sufficient novelty and value to the community. Reviewer uBjW strongly supports acceptance, arguing that the universal computation model itself is an independent and non-trivial contribution that closes important gaps in parallel optimization. In contrast, reviewer FEY6 believes that the overlap in the general setup and ideas with [1] diminishes the value of the new results, noting that the technical approach is very similar, with only some parts of the proof (e.g., concentration inequalities) requiring modification.

The opinions on the paper remained divided after the discussion.

To some extent, I agree with reviewer FEY6 that the additional novelty compared to [1] is modest, and perhaps a comprehensive journal publication would have provided the most value to the community. However, two reviewers argue that the paper can also stand alone as a publication at ICLR.

[1]: A. Tyurin, P. Richtarik. Optimal Time Complexities of Parallel Stochastic Optimization Methods Under a Fixed Computation Model, https://arxiv.org/abs/2305.12387

**Additional Comments On Reviewer Discussion:**

The discussion between the reviewers and the authors helped clarify some technical questions. The intensive exchange with reviewer FEY6, which focused on the overlap with prior work [1] and the novelty of the contributions, is summarized above.

---

### Decision · Program_Chairs · 2025-01-22

Accept (Poster)